# REMOVING ASPECT RATIO IN THE RUNNING TIME FOR CONSTRAINED $k$-CENTER CLUSTERING

## ABSTRACT

In this paper, we consider the constrained $k$-center problems. Existing algorithms for these problems often rely on optimal radius guessing strategy, leading to an overall running time that is dependent on the aspect ratio $\Delta$ (the ratio between the maximum and minimum pairwise distances). This dependency may potentially limit the scalability of the algorithms for handling large-scale datasets. To overcome the aspect ratio dependency issue, we propose a multi-scaling method. Multi-scaling partitions the clustering instance based on relative distances between data points. It then generates a set of candidate radii whose size is independent of $\Delta$, ensuring the existence of at least one radius that can closely approximate the optimal one for any constrained $k$-center instance. This narrows the search space for radius guessing and removes the running time dependency on the aspect ratio. To further improve the efficiency of multi-scaling, we introduce a problem-specific data reduction method that allows multi-scaling to operate on a smaller unweighted instance while preserving theoretical guarantees. These techniques enable us to obtain approximation results for a series of constrained $k$-center problems with near-linear running time in the data size. Empirical experiments show that our proposed methods achieve better performances compared with the SOTA algorithms on both small and large-scale clustering datasets.

## 1 INTRODUCTION

Clustering is one of the fundamental tasks in machine learning, where the objective is to partition the data into different groups such that points within the same group share high similarity as much as possible. Among various mathematical formulations, the $k$-center clustering is one of the most popular problems, aiming to minimize the maximum distances between data points and their closest centers. The $k$-center clustering ensures that data point are not too far from its assigned center, making it important for real-world applications requiring fairness and balanced resource allocations.

The standard $k$-center problem has been extensively studied over the past decades and is known to be NP-hard, even in the plane (Megiddo & Supowit, 1984; Feder & Greene, 1988). In the metric space, Gonzalez (1985) has demonstrated that a 2-approximation can be achieved with linear running time in the data size using a simple furthest-first greedy strategy, where the approximation guarantee on clustering quality even matches the theoretical lower bound of the problem. However, it was pointed out in the literature (Ding & Xu, 2015; Abbasi et al., 2023) that the standard (unconstrained) clustering problem may not fully capture the requirements in many real-world scenarios, since various constraints are frequently imposed on data points or clusters. One typical example is fair clustering in machine learning (Backurs et al., 2019; Chen et al., 2019; Chierichetti et al., 2017), which arises from algorithmic biases in practical settings and optimization tasks. These constraints can cause significant deviations of clustering partitions from the Voronoi diagram structures (a fundamental building block for clustering algorithm design, Ding & Xu (2015)), which poses considerable challenges to develop algorithms that ensure both computational efficiency and theoretical guarantees.

Clustering with additional constraints, as noted in the literature (Ding & Xu, 2015), falls into a broader class known as constrained clustering problems. In recent years, there has been growing interest in designing efficient algorithms for constrained $k$-center problems from both theoretical and practical perspectives. To address a broad range of constraints, several unified frameworks have been proposed (Goyal & Jaiswal, 2023; Abbasi et al., 2023). However, their running time grows

exponentially with parameter $k$, making them impractical for scenarios when $k$ is large. In parallel, other works focus on approximation algorithms design for specific constraints (Nguyen et al., 2022; Jones et al., 2020; Ding et al., 2019; Aghamolaei & Ghodsi, 2018; Pietracaprina et al., 2020; Huang et al., 2019; Ceccarello et al., 2019). While these approaches provide theoretical guarantees, their running time often exhibits high-order polynomial complexity or depends on the aspect ratio (i.e., the ratio between the maximum and minimum pairwise distances), making them less practical for large-scale datasets. To the best of our knowledge, there still remains a gap in fast algorithms design for various constrained $k$-center problems. In the following, we briefly outline the key challenges.

A commonly encountered challenge lies in the running time dependency on the aspect ratio, which mainly arises from optimal radius guessing. For the $k$-center objective, the solution is characterized by the smallest radius that allows all points to be covered with $k$ centers. This property makes the optimal radius especially helpful in approximation algorithm design, where a common strategy is to guess its value and cover the points with uniform-radius balls. The radius guessing and ball coverage strategies are fundamental to $k$-center problems and have been widely used in sequential (Backurs et al., 2019; Bhaskara et al., 2019; Friedler & Mount, 2010), distributed (Bateni et al., 2023; Huang et al., 2023; Li & Guo, 2018), and fully-dynamic settings (Chan et al., 2018; 2023; Biabani et al., 2024). These strategies greatly simplify the optimization tasks while enabling good approximations.

However, obtaining fast and accurate estimations for the optimal radius is highly nontrivial. A naive approach involves checking all $\Theta(n^2)$ pairwise distances, which incurs a quadratic factor loss on the running time. Alternatively, one can enumerate candidate radii within a range defined by approximate upper and lower bounds. While this approach reduces the search space, it still incurs a multiplicative $O(\log \log(n\Delta))$ overhead even if combining with binary searching strategy. Under the assumption that $\Delta$ is bounded by a polynomial function of data size, several near-linear time approximation results were known (Bhaskara et al., 2019; Friedler & Mount, 2010). However, as pointed out in the literature (Bhattacharjee & Moshkovitz, 2021), assuming a bounded $\Delta$ is too restrictive since it does not include natural inputs generated from mixture models. Moreover, in the worst case, $\Delta$ can be arbitrarily large (Nguyen et al., 2022; Bhattacharjee & Moshkovitz, 2021), which may limit the scalability of algorithms dependent on $\Delta$. Experiments show that even small datasets can have very large aspect ratios (also see Table 3 in our Appendix A.7), forcing $\Delta$-dependent algorithms to repeat radius guessing process many times (often more than 10 times) to obtain a good approximation. Therefore, a critical challenge for designing fast approximation algorithms with provable theoretical guarantees for constrained $k$-center clustering lies in minimizing the impact of the aspect ratio $\Delta$.

For constrained $k$-center problems, coresets and sampling-based methods are also commonly used techniques (Aghamolaei & Ghodsi, 2018; Pietracaprina et al., 2020; Huang et al., 2019; Ceccarello et al., 2019; Ding et al., 2019; Huang et al., 2021; 2018) for designing fast algorithms with theoretical guarantees. Coresets can provide compact representations of the data, enabling faster optimizations. However, their sizes often depend on constraint-specific parameters. For example, the coreset sizes for $k$-center with outliers are either linear in $n/z$ or $z$ ($n$ is data size and $z$ is the number of outliers, Huang et al. (2018); Ceccarello et al. (2019)), whereas those for $(\alpha, \beta)$-fair clustering are exponential in the doubling dimension (Huang et al., 2019). Furthermore, although coresets can effectively reduce the data size, they do not eliminate the dependency of the aspect ratio on the running time. On the other hand, several sampling-based methods were proposed for $k$-center with outliers (Ding et al., 2019; Huang et al., 2021). Although effective, the proposed algorithms must either open $O(k)$ centers or discard $O(z)$ outliers to ensure the theoretical guarantees. On metrics with bounded doubling dimension, several fast approximation results were proposed for $k$-center with outliers (Biabani et al., 2023; De Berg et al., 2023; Pellizzoni et al., 2023) and fair $k$-center problems (Ceccarello et al., 2024). However, the running time of these algorithms have exponential dependence in the doubling dimension, where the doubling dimension can become $\Theta(d)$ in the $d$-dimensional Euclidean space. This may limit the overall scalability in high-dimensional settings.

## 1.1 OUR CONTRIBUTION

This paper mainly focuses on constrained $k$-center problems in the $d$-dimensional Euclidean space, a setting that naturally arises in many real-world clustering applications. The primary contribution is a new framework that theoretically eliminates aspect-ratio dependence for a broad family of constrained $k$-center problems. Existing approaches typically rely on radius guessing and incur an

$O(\log\log\Delta)$ bottleneck in the running time, which may become dominant in high–aspect-ratio scenarios (as shown in previous literature). In contrast, we propose new methods that can be adapted to a series of constrained $k$-center problems with running time independent of $\Delta$ (ideally near-linear in the data size). These problems include, but are not limited to, $k$-center with outliers (Charikar et al., 2001), individual fair $k$-center (Mahabadi & Vakilian, 2020), proportionally fair $k$-center (Chen et al., 2019), and $(\alpha,\beta)$-fair $k$-center (Chierichetti et al., 2017). Our proposed methods are mainly based on a multi-scaling technique. The intuitive idea behind multi-scaling is to leverage the relative interpoint distances to partition the data into different blocks within near-linear time (see our Figure 1 for an example). These partitions can yield a set of candidate radii with size $O(n\log(nd))$, which is independent of the aspect ratio $\Delta$. For any constrained $k$-center instance, the radii set guarantees the existence of at least one radius that can closely approximate the optimal one. Thus, instead of exhaustive enumeration, multi-scaling can serve as a pre-processing step to eliminate aspect ratio dependency by generating approximate candidate radii. These radii can be integrated with any existing algorithm to compute approximate solutions with binary searching strategy, while preserving its original theoretical guarantees.

| Problem | Approximation | Time | Constraints | Ref |
|---|---|---|---|---|
| (k,z)-center | $3+\epsilon$ | $O(n^2 d\cdot\frac{\log\log(n\Delta)}{\epsilon})$ | - | Charikar et al. (2001) |
| | $2$ | $d\text{poly}(n)$ | - | Chakrabarty et al. (2020) |
| | $13+\epsilon$ | $O(nd(k+z)+d(k+z)^2\log\log(n\Delta)/\epsilon)$ | - | Malkomes et al. (2015) |
| | $5$ | $d\text{poly}(k)$ | $\|P_n^*\|=\Omega(n/k)$ $z=\Omega(n/k)$ | Huang et al. (2021) |
| | $\mathcal{A}(r)+\epsilon$ | $O(nd\log^2 n/\epsilon^2 + \mathcal{A}(n,d,k)\cdot\frac{\log(n\log d)}{\epsilon})$ | - | Ours (Multi-Scaling) |
| | $\mathcal{A}(r)+\epsilon$ | $\tilde{O}(nd(k+z)/\epsilon^2)+O(\mathcal{A}(n,d,k)\cdot\frac{\log((k+z)\log d)}{\epsilon})$ | - | Ours (Multi-Scaling with DR) |
| | $(2+\epsilon, O(\log k))$ | $O(\frac{ndk\log\log(n\Delta)}{\epsilon})$ | - | Bhaskara et al. (2019) |
| | $(2, 1+\epsilon)$ | $O(ndk/\epsilon)$ | $O(\frac{k}{\epsilon})$ centers opened | Ding et al. (2019) |
| | $(14+\epsilon, 1+\epsilon)$ | $O\left((\frac{ndk\log k}{\epsilon}+d(\frac{k\log k}{\epsilon})^2)\cdot\frac{\log\log(n\Delta)}{\epsilon}\right)$ | - | Chan et al. (2023) |
| | $(4+\epsilon, 1+\epsilon)$ | $O(\frac{ndk^3\log\log(n\Delta)}{\epsilon^2})$ | - | Biabani et al. (2024) |
| | $(\mathcal{A}'(r)+\epsilon, \mathcal{A}'(z))$ | $O(nd\log^2 n/\epsilon^2 + \mathcal{A}'(n,d,k)\cdot\frac{\log(n\log d)}{\epsilon})$ | - | Ours (Multi-Scaling) |
| | $(2\mathcal{A}'(r)+\epsilon, \mathcal{A}'(z))$ | $\tilde{O}(nd/\epsilon^2)+O(\mathcal{A}'(n,d,k)\cdot\frac{\log(kd\log n)}{\epsilon})$ | - | Ours (Multi-Scaling with DR) |
| Idv-Fair $k$-center | $(O(\log n), 7)$ | $O(dn^5k^5\log(n\Delta))$ | - | Mahabadi & Vakilian (2020) |
| | $(2+\epsilon, 3)$ | $O(n^4kd)$ | - | Negahbani & Chakrabarty (2021) |
| | $(3+\epsilon, 3)$ | high-order polynomial | - | Vakilian & Yalciner (2022) |
| | $(2+\epsilon, 2)$ | $O(n^2+ndk\cdot\frac{\log\log(n\Delta)}{\epsilon})$ | - | Ebbens et al. (2025) |
| | $(2+\epsilon, 10)$ | $O(ndk\log(n/\delta)+k^2 d/\epsilon)$ | - | Ebbens et al. (2025) |
| | $(\mathcal{A}'(r_1)+\epsilon, \mathcal{A}'(r_2))$ | $O(nd\log^2 n/\epsilon^2 + \mathcal{A}'(n,d,k)\cdot\frac{\log(n\log d)}{\epsilon})$ | - | Ours (Multi-Scaling) |
| | $(4(1+\epsilon), 22)$ | $O(ndk+dk^2\log^2(n/\eta)/\epsilon)$ | - | Ours (Multi-Scaling with DR) |
| Prop-Fair $k$-center | $(\mathcal{A}'(r_1)+\epsilon, \mathcal{A}'(r_2))$ | $O(nd\log^2 n/\epsilon^2 + \mathcal{A}'(n,d,k)\cdot\frac{\log(n\log d)}{\epsilon})$ | - | Ours (Multi-Scaling) |
| $(\alpha,\beta)$-Fair $k$-center | $4$ | high-order polynomial | 7 additive violation | Bera et al. (2019) |
| | $3+\epsilon$ | $O(ndk+\frac{\log\log(n\Delta)}{\epsilon}\cdot(ndk\Gamma+LP(nk,3nk)))$ | 0 additive violation in expectation | Harb & Shan (2020) |
| | $\mathcal{A}(r_1)+\epsilon$ | $O(nd\log^2 n/\epsilon^2+\mathcal{A}(n,d,k)\cdot\frac{\log(n\log d)}{\epsilon})$ | $\mathcal{A}(v_1)$ | Ours (Multi-Scaling) |
| | $8+\epsilon$ | $\tilde{O}(\Gamma ndk/\epsilon^2)+O(dLP(k^2\Gamma,k^2\Gamma)\log(n\log(d))/\epsilon)$ | 7 additive violation | Ours (Multi-Scaling with DR) |

Table 1: Comparison of the results for constrained $k$-center problems. Here, $n$ is the data size, $d$ is dimension, $\Delta$ is aspect ratio, $\eta$ and $\delta$ are the success probability parameters, and $\Gamma$ is the number of protected groups for $(\alpha,\beta)$-fair clustering. $LP(m_1,m_2)$ denotes the time to solve a linear program with $m_1$ variables and $m_2$ constraints. $\mathcal{A}$ denotes any radius-guessing based single-criteria algorithm, where $\mathcal{A}(n,d,k)$ is its running time for a fixed radius, and $\mathcal{A}(r_1)$ is its approximation ratio ($\mathcal{A}(v_1)$ is the fairness violation if applicable). $\mathcal{A}'$ denotes any radius-guessing based bi-criteria algorithm, where $\mathcal{A}'(n,d,k)$ is its running time for a fixed radius, and $(\mathcal{A}'(r_1), \mathcal{A}'(r_2))$ (or $(\mathcal{A}(r), \mathcal{A}(z))$ for the $k$-center with outliers) is its approximation guarantees. Results on doubling metrics are excluded since the running time is exponentially dependent on $d$. Here, $(k,z)$-center denotes the $k$-center with outliers problem, Idv-Fair $k$-center denotes the individual fair $k$-center problem, and Prop-Fair $k$-center denotes the proportionally fair $k$-center problem.

Although multi-scaling removes the aspect ratio dependency in optimal radius guessing, it still introduces a multiplicative $O(\log(n\log d))$ running time overhead, which may limit its scalability for handling large-scale datasets. To address this issue, we propose problem-specific data reduction methods to further accelerate multi-scaling and the clustering processes. The key idea behind is to construct a small set of unweighted points, called a summary, that effectively represents the original dataset. Unlike coresets, summaries incur much larger approximation loss rather than a $(1+\epsilon)$. However, they have much smaller sizes and can be constructed more efficiently, which can provide substantial speedups. We show that it suffices to apply multi-scaling on the summaries to further reduce the running time overhead for radii set construction while preserving similar theoretical

guarantees. From a theoretical perspective, the proposed summaries are proven to preserve approximation guarantees when combined with multi-scaling, yielding better runtime complexities than pure multi-scaling strategy. From a practical perspective, the summaries substantially reduce both preprocessing and clustering time, allowing our methods to scale efficiently to large-scale datasets where pure multi-scaling or existing baselines become computationally expensive.

- We propose a multi-scaling method for constrained $k$-center problems that gives the first framework eliminating aspect-ratio dependence theoretically for a broad family of constrained variants. It serves as a preprocessing step that constructs a $\Delta$-independent candidate radii set in near-linear time and can be combined with any existing radius-guessing algorithm while preserving its approximation guarantees.

- To further accelerate the clustering process, we introduce problem-specific data reduction methods that construct small summaries. By applying multi-scaling to these summaries, the running time overhead can be further reduced while maintaining similar theoretical guarantees. Data reduction is particularly effective when the number of clusters $k$ is much smaller than the data size $n$, which is common in practice. In this regime, running multi-scaling on the compressed instance significantly lowers the preprocessing cost while still eliminating all $\Delta$-dependence and maintaining similar theoretical guarantees.

Table 1 presents the comparison results with the existing work. For $k$-center with outliers, our method can achieve running time without relying on the aspect ratio $\Delta$, while matching the best known theoretical guarantee. For fair clustering problem, our algorithms are much faster than previous methods (Mahabadi & Vakilian, 2020; Negahbani & Chakrabarty, 2021; Vakilian & Yalciner, 2022; Bera et al., 2019; Harb & Shan, 2020). Experiments show that our method can achieve a 20% average reduction in running time with comparable clustering qualities. Due to space limitations, a more detailed overview of prior results is provided in Appendix A.1. We also leave the extensions of our methods to metric spaces in Appendix B. To summarize, our main contributions are as follows.

## 2 PRELIMINARIES

Throughout this paper, we use $P$ to denote the set of given data points with size $n$ in a $d$-dimensional Euclidean space. For two points $p, q \in P$, let $\delta(p, q)$ denote their Euclidean distance. For any point $p \in P$ and a set $C \subset \mathbb{R}^d$ of centers, let $\delta(p, C) = \min_{c \in C} \delta(p, c)$ be the distance from $p$ to its nearest center in $C$. The aspect ratio is defined as $\Delta = \frac{\max_{p,q \in P} \delta(p,q)}{\min_{p,q \in P, p \neq q} \delta(p,q)}$. The goal of constrained $k$-center aims to find a set $C \subset \mathbb{R}^d$ of centers while minimizing $\max_{p \in P} \delta(p, C)$ and satisfying the constraints. Let $L^*$ be the optimal clustering radius of the given instance. We use $\mathcal{H}(P) = \{P_1^*, P_2^*, ..., P_k^*\}$ and $C^* = \{c_1^*, c_2^*, ..., c_k^*\}$ to denote the sets of optimal clustering partitions and optimal clustering centers, respectively. Given a positive integer $t \in \mathbb{N}$, let $[t] = \{1, 2, ..., t\}$. For a point $c \in \mathbb{R}^d$ and radius $L$, let $\mathcal{B}_P(c, L) = \{p \in P : \delta(p, c) \leq L\}$ denote the set of points in $P$ within distance $L$ to $c$. We use $\tilde{O}$ notation to compress the polylog$(n, d)$ terms in the complexity.

**Definition 1.** $k$-*center with Outliers* (Charikar et al., 2001). *Given a dataset $P \subset \mathbb{R}^d$ and a non-negative integer $z \in \mathbb{N}$ with $z < |P|$, the goal of the $k$-center with outliers problem is to discard up to $z$ outliers while optimizing the $k$-center objective in the remaining data points.*

**Definition 2.** *Individual Fair $k$-center* (Mahabadi & Vakilian, 2020). *Given a dataset $P \subset \mathbb{R}^d$ and a parameter $\tau > 0$, the individual fair $k$-center problem aims to find a set $C \subset \mathbb{R}^d$ of $k$ centers that minimizes $\max_{p \in P} \delta(p, C)$, subject to the fairness constraint $\delta(p, C) \leq \tau r_p$ for every $p \in P$, where $r_p$ is the distance from $p$ to its $\lceil n/k \rceil$-nearest neighbor in $P$.*

**Definition 3.** *Proportionally Fair $k$-center* (Chen et al., 2019). *Given a dataset $P \subset \mathbb{R}^d$ and a parameter $\rho$, a set $X \subset \mathbb{R}^d$ with $|X| = k$ is $\rho$-proportional if $\forall S \subseteq P$ with $|S| \geq \lceil n/k \rceil$ and $\forall y \in P$, there exists a data point $i \in S$ such that $\rho\delta(i, y) \geq \delta(i, X)$. The goal of proportionally fair $k$-center is to find a set $C \subset \mathbb{R}^d$ such that $\max_{p \in P} \delta(p, C)$ is minimized and $C$ is $\rho$-proportional.*

**Definition 4.** $(\alpha, \beta)$-*Fair $k$-center* (Bera et al., 2019). *In this problem, we are given a dataset $P \subset \mathbb{R}^d$ and two fairness vectors $\vec{\alpha}$ and $\vec{\beta}$. The dataset $P$ has been divided into $\Gamma$ groups $\{\mathcal{X}_1, \mathcal{X}_2, \ldots, \mathcal{X}_\Gamma\}$, where points in each $\mathcal{X}_i$ share the same color $i \in [\Gamma]$. $(\alpha, \beta)$-fair $k$-center aims to find a set $C \subset \mathbb{R}^d$ of centers with size $k$, and an assignment $\phi : P \to C$ such that $\max_{p \in P} \delta(p, \phi(p))$ is minimized and $\alpha_l \leq \frac{|\{p \in \mathcal{X}_l : \phi(p) = c_i\}|}{|\{p \in P : \phi(p) = c_i\}|} \leq \beta_l$ holds for each $l \in [\Gamma]$ and $c_i \in C$.*

For a $k$-center with outliers instance $(P, k, d, z)$, an algorithm achieves an $(\alpha', \beta')$-approximation if it discards at most $\beta'z$ outliers and the clustering cost is at most $\alpha'L^*$. For individual fair $k$-center problem, we say that an algorithm achieves $(\alpha', \beta')$-approximation if it outputs a solution $C$ with size $k$ such that the clustering cost is at most $\alpha'L^*$, and $\delta(p, C) \leq \beta'\tau r_p$ holds for each $p \in P$. For the $(\alpha, \beta)$-fair $k$-center problem, a set $C$ of centers with an assignment $\phi$ has $f$ additive fairness violation if the number of points in any protected group violating the fairness constraints is at most $f$. We say that a data point $p \in P$ is covered by a ball centered at $c$ with radius $L$ if $p \in \mathcal{B}_P(c, L)$.

# 3 ASPECT RATIO INDEPENDENT METHODS FOR CONSTRAINED $k$-CENTER

In this section, we present new methods for the constrained $k$-center problems. The main objective is to design efficient algorithms (ideally with near-linear running time in the data size) that have running time independent of the aspect ratio $\Delta$. The proposed methods are built on a newly proposed multi-scaling technique. The high-level idea behind multi-scaling is to decompose the given instance into different partitions based on relative distances between data points. By leveraging the partitions obtained, a radii set with size $O(n \log(nd))$ can be constructed. We show that for any constrained $k$-center instance, the constructed radii set contains at least one radius that closely approximates the optimal value. Thus, multi-scaling can serve as a pre-processing step to be combined with any existing algorithm for clustering, while the theoretical guarantees can be well preserved.

## 3.1 THE MULTI-SCALING TECHNIQUE

In this subsection, we present the multi-scaling technique. Multi-scaling begins by embedding the clustering instance into a hierarchical tree structure (i.e., the HST; see Chapter 11 in Har-Peled (2011)), where each node in the tree represents a subset of data points. HST ensures bounded diameter growth between parent and child nodes (see Figure 1 for an example), which naturally induces partitions of the dataset. Based on the tree structure, multi-scaling performs a tree mapping process that assigns the nodes to clustering partitions according to the diameter variations between each node and its parent. This process iterates through all nodes, capturing clustering scales from fine-grained levels (leaf-level) to global level (root-level). Each partition is defined by a parameter $\lambda$, which controls the relative separation between clusters. The resulting diameters from the partitions span multiple clustering scales and thus can be used to approximate the optimal clustering radius.

The multi-scaling process uses two key parameters, $\lambda$ and $\rho$. The parameter $\lambda$ controls the granularity of the partition. A smaller $\lambda$ produces finer partitions, yielding more blocks with smaller gaps between them, which leads to a larger set of candidate radii and smaller approximation loss. Conversely, a larger $\lambda$ yields coarser partitions with fewer candidates and faster runtime. The parameter $\rho$ serves as a scaling factor (through $\gamma = \rho \times$ distortion) to compensate for the distortion introduced by the HST embedding. It compresses the distance range (scaled down by $\gamma$) so that the additional approximation loss caused by the embedding is absorbed. We will show that once $\rho$ exceeds a certain threshold, the distortion-induced loss becomes arbitrarily small, at the cost of only a logarithmic factor in the running time.

A closely related concept for HST is that of navigating nets (Krauthgamer & Lee, 2004) and cover trees (Beygelzimer et al., 2006). The key differences lie in the structural design and the running time complexities. Navigating nets and cover trees are designed for nearest neighbor searching. In contrast, HST is designed for maintaining fast approximations for pairwise distances. Thus, they have different tree structures. Regarding running time, navigating nets and cover trees are efficient when the intrinsic dimension (i.e., the doubling dimension) of the dataset is low. However, in a general $d$-dimensional Euclidean space, the running time may incur exponential dependence on $d$. However, HST scales near-linearly with $d$, making it more suitable for high-dimensional settings.

In the following, we give a comprehensive analysis for the proposed multi-scaling technique. The formal description for multi-scaling is given in Algorithm 1, where it mainly consists of two fundamental components: (1) HST construction (step 1 of Algorithm 1); (2) tree mapping (steps 2-8 of Algorithm 1). Given a dataset $P \subset \mathbb{R}^d$, an HST (denoted as $\mathcal{T}$) embeds $P$ into a tree structure satisfying the following properties: (1) The root node of the tree represents the whole dataset $P$, and each node in the tree is a subset $P(v) \subseteq P$ of $P$; (2) Let $Dia(P(v))$ be the diameter of $P(v)$. For each node $v$ in the tree $\mathcal{T}$, a size value $s(v)$ is associated with node $v$ such that $s(v)$ is a polynomial

---

**Algorithm 1** Multi-Scaling$(P, \lambda, \rho)$

---

**Input:** A set $P$ of dataset, parameters $0 < \lambda \le 1$, $\rho \ge 1$.
**Output:** A list $\mathcal{L}$ of candidate clustering radii.
1: Construct an HST $\mathcal{T}$ of $P$.
2: Set $\gamma = \rho \mathcal{P}_{HST}(n, d) + 1$, where $\mathcal{P}_{HST}(n, d)$ is the distortion polynomial of $\mathcal{T}$.
3: Initialize $\mathcal{D} = \emptyset$, $\mathcal{B} = \emptyset$.
4: For every non-root node $v$ of $\mathcal{T}$, enumerate in a bottom-up manner in $\mathcal{T}$ such that the children of a node is always visited earlier than the parent node.
5: **for** each $v \in \mathcal{T}$ **do**
6:      Let $v_p$ be the parent of $v$ in $\mathcal{T}$, $r_H = s(v_p)/\lambda$, and $r_L = \max\{s(v)/\lambda, s(v_p)/\gamma\}$.
7:      $\kappa(v) = \{i \in \mathbb{Z} : r_L \le (1 + \lambda)^i < r_H\}$.
8:      For each integer $t \in \kappa(v)$, if there is no bucket in $\mathcal{B}$ associated with an integer $t \in \kappa(v)$, then construct a bucket $b(t) = \{v\}$ and add $b(t)$ to $\mathcal{B}$. Otherwise, directly insert $v$ into $b(t) \in \mathcal{B}$.
9: $\mathcal{D} = \{t \in \mathbb{Z} : b(t) \in \mathcal{B}\}$, $\mathcal{L}(\mathcal{D}) = \{\lambda(1 + \lambda)^{t+1} : t \in \mathcal{D}\}$, $\mathcal{L} = \mathcal{L}(\mathcal{D}) \cup \{0\}$.
10: **return** $\mathcal{L}$.

---

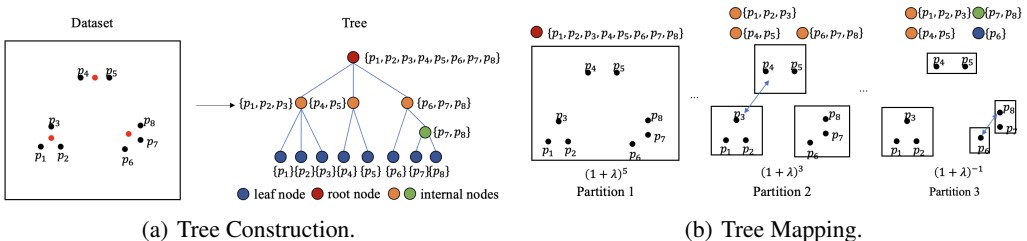

(a) Tree Construction.             (b) Tree Mapping.

Figure 1: An illustration of the multi-scaling process

approximation of $Dia(P(v))$, i.e., $Dia(P(v)) \le s(v)$, and $s(v)/Dia(P(v))$ is a polynomial function $\mathcal{P}_{HST}(n, d) \ge 1$ (which we call the distortion polynomial); (3) Every leaf node $v$ of the tree $\mathcal{T}$ contains a single data point $p \in P$ with size $s(v) = 0$; (4) For any two children $v_1$ and $v_2$ of a tree node $v$, it satisfies that $\max\{s(v_1), s(v_2)\} < s(v)$; (5) For every node $v$ in the tree $\mathcal{T}$ with its parent node $v_p$, $s(v_p)/r_{out}$ is bounded by the distortion polynomial $\mathcal{P}_{HST}(n, d)$, where $r_{out}$ is the minimum pairwise distance between points in $P(v)$ and $P \backslash P(v)$.

As shown in the literature (Har-Peled, 2011; Huang & Xu, 2022), an HST can be constructed in time $O(dn \log^2 n)$ (see Chapter 11, Theorem 11.18, and Lemma 11.19 in Har-Peled (2011) for details) with $\mathcal{P}_{HST}(n, d) = dn$. Due to space limitations, we deliver all the proofs to the Appendix.

**Lemma 1.** (Har-Peled, 2011) *An HST can be constructed in time $O(dn \log^2 n)$ with a distortion polynomial $\mathcal{P}_{HST}(n, d) = dn$, and the number of tree nodes in $\mathcal{T}$ can be upper bounded by $O(n)$.*

Next, we introduce the tree mapping strategy. The intuitive idea behind is as follows: given an HST (denoted as $\mathcal{T}$) of a dataset $P$ with parameter $\rho \ge 1$, tree mapping constructs buckets based on the nodes in $\mathcal{T}$, where each bucket $b(t)$ consists of clusters that defines a partition of $P$ based on the relative distance scale $(1 + \lambda)^t$. Within a specific bucket $b(t)$, clusters are well-separated under this distance scale $(1 + \lambda)^t$, and the union of the clusters nearly covers the full dataset $P$ except for some points that lie far from all clusters. This ensures that inter-cluster distances within the same bucket remain at least on the order of $(1 + \lambda)^t$, where the partitions obtained can naturally be used for candidate radii set construction. During the tree mapping process, the buckets are created via a bottom-up traversal of the tree $\mathcal{T}$, visiting each node once. Thus, the running time of tree mapping is only dependent on the number of tree nodes. The following lemma shows that the full bucket list $\mathcal{B}$ can be constructed in $O(nd \log^2(n)/\lambda^2)$ time, with size $|\mathcal{B}| = O(n \log(nd)/\lambda^2)$.

**Lemma 2.** *Algorithm 1 takes time $O(nd \log^2(n)/\lambda^2)$ to construct an integer list $\mathcal{D}$ and a bucket list $\mathcal{B}$ with $|\mathcal{D}| = O(n \log(nd)/\lambda^2)$ and $|\mathcal{B}| = O(n \log(nd)/\lambda^2)$.*

Using HST and tree mapping, the radii set $\mathcal{L}$ (step 9 of Algorithm 1) is constructed by converting clustering partitions into candidate radii. Specifically, for each integer $t$ associated with a bucket

$b(t)$ constructed during tree mapping, a radius $r_t = \lambda(1 + \lambda)^{t+1}$ is added to $\mathcal{L}$. This construction is motivated by the inherent hierarchical structure of HST. For any pair of points $q_1, q_2 \in P$, traversing from their leaf nodes up the tree ensures that they are eventually merged at a common ancestor. Intuitively, the bounded diameter growth between parent and child nodes ensures that at least one bucket captures the scale at which the pair is well separated and then merged. The relative distance at this scale (associated with the buckets) can provide a good approximation of their true distance. The following lemma shows that, for a given dataset $P \subset \mathbb{R}^d$, multi-scaling can approximate any pairwise distance by leveraging diameter changes along their paths to the root in the HST.

**Lemma 3.** *Let $P \subset \mathbb{R}^d$ be a given dataset. For each pair of points $q_1, q_2 \in P$, let $l_{\min}$ be an approximate distance lower bound such that $l_{\min} \leq \delta(q_1, q_2) \leq \Psi l_{\min}$ holds for some $\Psi > 1$. $\mathcal{L}$ contains at least one radius $L \in \mathcal{L}$ such that $l_{\min} \leq L \leq (1+\lambda)l_{\min}$ by setting the input parameter $\rho$ for Multi-Scaling as $\rho \geq 2\Psi$.*

By Lemma 3, multi-scaling provides accurate approximations for pairwise distances. We now show that there exists at least one pair of points $p', q' \in P$ such that $\delta(p', q')$ can serve as a good estimate for the optimal clustering radius $L^*$. Given a constrained clustering instance $P$, let $P^*_{\max} \in \mathcal{H}(P)$ be the optimal cluster with the largest radius. Then, there are two cases that may occur: (1) there exists a pair of points $p', q' \in P^*_{\max}$ such that $\delta(p', q') \geq L^*/2$; or (2) all pairs $p, q \in P^*_{\max}$ satisfy $\delta(p, q) < L^*/2$. In case (1), the triangle inequality gives $\delta(p', q') \leq 2L^*$, where setting $l_{\min} = L^*/2$ yields $l_{\min} \leq \delta(p', q') \leq 4l_{\min}$. In case (2), observe that replacing $P^*_{\max}$ with a ball centered at any $p' \in P^*_{\max}$ would yield a new cluster with radius strictly smaller than $L^*$ (we will show that this also holds for individual fair $k$-center problem in Appendix). This contradicts with the definition of $P^*_{\max}$ that it has the optimal clustering radius $L^*$, which implies that case (1) must hold. Therefore, there exists at least a pair of points $(p', q') \in P$ such that $\delta(p', q')$ lies within range $[l_{\min}, 4l_{\min}]$ for some $l_{\min} = L^*/2$. Using Lemma 3 and setting the multi-scaling parameter $\rho$ as $\rho = 8$, Algorithm 1 guarantees that the radii set $\mathcal{L}$ contains a value $L \in \mathcal{L}$ such that $L^* \leq L \leq (1+\lambda)L^*$.

**Corollary 1.** *By setting $\rho = 8$, Algorithm 1 can return a radii set $\mathcal{L}$ in time $O(nd \log^2 n/\lambda^2)$ such that there exists at least one radius $L \in \mathcal{L}$ with $L^* \leq L \leq (1+\lambda)L^*$ for any constrained $k$-center instance $(P, k, d)$.*

As a direct consequence of Corollary 1, multi-scaling can serve as a pre-processing step to generate a set of radii with size $O(n \log(nd)/\lambda^2)$ in time $O(nd \log^2(n)/\lambda^2)$ (or $\tilde{O}(nd/\lambda^2)$). The generated radii set guarantees that there exists at least one radius that can closely approximate the optimal value. Therefore, it can then be combined with any existing algorithm that requires radius guessing to compute the final result. With binary searching strategy, the overhead for radius guessing can be reduced to $O(\log(n \log d))$ while incurring only an additive $\epsilon$ factor loss on clustering quality.

**Corollary 2.** *Let $\mathcal{A}$ be an $\mathcal{A}(r_1)$-approximation algorithm (or bi-criteria $(\mathcal{A}(r_1), \mathcal{A}(r_2))$-approximation) for a constrained $k$-center problem that relies on radius guessing with running time $T(n, d, k)$ for a fixed radius. By combining $\mathcal{A}$ with multi-scaling, an $(\mathcal{A}(r_1) + \epsilon)$-approximation (or $(\mathcal{A}(r_1) + \epsilon, \mathcal{A}(r_2))$-approximation) can be achieved in time $O\left(\frac{nd \log^2 n}{\epsilon^2} + T(n, d, k) \cdot \frac{\log(n \log d)}{\epsilon}\right)$.*

### 3.2 Problem-Specific Data Reduction

In the last subsection, we showed that a set of candidate radii with size independent of $\Delta$ can be constructed in near-linear time. By combining with existing algorithms, this helps to remove the aspect ratio dependency for constrained $k$-center problems. However, even if combining with binary search, it still incurs a multiplicative $O(\log(n \log d))$ overhead. To further speed up the radius searching process, we propose problem-specific data reduction methods. Intuitively, we construct summaries as small subsets of unweighted points that closely approximate the original dataset, enabling faster multi-scaling by running the algorithms on the summaries. Due to space constraints, we use $k$-center with outliers as an example to illustrate how data reduction accelerates the radius searching process. Detailed analyses for other problems are provided in Appendices A.4–A.6.

Given a $k$-center with outliers instance $(P, k, z, d)$, Algorithm 2 (MS-DR: Multi-Scaling with Data Reduction) outlines how to construct a set of candidate radii via data reduction and multi-scaling. The key idea is to compress $P$ into a compact summary with size $k + z$ in $O(nd(k + z))$ time. The optimal radius can then be estimated from either the distances between data points and their

closest summary representatives or from pairwise distances within the summary. We will show that applying multi-scaling to this summary yields a desired candidate radii set of size $\tilde{O}(k + z)$.

As demonstrated in the proposed algorithm, in step 2 of Algorithm 2, a summary $U$ is constructed using the fast approximation scheme from Gonzalez (1985), which solves the standard $k$-center problem using greedy strategy with a 2-approximation in $O(ndk)$ time. By replacing $k$ with $k + z$, this yields a 2-approximation for $k$-center with outliers in time $O(nd(k + z))$.

---

**Algorithm 2** MS-DR$(P, k, d, z, \lambda)$

---

**Input:** A $k$-center with outliers instance $(P, k, d, z)$, a parameter $0 < \lambda \leq 1$.
**Output:** A set $\mathcal{L}$ of candidate clustering radii.
1: Initialize $\mathcal{L} = \emptyset$.
2: Call the 2-approximation algorithm from Gonzalez (1985) to compute a $(k + z)$-center solution in $O(nd(k + z))$ time, and let $U$ be the returned center set.
3: $L_1 = \max_{p \in P} \delta(p, U)$.
4: $\mathcal{L}_1 = \{(1 + \lambda)^i : L_1/2 \leq (1 + \lambda)^i \leq 3L_1, i \in \mathbb{Z}\} \cup \{L_1/2\} \cup \{3L_1\}$.
5: $\mathcal{L}_2 = \text{Multi-Scaling}(U, \lambda, 36)$.
6: $\mathcal{L}_3 = \{3L' : L' \in \mathcal{L}_2\}$.
7: $\mathcal{L} = \mathcal{L}_1 \cup \mathcal{L}_3$.
8: **return** $\mathcal{L}$.

---

**Lemma 4.** *(Gonzalez, 1985) For the standard $k$-center problem, there exists an algorithm that can return a 2-approximate solution within time $O(ndk)$.*

**Corollary 3.** *For the $k$-center with outliers, there exists an algorithm that can achieve 2-approximation by opening $(k + z)$ centers, where the running time is $O(nd(k + z))$.*

According to Corollary 3, the set $U$ constructed before step 3 of Algorithm 2 guarantees a 2-approximation in clustering quality. Thus, for every $p \in P$, we have $\delta(p, U) \leq 2L^*$, where $L^*$ is the optimal clustering radius. Let $L_1 = \max_{p \in P} \delta(p, U)$ (step 3 of Algorithm 2) denote the maximum distance from points in $P$ to $U$. Then, there are two subcases that may happen: (1) $L_1 \geq \frac{L^*}{3}$; (2) $L_1 < \frac{L^*}{3}$. If subcase (1) happens, since $\frac{L^*}{3} \leq L_1 \leq 2L^*$, we obtain both upper and lower bounds on $L^*$ as $\frac{L_1}{2} \leq L^* \leq 3L_1$. By enumerating integer powers of $(1 + \lambda)$ within range $[\frac{L_1}{2}, 3L_1]$, we can construct a set $\mathcal{L}_1$ of candidate radii (step 4 of Algorithm 2). Since the interval $[\frac{L_1}{2}, 3L_1]$ includes $L^*$, there must exist at least one radius $L_f \in \mathcal{L}_1$ such that $L^* \leq L_f \leq (1 + \lambda)L^*$.

**Lemma 5.** *If $L_1 \geq \frac{L^*}{3}$, there exists at least one radius $L_f \in \mathcal{L}_1$ such that $L^* \leq L_f \leq (1 + \lambda)L^*$.*

Next, we consider a more complicated case (subcase (2)) where $L_1 < \frac{L^*}{3}$. We first show that there exists at least one pair of points $p', q' \in U$ such that $\frac{L^*}{3} \leq \delta(p', q') \leq 6L^*$. Let $p_{\max}, q_{\max} \in P^*_{\max}$ be the pair of points in $P^*_{\max}$ with the maximum pairwise distance. For each data point $p \in P$, denote $s_p$ as the closest representation in $U$ to $p$. Since $\delta(p_{\max}, q_{\max}) \geq L^*$, we have $\delta(s_{p_{\max}}, s_{q_{\max}}) \geq \delta(p_{\max}, q_{\max}) - \delta(p_{\max}, s_{p_{\max}}) - \delta(q_{\max}, s_{q_{\max}}) \geq \frac{L^*}{3}$, where the first inequality follows from triangle inequality. On the other hand, it holds that $\delta(s_{p_{\max}}, s_{q_{\max}}) \leq \delta(p_{\max}, s_{p_{\max}}) + \delta(p_{\max}, q_{\max}) + \delta(q_{\max}, s_{q_{\max}}) \leq 6L^*$. Hence, we have $\frac{L^*}{3} \leq \delta(s_{p_{\max}}, s_{q_{\max}}) \leq 6L^*$. Then, by executing a multi-scaling process on $U$ and setting the input parameter $\rho$ for multi-scaling as $\rho \geq 36$, it can be guaranteed that the radii set $\mathcal{L}_2$ constructed (step 5 of Algorithm 2) should contain at least one radius $L'' \in \mathcal{L}_2$ that can well approximate the optimal clustering radius $L^*$.

**Lemma 6.** *If $L_1 < \frac{L^*}{3}$, there exists at least one radius $L'' \in \mathcal{L}_2$ such that $\frac{L^*}{3} \leq L'' \leq (1 + \lambda)\frac{L^*}{3}$.*

According to Lemma 6, it is obvious that there exists at least one radius $L' \in \mathcal{L}_3$ (obtained in step 6 of Algorithm 2) such that $L^* \leq L' \leq (1 + \lambda)L^*$. Putting all these together, a radii set with size $O((k + z) \log((k + z)d)/\lambda^2)$ can be constructed in time $O((k + z)(nd + d \log^2(k + z)/\lambda^2))$, which contains one radius that well approximates the optimal clustering radius.

**Lemma 7.** *Given a parameter $0 < \lambda < 1$, for a $k$-center with outliers instance $(P, k, z, d)$ where $|P| = n$, a radii set $\mathcal{L}$ with size $O((k + z) \log((k + z)d)/\lambda^2)$ can be constructed in time $O((k + z)(nd + d \log^2(k + z)/\lambda^2))$ such that there exists one radius $L \in \mathcal{L}$ satisfying $L^* \leq L \leq (1 + \lambda)L^*$.*

We now present the algorithm for $k$-center with outliers using data reduction, which is referred as the FKOC algorithm (Fast $k$-center with Outliers Clustering). Due to space limitations, the formal description for FKOC is presented in Appendix A.3. FKOC combines data reduction and multi-scaling to achieve near-linear running time in data size. Specifically, when the number of outliers is large (i.e., $z \geq k^2 \log n/\lambda^2$), it applies a sampling scheme (Lemma 8) to obtain a small subset $S$ for data compression. Then, a summary-based multi-scaling is executed on $S$ to construct a candidate radii set. Finally, any radius-guessing based algorithm $\mathcal{F}$ (ideally with linear running time in data size) can be adapted to compute the final solutions, where only a 2-approximation loss is incurred.

**Lemma 8.** *(Charikar et al. (2003)) Let $(P, k, d, z)$ be a $k$-center with outliers instance, and let $S \subseteq P$ be a random subset of size $O\left(\frac{nk \log n}{\lambda^2 z}\right)$ for some $\lambda \leq \frac{1}{6}$. Define a new instance $(S, k, z', d)$ with $z' = (1 + \lambda)\frac{z|S|}{n}$. If algorithm $\mathcal{A}$ returns a $\zeta$-approximate solution on $S$ while discarding at most $(1 + \lambda)z'$ outliers, then the solution returned by $\mathcal{A}$ on $S$ can achieve $2\zeta$-approximation on $P$ with $\left(1 + \frac{(1+\lambda)^2}{1-\lambda}\right) z$ outliers discarded.*

According to Lemma 8, since $\lambda \leq 1/6$, we have $\frac{1}{1-\lambda} \leq (1 + 2\lambda)$. Hence, it holds that $\frac{(1+\lambda)^2}{1-\lambda} \leq (1 + 2\lambda)(1 + \lambda)^2 \leq 1 + 12\lambda$. By replacing $\lambda$ with $\epsilon_1$ for some $\epsilon_1 \leq \frac{\lambda}{12}$, the number of outliers discarded can be carefully bounded. Putting all these together, we can get the following result.

**Theorem 1.** *For $k$-center with outliers, let $\mathcal{F}$ be any radius-guessing bi-criteria algorithm with running time $\mathcal{F}(n, d, k)$ for a fixed radius, where the approximation ratio for $\mathcal{F}$ is $(\mathcal{F}(r_1), \mathcal{F}(r_2))$. Then, there exists an algorithm that returns a $(2\mathcal{F}(r_1) + \epsilon, \mathcal{F}(r_2))$-approximate solution in time $\tilde{O}(nd/\epsilon^2) + O(\mathcal{F}(n, d, k) \cdot \frac{\log(kd \log n)}{\epsilon})$.*

## 4 EXPERIMENTS

In this section, we present our experimental results, where all the experiments are conducted on a 72-core Intel Xeon Gold 6230 machine with 500GB memory. Due to space limitations, we only present the results for $k$-center with outliers here and leave other experiments to Appendix A.7.

**Datasets** We conduct the experiments on 3 small datasets (NIPS: $11,463 \times 50$, SKIN: $245,057 \times 3$, COVERTYPE: $581,012 \times 54$), and 3 large-scale datasets (SUSY: $5,000,000 \times 18$, HIGGS: $11,000,000 \times 27$, SIFT: $100,000,000 \times 128$). These datasets have been widely used in previous work related to $k$-center with outliers (Bhaskara et al., 2019; Huang et al., 2023; Li & Guo, 2018).

**Algorithms and Parameter Settings.** To ensure a fair comparison, we evaluate both our pure multi-scaling method (denoted as "Ours") and the data-reduction–based multi-scaling method (denoted as "Ours + DR"), with each combined with an existing clustering algorithm. The comparison set includes a Greedy algorithm (Bhaskara et al., 2019), a Two-Stage clustering algorithm (Chan et al., 2018), and a sampling-based algorithm (Biabani et al., 2024), while algorithms with quadratic running time or strong assumptions are excluded (such as Charikar et al. (2001); Ding & Xu (2015); Chakrabarty et al. (2020); Huang et al. (2021)). To validate the benefit of the data-reduction step itself and to enable a fair comparison, we also consider a variant that applies our data reduction to an existing method. In particular, we use Greedy (the existing fastest algorithm) as an example and denote this baseline as "Greedy + DR". In our experiments, the multi-scaling process is further accelerated by stopping tree construction once a node contains fewer than $\frac{\xi n}{k}$ points (with $\xi = 0.01$), after which we apply the Greedy algorithm (the fastest existing method) to produce the final clustering. For all algorithms, we fix $\lambda = 0.1$. In Appendix A.7, we report additional evaluations on the impact of stopping criteria and parameter choices.

**Experimental Setup.** In our experiments, datasets are normalized to $[0, 1]^d$ as a standard pre-processing step. Each algorithm is run 10 times, and we report the mean with deviation. Following (Bhaskara et al., 2019), the number of outliers $z$ varies from $1\% n$ to $5\% n$ and $k$ varies from 10 and 50, a setting sufficient to capture representative behavior. For algorithms requiring radius guessing, we adopt the SOTA method (Cohen-Addad et al., 2022) to estimate the bounds for $L^*$. Each algorithm discards exactly the furthest $z$ outliers based on its final centers to ensure fairness.

**Results.** Tables 2 and 11 (Appendix A.7) present the clustering results on dataset SIFT. For such a dataset with 100 million points, only Greedy (Bhaskara et al., 2019) and our algorithms can return

| Dataset | Method | $k$ | Cost | Time(s) |
|---------|--------|-----|------|---------|
| SIFT | Ours
Greedy
Ours + DR
Greedy + DR | 10 | $0.5304 \pm 0.0067$
$0.5109 \pm 0.0008$
$0.5207 \pm 0.0005$
$\mathbf{0.5092 \pm 0.0036}$ | $1585.77 \pm 91.85$
$3221.04 \pm 103.41$
$\mathbf{39.51 \pm 6.45}$
$1162.71 \pm 428.22$ |
| SIFT | Ours
Greedy
Ours + DR
Greedy + DR | 20 | $0.5185 \pm 0.0007$
$\mathbf{0.4811 \pm 0.0005}$
$0.4822 \pm 0.0011$
$0.4852 \pm 0.0018$ | $2309.70 \pm 120.25$
$4627.59 \pm 193.38$
$\mathbf{31.57 \pm 9.86}$
$856.07 \pm 49.49$ |
| SIFT | Ours
Greedy
Ours + DR
Greedy + DR | 30 | $0.5025 \pm 0.0029$
$0.4704 \pm 0.0004$
$\mathbf{0.4703 \pm 0.0006}$
$0.4712 \pm 0.0024$ | $3326.75 \pm 59.22$
$5905.89 \pm 76.39$
$\mathbf{81.11 \pm 15.14}$
$898.63 \pm 45.53$ |
| SIFT | Ours
Greedy
Ours + DR
Greedy + DR | 40 | $0.4950 \pm 0.0021$
$0.4656 \pm 0.0003$
$0.4633 \pm 0.0011$
$\mathbf{0.4630 \pm 0.0010}$ | $4354.95 \pm 122.67$
$6869.85 \pm 543.30$
$\mathbf{118.51 \pm 17.38}$
$966.45 \pm 21.28$ |
| SIFT | Ours
Greedy
Ours + DR
Greedy + DR | 50 | $0.4615 \pm 0.0002$
$0.4575 \pm 0.0013$
$0.4574 \pm 0.0001$
$\mathbf{0.4559 \pm 0.0017}$ | $5276.93 \pm 100.73$
$8607.47 \pm 52.69$
$\mathbf{86.66 \pm 9.43}$
$969.24 \pm 36.74$ |

Table 2: Comparison results on the SIFT dataset with varying $k$ and fixed $z = 1\%n$, where algorithms with running time exceeding 24 hours are excluded.

a feasible solution within 24 hours, while other algorithms fail. Comparing only methods without data reduction, on average, our algorithm is 1.8 times faster than Greedy with comparable clustering quality. Tables 9-10 (in Appendix A.7) present results for other large-scale datasets, where our pure multi-scaling method achieves an average of 1.52 times speedup with only a 2.1% increase in clustering cost. For small datasets (Tables 5-7 in Appendix A.7), on average, the pure multi-scaling method reduces clustering cost and running time by 7.6% and 20%, respectively. Overall, as dataset sizes increase, multi-scaling scales more efficiently than the existing approaches.

When incorporating our data-reduction step, the efficiency gains become much more pronounced. On the SIFT dataset, our data-reduction–based multi-scaling method reduces the running time from thousands of seconds for Greedy to tens of seconds, yielding over 10x speedup with comparable clustering quality. Although data reduction also accelerates Greedy, our multi-scaling variant still achieves an average of 8x speedup over data-reduction–based Greedy, showing that the gains are not due to data reduction alone but also to the $\Delta$-independent radii construction. This is consistent with the observation that, on such large-scale data, estimating the radius range can dominate the runtime. For the other datasets (NIPS, SKIN, SUSY, HIGGS, and COVERTYPE), our method is also the fastest in most cases, achieving on average about 5x speedup over Greedy and roughly 2x speedup over data-reduction–based Greedy, while maintaining comparable or slightly better clustering quality. These results indicate that combining multi-scaling with data reduction is highly effective for constrained $k$-center clustering across diverse datasets, and is particularly beneficial on large-scale, high–aspect-ratio instances.

## 5 CONCLUSIONS AND DISCUSSIONS

This paper presents fast approximation algorithms for a series of constrained $k$-center problems with running time independent of the aspect ratio $\Delta$. Our framework gives, to the best of our knowledge, the first way to remove aspect-ratio dependence for a broad family of constrained $k$-center variants, while preserving the approximation guarantees. Experiments show that our methods scale better than prior approaches, especially on large datasets.

A natural direction for future work is to extend multi-scaling beyond $k$-center, for example to $k$-median and $k$-means. For these objectives, aspect-ratio dependence typically arises from unstable initializations rather than radius guessing, so an extension would require constructing multi-scale candidate partitions instead of radii lists. Another direction is to adapt the multi-scaling and summary-based pipeline to other computational models, such as distributed and streaming clustering. We leave these extensions to other clustering objectives and to distributed/streaming scenarios as promising topics for future work.

## ETHIC STATEMENT

This work focuses on fast algorithms design for constrained $k$-center problems, aiming to remove aspect-ratio dependence from the running time. The contributions are purely algorithmic and theoretical, with experiments conducted on standard benchmark datasets, and we do not anticipate negative societal impacts. While clustering methods can influence applications where fairness or privacy is important, our study does not involve sensitive data, human subjects, or identifiable information, and no specific ethical concerns arise beyond these general considerations.

## REPRODUCIBILITY STATEMENT

We have made extensive efforts to ensure the reproducibility of our results. All theoretical claims are stated with precise assumptions and supported by complete proofs in the appendix. The algorithms are described in detail in the main text, with pseudocode included for clarity. Experimental settings, parameter choices, and evaluation protocols are clearly discussed in both the main context and appendix. We use only publicly available benchmark datasets, and the data processing steps are fully described in the main context and appendix.

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

# A  APPENDIX: MISSING PROOFS AND COMPLEMENTARY EXPERIMENTS

## A.1  INTRODUCTION TO THE EXISTING ALGORITHMS FOR CONSTRAINED $k$-CENTER

In the following, we summarize the approximation results for the constrained $k$-center problems studied in this paper.

$k$**-center with Outliers Problem.**   The $k$-center with outliers problem was motivated by Charikar et al. (Charikar et al., 2001), where a given number $z$ of data points can be discarded as outliers when trying to optimize the clustering cost. Charikar et al. (2001) first gave a deterministic 3-approximation algorithm in metric space using greedy ball coverage strategy, which has polynomial running time. Chakrabarty & Negahbani (2019) proposed a reduction-based method for metric $k$-center with outliers problem, which achieves 2-approximation in polynomial time and matches the lower bound of inapproximability for the $k$-center problem (Gonzalez, 1985). It was also proved in the literature (Grunau & Rozhoň, 2022) that achieving any constant approximation with exactly $z$ outliers discarded for the $k$-center with outliers problem requires running time of $\Omega(z^2)$. When the given dataset has heavy noise, i.e., $z = \Omega(n)$, the lower bound running time becomes quadratic. Thus, on the practical side, by relaxing the number of centers opened or the number of outliers discarded, several fast approximation schemes were proposed. Ding et al. (2019) gave a sampling-based algorithm that can achieve a 2-approximate solution in time $O(ndk/\epsilon)$ with $O(k/\epsilon)$ centers opened and $(1 + \epsilon)z$ outliers discarded. Bhaskara et al. (2019) proposed a greedy method that can achieve a $(2 + \epsilon)$-approximate solution in time $O(ndk \log \log(n\Delta)/\epsilon)$ with exactly $k$ centers opened and $O(z \log k)$ outliers discarded, where the $O(\log \log(n\Delta))$ term is the loss for optimal radius guessing process. Recently, a lower bound running time of $O(nk^2/z)$ for achieving any $O(1)$-approximate solution with $O(z)$ outliers discarded was proved by Grunau and Rozhoň (Grunau & Rozhoň, 2022). In distributed settings, there are several approximation results that can achieve constant approximate solutions with $(1+\epsilon)z$ outliers discarded (Li & Guo, 2018; Huang et al., 2023). However, either the communication cost (Li & Guo, 2018) or the running time on the coordinator (Huang et al., 2023) has linear dependence in an $O(\log(n\Delta))$ term caused by optimal radius guessing process. In fully dynamic settings, a $(14 + \epsilon)$-approximate solution can be achieved with $(1 + \epsilon)z$ outliers discarded and amortized update time $O(\frac{k^2 \log(n\Delta)}{\epsilon})$ assuming prior knowledge on the range for $\Delta$ (Chan et al., 2023). Recently, Biabani et al. (2024) gave an improved $(4 + \epsilon)$-approximate solution with $(1+\epsilon)z$ outliers discarded and amortized update time $O(\frac{k^6 \log k \log(n\Delta)}{\epsilon^3})$ assuming prior knowledge on the range for $\Delta$. These results can be extended to static settings, where a $(14 + \epsilon, 1 + \epsilon)$-approximate solution can be obtained in time $O\left((\frac{ndk \log k}{\epsilon} + d(\frac{k \log k}{\epsilon})^2) \cdot \frac{\log \log(n\Delta)}{\epsilon}\right)$ (Chan et al., 2023), or a $(4 + \epsilon, 1 + \epsilon)$-approximate solution can be obtained in time $O(\frac{ndk^3 \log \log(n\Delta)}{\epsilon})$, respectively (Biabani et al., 2024). In metrics with bounded doubling dimension, several fast approximation results were known for the fully-dynamic settings (Biabani et al., 2023; De Berg et al., 2023) with improved update time. However, the query time still depends on $\Delta$.

**Individual Fair $k$-center Problem.**   The notion of individual fairness was first proposed by Jung et al. (Jung et al., 2020) to guarantee that data points should not be assigned to the centers too far from them. Given a clustering instance $(P, k, \tau)$, the goal of individual fair $k$-clustering problem is to minimize the clustering objective function while making sure that each data point should be assigned to a center with distance no more than $\tau$ times the distance to its $\lceil |P|/k \rceil$-closest neighbor in $|P|$. Jung et al. (2020) proposed a greedy algorithm which can guarantee a 2-approximation on fairness without approximation guarantees on the clustering quality, i.e., the algorithm can only guarantee that each data point has a center with distance at most twice the distance to its $\lceil |P|/k \rceil$-closest neighbor. Mahabadi & Vakilian (2020) proposed a local search framework for the individual fair $k$-clustering problems with running time $\tilde{O}(k^5 n^5)$, where a bi-criteria $(O(\log n), 7)$-approximation can be obtained for the $k$-center objective (i.e., with $O(\log n)$-approximation on the clustering quality and 7-approximation on the fairness guarantee). Negahbani & Chakrabarty (2021) gave an improved framework based on linear programming rounding for the individual fair $k$-clustering problems with polynomial running time $\tilde{O}(kn^4)$, where a bi-criteria $(2 + \epsilon, 3)$-approximation can be obtained for the $k$-center objective. Vakilian & Yalciner (2022) reduced the fair constraints to matroid constraints and designed a unified polynomial-time framework for the individual fair clustering problems, where a $(3 + \epsilon, 3)$-approximate solution can be obtained for the $k$-center objective. Recently, Ebbens et al.

(2025) proposed a $(2+\epsilon, 2)$-approximation algorithm using greedy ball coverage and optimal radius guessing strategies with running time $O(n^2 + ndk \log \log(n\Delta)/\epsilon)$. Then, to further accelerate the fair radius estimation process, an algorithm with $(2 + \epsilon, 10)$-approximate solution and running time $O(ndk \log(n/\eta) + k^2 d/\epsilon)$ was also proposed in Ebbens et al. (2025), which is the first algorithm to achieve near-linear running time in the data size, where $\eta$ is a parameter within range $(0, 1)$ for controlling the success probability.

**Proportionally Fair $k$-center Problem.**   The proportionally fair $k$-clustering problem was motivated by the fair allocation of public resources (Chen et al., 2019), where individuals are assumed to be closer to their center in terms of distance. Given a dataset $P$ and a set $X \subseteq P$ of centers with size at most $k$, a set $S \subseteq P$ of at least $\lceil |P|/k \rceil$ data points is called a blocking coalition if $\forall i \in S$, there exists a data point $y \in P$ such that the distance between $y$ and $i$ is smaller than the distance between $i$ to its closest center in $X$. A set $X$ of centers is proportional if there is no blocking coalition against $X$. Chen et al. (2019) gave a greedy algorithm which can find a $(1+\sqrt{2})$-proportional set $X$ in linear time in the data size without approximation guarantees on clustering quality. They also considered proportionality as a constraint when solving the $k$-median problem, where an approximation algorithm was given such that an $O(1)$-proportional clustering can be obtained in polynomial time with constant approximation on the $k$-median objective. Micha & Shah (2020) proved that the greedy algorithm proposed by Chen et al. (2019) can provide a better 2-approximation when $d = L^2$ in infinite metric space $\mathcal{M} = \mathbb{R}^t$. They also showed that when $t$ is a constant, a $2(1 + \epsilon)$-proportional fair clustering for any fixed $\epsilon > 0$ can be achieved using a PTAS for a sub-routine called greedy capture. However, to the best of our knowledge, there is no known approximation result that can guarantee the proportionality and clustering quality for the $k$-center objective simultaneously.

$(\alpha, \beta)$**-Fair $k$-center Problem.**   The $(\alpha, \beta)$-fair clustering problem was introduced to guarantee that sensitive attributes should be properly reflected in each cluster (Chierichetti et al., 2017). The initial work (Chierichetti et al., 2017) focused on the scenario with 2 protected groups. Ahmadian et al. (2019) extended the fairness notion to unlimited number of protected groups with proportional upper bound requirements on each protected group of each cluster, where a 3-approximate solution with 2 additive fairness violation can be obtained in polynomial time using linear programming rounding method. Bera et al. (2019) further strengthened this notion, assuring protection against under-representation of any protected group in each cluster (with both upper bound and lower bound requirements on the proportions for each protected group of each cluster). They proposed a unified framework for $(\alpha, \beta)$-fair clustering in $l_p$ metrics. Given any $\rho$-approximate algorithm for the vanilla $k$-clustering problem, a $(\rho+2)$-approximate solution with $(4\upsilon+3)$-additive violation can be obtained in polynomial time based on LP rounding techniques, where $\upsilon$ is the maximum number of groups that a single data point can belong to (a common setting is $\upsilon = 1$). Harb & Shan (2020) proposed a linear programming method, which does not necessarily depend on the number of points in the input during the linear programming rounding process if the number of clusters $k$ is fixed. With this technique, a 3-approximation can be achieved with 0 fairness violation in expectation. However, the proposed method requires that $k$ is a fixed constant. Moreover, the proposed method also relies on a guess for the optimal clustering radius, which incurs an $O(\log \log(n\Delta))$ loss on the running time.

## A.2   MISSING PROOFS IN MULTI-SCALING

**Lemma 2.** *Algorithm 1 takes time $O(nd \log^2(n)/\lambda^2)$ to construct an integer list $\mathcal{D}$ and a bucket list $\mathcal{B}$ with $|\mathcal{D}| = O(n \log(nd)/\lambda^2)$ and $|\mathcal{B}| = O(n \log(nd)/\lambda^2)$.*

*Proof.* According to Lemma 1, the HST construction takes $O(nd \log^2 n)$ time. In the tree mapping process of Algorithm 1 (steps 2-8 of Algorithm 1), each node in the HST $\mathcal{T}$ is visited once. If $r_L \leq (1 + \lambda)^t < r_H$ holds in step 7 of Algorithm 1, a bucket $b(t)$ is inserted into the bucket list $\mathcal{B}$ (step 8 of Algorithm 1). Since $r_H = s(v_p)/\lambda$ and $r_L \geq s(v_p)/\gamma$, we have $\frac{r_H}{r_L} \leq \frac{\gamma}{\lambda} = O(nd/\lambda)$, where the inequality follows from that $\gamma = \rho \mathcal{P}_{HST}(n, d) + 1$ for some constant $\rho$ and $\mathcal{P}_{HST}(n, d) = nd$ using Lemma 1. Hence, the number of buckets inserted into $\mathcal{B}$ within a single node $v \in \mathcal{T}$ is at most $O(\log(nd)/\lambda^2)$. According to Lemma 1, observe that the number of nodes in an HST $\mathcal{T}$ can be bounded by $O(n)$. Thus, the total number of buckets in $\mathcal{B}$ is at most $O(n \log(nd)/\lambda^2)$, and the total number of integers in $\mathcal{D}$ is at most $O(n \log(nd)/\lambda^2)$. The overall running time can be bounded by $O(nd \log^2 n + n(\log(nd)/\lambda^2))$, which is $O(nd \log^2(n)/\lambda^2)$. $\square$

**Lemma 3.** *Let $P \subset \mathbb{R}^d$ be a given dataset. For each pair of points $q_1, q_2 \in P$, let $l_{\min}$ be an approximate distance lower bound such that $l_{\min} \leq \delta(q_1, q_2) \leq \Psi l_{\min}$ holds for some $\Psi > 1$. $\mathcal{L}$ contains at least one radius $L \in \mathcal{L}$ such that $l_{\min} \leq L \leq (1 + \lambda)l_{\min}$ by setting the input parameter $\rho$ for Multi-Scaling as $\rho \geq 2\Psi$.*

*Proof.* Observe that there must exist an integer $t' \in \mathbb{Z}$ such that $\lambda(1 + \lambda)^{t'} < l_{\min} \leq (1 + \lambda)^{t'+1}\lambda$. Thus, there are two cases that may happen: (1) $t' \in \mathcal{D}$; (2) $t' \notin \mathcal{D}$, where $\mathcal{D}$ is the integer set constructed in step 9 of Algorithm 1. If case (1) happens, there must exist a candidate clustering radius $L = \lambda(1 + \lambda)^{t'+1}$ with $L \in \mathcal{L}$, such that $l_{\min} \leq L < (1 + \lambda)l_{\min}$. Next, we consider a more complicated case when $t' \notin \mathcal{D}$.

Denote the root node of the HST $\mathcal{T}$ as $r$. Let $v(q_1)$ and $v(q_2)$ be the leaf nodes corresponding to data point $q_1$ and $q_2$, respectively. For any leaf node $v(l) \in \mathcal{T}$, according to items (1), (2) and (3) of the properties for the tree, each leaf node $v(l)$ in the tree has size $s(v(l)) = 0$. Thus, we have $s(v(q_1)) = s(v(q_2)) = 0$. Since $\delta(q_1, q_2) \geq l_{\min}$, it also holds that $s(r)/\lambda \geq \frac{l_{\min}}{\lambda} > (1 + \lambda)^{t'}$. Let $\mathcal{P}^\dagger(q_1)$ and $\mathcal{P}^\dagger(q_2)$ denote the paths from the nodes $v(q_1)$ and $v(q_2)$ to the root node $r$, respectively. Let $\mathcal{M}$ denote the set of nodes in $\mathcal{T}$ satisfying the following conditions: (1) each node $v \in \mathcal{M}$ is either a non-root node along the path $\mathcal{P}^\dagger(q_1)$ from $v(q_1)$ to the root $r$, or a non-root node along the path $\mathcal{P}^\dagger(q_2)$ from $v(q_2)$ to the root $r$; (2) for each node $v \in \mathcal{M}$ with its parent node $v_p$, $v$ satisfies that $s(v)/\lambda \leq (1 + \lambda)^{t'}$ and $s(v_p)/\lambda > (1 + \lambda)^{t'}$. It holds trivially that $|\mathcal{M}| > 0$ since $s(v(q_1)) = s(v(q_2)) = 0$ and $s(r) > (1 + \lambda)^{t'}$

For each node $v' \in \mathcal{M}$, denote $r_L(v') = \max\{s(v')/\lambda, s(v'_p)/\gamma\}$ and $r_H(v') = s(v'_p)/\lambda$, where $\gamma$ is the distortion polynomial of the tree with $\gamma = \rho\mathcal{P}_{HST}(n, d) + 1$ for some $\rho \geq 2\Psi$ (step 2 of Algorithm 1). Observe that $r_H(v') > (1 + \lambda)^{t'}$ and $s(v')/\lambda \leq (1 + \lambda)^{t'}$. Since $t' \notin \mathcal{D}$ in case (2), $v'$ is not inserted into the bucket list $\mathcal{B}$ in step 8 of Algorithm 1. Hence, the node $v'$ must satisfy that $s(v'_p)/\gamma > (1 + \lambda)^{t'}$. According to item (5) of the properties for an HST, it holds that $s(v'_p) \leq r_{out}(v') \cdot \mathcal{P}_{HST}(n, d)$. Then, there are two subcases that may happen: (1) $\{q_1, q_2\} \backslash P(v') \neq \emptyset$; (2) $\{q_1, q_2\} \subseteq P(v')$.

If subcase (1) happens, for an arbitrary tree node $v' \in \mathcal{M}$, we can assume that $q_1 \in P(v')$ and $q_2 \notin P(v')$ without loss of generality. Since $\delta(q_1, q_2) \leq \Psi l_{\min}$. Hence, we have $r_{out}(v') \leq \Psi l_{\min}$ and $s(v'_p) \leq \Psi l_{\min} \cdot \mathcal{P}_{HST}(n, d)$, where the inequality follows from item (5) of the properties for an HST that $\frac{s(v'_p)}{r_{out}(v')} \leq \mathcal{P}_{HST}(n, d)$. This implies that $l_{\min} \geq \frac{s(v'_p)}{\Psi\mathcal{P}_{HST}(n, d)} > \frac{\gamma(1+\lambda)^{t'}}{\Psi\mathcal{P}_{HST}(n, d)}$, where the last inequality follows from $s(v'_p) > \gamma(1 + \lambda)^{t'}$. Since $\gamma > \rho\mathcal{P}_{HST}(n, d) \geq 2\Psi\mathcal{P}_{HST}(n, d)$ and $0 < \lambda \leq 1$, we can get that $l_{\min} > 2(1 + \lambda)^{t'} \geq \lambda(1 + \lambda)^{t'+1}$, which contradicts with the assumption in case (2) that $l_{\min} \leq \lambda(1 + \lambda)^{t'+1}$. Hence, subcase (1) can never happen.

Then we consider that subcase (2) happens. Observe that we have $\delta(q_1, q_2) \geq l_{\min}$. In this subcase, since $\{q_1, q_2\} \subseteq P(v')$, we have $Dia(P(v')) \geq l_{\min}$ and $s(v') \geq Dia(P(v')) \geq l_{\min} > \lambda(1+\lambda)^{t'}$. This contradicts with the definition of $\mathcal{M}$ that each node $v \in \mathcal{M}$ should satisfy $s(v)/\lambda \leq (1 + \lambda)^{t'}$. This implies that subcase (2) will not happen.

Putting all these together, we can get that both subcase (1) and subcase (2) will not happen during the tree mapping process. This implies that the integer $t'$ with $\lambda(1 + \lambda)^{t'} < l_{\min} \leq (1 + \lambda)^{t'+1}\lambda$ must belong to the integer list $\mathcal{D}$ constructed in step 9 of Algorithm 1. Hence, the properties stated in Lemma 3 hold for the radii set $\mathcal{L}$ constructed using Algorithm 1. □

### A.3 Missing Proofs for $k$-center with Outliers

**Lemma 5.** *If $L_1 \geq \frac{L^*}{3}$, there exists at least one radius $L_f \in \mathcal{L}_1$ such that $L^* \leq L_f \leq (1 + \lambda)L^*$.*

*Proof.* If $L_1 \geq \frac{L^*}{3}$, $L^*$ lies in the interval $[\frac{L_1}{2}, 3L_1]$. Without loss of generality, we can assume that $(1 + \lambda)^j \leq L^* < (1 + \lambda)^{j+1}$ for some integer $j \in \mathbb{Z}$. Let $L_f = \min\{3L_1, (1 + \lambda)^{j+1}\}$. It holds trivially that $L_f \in \mathcal{L}_1$. Then, we have $L_f \leq (1 + \lambda)^{j+1} = (1 + \lambda)^j(1 + \lambda) \leq (1 + \lambda)L^*$. Next, we

consider the lower bound for $L_f$. Since $3L_1 \geq L^*$ and $(1+\lambda)^{j+1} > L^*$, we have $L_f \geq L^*$. Then, Lemma 4 can be concluded. $\qquad\square$

**Lemma 6.** *If $L_1 < \frac{L^*}{3}$, there exists at least one radius $L'' \in \mathcal{L}_2$ such that $\frac{L^*}{3} \leq L'' \leq (1+\lambda)\frac{L^*}{3}$.*

*Proof.* Let $l_{\min} = \frac{L^*}{3}$. Since $\frac{L^*}{3} \leq \delta(s_{p_{\max}}, s_{q_{\max}}) \leq 6L^*$, it holds that $l_{\min} \leq \delta(s_{p_{\max}}, s_{q_{\max}}) \leq 18l_{\min}$. According to Lemma 3, by setting the input parameter $\rho$ for Algorithm 1 as $\rho \geq 36$, we can get that the constructed radii set $\mathcal{L}_2$ in step 5 of Algorithm 2 contains at least one radius $L'' \in \mathcal{L}_2$ with $\frac{L^*}{3} \leq L'' \leq (1+\lambda)\frac{L^*}{3}$ $\qquad\square$

**Lemma 7.** *Given a parameter $0 < \lambda < 1$, for a $k$-center with outliers instance $(P, k, z, d)$ where $|P| = n$, a radii set $\mathcal{L}$ with size $O((k+z)\log((k+z)d)/\lambda^2)$ can be constructed in time $O((k+z)(nd+d\log^2(k+z)/\lambda^2))$ such that there exists one radius $L \in \mathcal{L}$ satisfying $L^* \leq L \leq (1+\lambda)L^*$.*

*Proof.* According to Lemma 5 and Lemma 6, it holds trivially that the set $\mathcal{L}$ returned by the MS-DR algorithm (Algorithm 2) contains at least one radius $L \in \mathcal{L}$ with $L^* \leq L \leq (1+\lambda)L^*$. Observe that $|\mathcal{L}| \leq |\mathcal{L}_1| + |\mathcal{L}_3|$, where $\mathcal{L}_1$ and $\mathcal{L}_3$ are the sets of radii obtained in step 4 and 6 of Algorithm 2, respectively. We can get that $|\mathcal{L}_1| = O(\frac{1}{\lambda})$ and $|\mathcal{L}_2| = (k+z)\log((k+z)d)/\lambda^2$ according to Lemma 2. Hence, $|\mathcal{L}| \leq (k+z)\log((k+z)d)/\lambda^2$. As for the running time, the greedy selection process (steps 2 of Algorithm 2) takes time $O(nd(k+z))$. For the radius construction process (steps 3-7 of Algorithm 2), the tree construction and tree mapping processes take time $O((k+z)d\log^2(k+z))$ and $O((k+z)\log((k+z)d)/\lambda^2)$ according to Lemma 1 and Lemma 2, respectively. Hence, the running time of the MS-DR algorithm should be $O((k+z)(nd + d\log^2(k+z) + \log(d(k+z))/\lambda^2)) = O((k+z)(nd + d\log^2(k+z)/\lambda^2))$. $\qquad\square$

The algorithm for solving the $k$-center with outliers problem is outlined in Algorithm 3 (the FKOC algorithm). The proposed algorithm applies sampling-based strategy to compress the data if the number of outliers $z$ is large. Specifically, if $z \geq k^2 \log n/\lambda^2$, FKOC takes a random sample $S$ with size $O(\frac{nk\log n}{z\lambda^2})$ to compress the data size for accelerations (Lemma 8). Then, summary-based multi-scaling is applied to obtain a set of candidate radii set on the data (step 5 of Algorithm 3). Once the radii set is constructed, any bi-criteria $(\zeta, \zeta')$-approximation algorithm $\mathcal{F}$ (such as the $(4, 1+\epsilon)$-approximation algorithm proposed by Biabani et al. (2024)) can be used to obtain the clustering solutions with a binary searching strategy.

**Theorem 1.** *For $k$-center with outliers, let $\mathcal{F}$ be any radius-guessing bi-criteria algorithm with running time $\mathcal{F}(n, d, k)$ for a fixed radius, where the approximation ratio for $\mathcal{F}$ is $(\mathcal{F}(r_1), \mathcal{F}(r_2))$. Then, there exists an algorithm that returns a $(2\mathcal{F}(r_1) + \epsilon, \mathcal{F}(r_2))$-approximate solution in time $\tilde{O}(nd/\epsilon^2) + O(\mathcal{F}(n, d, k) \cdot \frac{\log(kd\log n)}{\epsilon})$.*

*Proof.* We first consider the case where $z \geq k^2 \log n/\lambda^2$. In this case, according to Lemma 8, an instance $S$ with size $O(\frac{n}{k})$ can be constructed with $z' = O(\frac{k\log n}{\lambda})$. Thus, according to Lemma 7, a radii set $\mathcal{L}$ with size $O(\frac{kd\log n}{\lambda^3}\log(\frac{kd\log n}{\lambda}))$ can be constructed in time $O(\frac{ndk\log n}{\lambda^2} + \frac{k^2 d\log n\log^2(kd\log n)}{\lambda^5})$. Given any radius-guessing based bi-criteria algorithm $\mathcal{F}$ with running time $\mathcal{F}(n, d, k)$ for a fixed radius and approximation ratio $(\mathcal{F}(r_1), \mathcal{F}(r_2))$, a set $\mathcal{C}_m$ of centers with size $k$ and clustering radius at most $L_m$ can be obtained by discarding at most $\mathcal{F}(r_2)z$ outliers during the binary search process (steps 9-17 of Algorithm 3). Then, we can obtain a $(2\mathcal{F}(r_1) + \lambda)$-approximate solution with $\mathcal{F}(r_2)z$ outliers discarded (the 2 factor loss is due to approximation loss stated in Lemma 8). The running time can be bounded by $O(\mathcal{F}(|S|, d, k)\log|\mathcal{L}|)$. Since $|S| = O(\frac{n}{k}) \leq O(n)$, the binary search process takes time $O(\mathcal{F}(n, d, k) \cdot \frac{\log(kd\log n)}{\lambda})$. Hence, the total running time is $O(\frac{ndk\log n}{\lambda^2} + \frac{k^2 d\log n\log^2(kd\log n)}{\lambda^5} + \mathcal{F}(n, d, k) \cdot \frac{\log(kd\log n)}{\lambda})$.

On the other hand, if $z \leq k^2 \log n/\lambda^2$, a radii set $\mathcal{L}$ with size $O(\frac{kd\log n}{\lambda^3}\log(\frac{kd\log n}{\lambda}))$ can be constructed in time $O(\frac{ndk^2\log n}{\lambda^2} + \frac{k^2 d\log n\log^2(kd\log n)}{\lambda^5})$. During the binary search process (steps 9-17 of Algorithm 3), a set $\mathcal{C}_m$ of centers with size $k$ and clustering radius at most $(1+\lambda)\mathcal{F}(r_1)$ can

---

**Algorithm 3** FKOC$(P, k, d, z, \lambda, \mathcal{F})$

---

**Input:** A $k$-center with outliers instance $(P, k, d, z)$, parameter $0 < \lambda \leq 1/6$, an $\mathcal{F}(r_1)$-approximation $k$-center with outliers algorithm $\mathcal{F}$ with $\mathcal{F}(r_2)z$ outliers discarded (ideally $(1 + \epsilon)z$).
**Output:** A set $C \subset \mathbb{R}^d$ of clustering centers.

1: **if** $z \geq k^2 \log n / \lambda^2$ **then**
2:     $\epsilon_1 = \frac{\lambda}{12}$.
3:     Let $S$ be a set of points with size $O(\frac{nk \log n}{\epsilon_1 z})$ sampled uniformly and independently from $P$.
4:     $P \leftarrow S, z \leftarrow (1 + \epsilon_1)\frac{z|S|}{n}$.
5: $\mathcal{L} =$ MS-DR$(P, k, d, z, \lambda)$.
6: Sort the radius in $\mathcal{L}$ with non-decreasing order.
7: Let $L_i$ denote the $i$-th clustering radius in $\mathcal{L}$, and initialize $u_{id} = |\mathcal{L}|, l_{id} = 1, L_{\min} = +\infty$, $C = \emptyset$.
8: **while** $l_{id} \leq u_{id}$ **do**
9:     $m = \lfloor (l_{id} + u_{id})/2 \rfloor$.
10:     $\mathcal{C}_m, \mathcal{Z}_m = \mathcal{F}(P, k, d, z, \lambda, L_m)$, where $\mathcal{C}_m$ and $\mathcal{Z}_m$ are the sets of centers and outliers returned by algorithm $\mathcal{F}$, respectively.
11:     **if** $|\mathcal{Z}_m| \leq \mathcal{F}(r_2)z$ **then**
12:        **if** $L_m < L_{\min}$ **then**
13:           $L_{\min} = \min\{L_m, L_{\min}\}$.
14:           $C = \mathcal{C}_m$.
15:        $u_{id} = m - 1$.
16:     **else**
17:        $l_{id} = m + 1$.
18: **return** $C$.

---

be obtained by discarding at most $\mathcal{F}(r_2)z$ outliers, where the running time for binary search can be bounded by $O(\mathcal{F}(n, d, k) \cdot \log |\mathcal{L}|)$. Hence, the total running time for the case if $z \leq k^2 \log n / \lambda^2$ is $O(\frac{ndk^2 \log n}{\lambda^2} + \frac{k^2 d \log n \log^2(kd \log n)}{\lambda^5} + \mathcal{F}(n, d, k) \cdot \frac{\log(kd \log(n))}{\lambda})$.

Putting all these together, since $\mathcal{F}(n, d, k)$ usually dominates $\frac{ndk^2 \log n}{\lambda^2}$ and $\frac{k^2 d \log n \log^2(kd \log n)}{\lambda^5}$, the running time for Algorithm 3 can be bounded by $\tilde{O}(nd/\lambda^2) + O(\mathcal{F}(n, d, k) \cdot \frac{\log(kd \log(n))}{\lambda})$, where the approximation ratio is $(2\mathcal{F}(r_1) + \lambda, \mathcal{F}(r_2))$. $\qquad\square$

### A.4 ALGORITHMS FOR INDIVIDUAL FAIR $k$-CENTER PROBLEM

For the individual fair $k$-center problem, we first show that there exists at least one pair of points $p', q' \in P$ such that $\delta(p', q')$ serves as a good estimate for the optimal clustering radius $L^*$. Given an individual fair clustering instance $(P, k, d, \tau)$, let $P^*_{\max} \in \mathcal{H}(P)$ be the optimal cluster with the largest radius. Let $p_{\max} = \arg\max_{p \in P} \delta(p, c^*_{\max}) = L^*$ be the point in $P^*_{\max}$ with the maximum distance to the optimal clustering center $c^*_{\max}$. Then, there are two cases that may happen: (1) $L^*/2 \leq \tau r_p$ holds for each $p \in P^*_{\max}$; (2) $\exists p \in P^*_{\max}$, s.t. $L^*/2 > \tau r_p$. For case (1), there are two subcases that may occur: (1) there exists a pair of points $p', q' \in P^*_{\max}$ such that $\delta(p', q') \geq L^*/2$; or (2) all pairs $p, q \in P^*_{\max}$ satisfy $\delta(p, q) < L^*/2$. In subcase (1), the triangle inequality gives $\delta(p', q') \leq 2L^*$, where setting $l_{\min} = L^*/2$ yields $l_{\min} \leq \delta(p', q') \leq 4l_{\min}$. In subcase (2), we construct a new cluster $P'$ by replacing $P^*_{\max}$ with a ball centered at any $p' \in P^*_{\max}$. For each data point $p \in P'$, we have $\delta(p, p') \leq L^*/2 < L^*$ and $\delta(p, p') \leq L^*/2 \leq \tau r_p$ according to triangle inequality and the case condition. It can be seen that the constructed new cluster is a feasible solution with clustering radius strictly smaller than $L^*$, which contradicts with the optimality for $P^*_{\max}$. This contradiction implies that subcase (1) must hold. Therefore, there exists a pair of points $(p', q') \in P$ such that $\delta(p', q')$ lies within range $[l_{\min}, 4l_{\min}]$ for some $l_{\min} = L^*/2$. Using Lemma 3 and setting the multi-scaling parameter $\rho$ as $\rho = 8$, Algorithm 1 guarantees that the radii set $\mathcal{L}$ contains a value $L$ such that $L^* \leq L \leq (1 + \lambda)L^*$.

Next we consider that case (2) happens. In this case, we have that $\tau r_p < L^*/2$ holds for some $p \in P^*_{\max}$ and $\tau \geq 1$. Let $p_{\max} \in P^*_{\max}$ be the point with the maximum distance to $c^*_{\max}$, i.e., $p_{\max} = \arg\max_{p \in P^*_{\max}} \delta(p, c^*_{\max})$. Observe that $p \neq p_{\max}$, as otherwise we have

$\delta(p_{\max}, c_{\max}^*) \leq \tau r_{p_{\max}} < L^*/2$, which contradicts with the fact that $\delta(p_{\max}, c_{\max}^*) = L^*$. Then, we have $\delta(p_{\max}, p) \geq L^*/2$ and $\delta(p_{\max}, p) \leq 2L^*$ using triangle inequality. Therefore, there still exists a pair $(p_{\max}, p) \in P$ such that $\delta(p_{\max}, p)$ lies within range $[l_{\min}, 4l_{\min}]$ for some $l_{\min} = L^*/2$. Using Lemma 3 and setting the multi-scaling parameter $\rho$ as $\rho = 8$, Algorithm 1 guarantees that the radii set $\mathcal{L}$ contains a value $L$ such that $L^* \leq L \leq (1 + \lambda)L^*$.

Based on the above observation, Corollary 2 shows that multi-scaling can be directly combined with any radius-guessing based bi-criteria fair clustering algorithm $\mathcal{F}$. Suppose $\mathcal{F}$ achieves an approximation of $(\mathcal{F}(r_1), \mathcal{F}(r_2))$. Then, the combination yields a $(\mathcal{F}(r_1) + \epsilon, \mathcal{F}(r_2))$-approximation. The overall running time is $O(nd \log^2 n/\epsilon^2 + \mathcal{F}(n, d, k) \cdot \frac{\log(n \log d)}{\epsilon})$, where $\mathcal{F}(n, d, k)$ denotes the running time of $\mathcal{F}$ for a fixed radius. Next, we present a faster algorithm for individual fair $k$-center clustering using data reduction. The proposed algorithm is described in Algorithm 4 (denoted as the SIFC algorithm: Sampling-Based Individual Fair Clustering), where the general idea is to use sampling-based strategies for fast fair radii estimation and data reduction, while ensuring the fairness violations.

---

**Algorithm 4** SIFC$(P, k, \tau, \lambda, \eta)$

**Input:** An individual fair $k$-center instance $(P, k, \tau)$, a parameter $0 < \lambda \leq 1$, and a parameter $0 < \eta < 1$.
**Output:** A set $\mathcal{C} \subset \mathbb{R}^d$ of clustering centers.
1: Initialize $C' = \emptyset$ and $\mathcal{C} = \emptyset$.
2: Add an arbitrary data point $p \in P$ to $C'$.
3: **for** $i = 1$ to $k - 1$ **do**
4:     Let $p = \arg\max_{q \in P} \delta(q, C')$, and add $p$ to $C'$.
5: $\mathcal{L} = \text{MS-DR}(P, k, d, 0, \lambda)$.
6: Randomly and independently sampling a set $S \subset P$ with size $O(k \log \frac{n}{\eta})$ from $P$, and set
    $\mathcal{U} = C' \cup S$.
7: $\tilde{r} = \text{FairSampling}(P, \mathcal{U}, k, \tau, \eta)$.
8: Sort the radii in $\mathcal{L}$ with non-decreasing order.
9: Let $L_i$ be the $i$-th clustering radius in $\mathcal{L}$, and initialize $u_{id} = |\mathcal{L}|, l_{id} = 1, f(\mathcal{L}) = +\infty$.
10: **while** $l_{id} \leq u_{id}$ **do**
11:     $m = \lfloor (l_{id} + u_{id})/2 \rfloor$.
12:     Initialize $\mathcal{U}_f = \mathcal{U}, C_f = \emptyset$.
13:     **for** $i = 1$ to $k$ **do**
14:         Let $m_i = \arg\min_{u \in \mathcal{U}_f} \tilde{r}(u)$, and add $m_i$ to $C_f$.
15:         Delete each data point $q \in \mathcal{U}_f$ if $\delta(q, m_i) \leq 2L_m$ and $\delta(q, m_i) \leq 2\tau\tilde{r}(q)$.
16:     **if** $|\mathcal{U}_f| = 0$ **then**
17:         **if** $L_m < f(\mathcal{L})$ **then**
18:             $f(\mathcal{L}) = L_m, \mathcal{C} = C_f$.
19:         $u_{id} = m - 1$.
20:     **else**
21:         $l_{id} = m + 1$.
22: **return** $\mathcal{C}$.

---

In steps 1-5 of SIFC algorithm, sampling and multi-scaling techniques are used to obtain clustering radii estimation for the given individual fair $k$-center instance. Then, in step 6 of SIFC algorithm, a set $S$ of samples with size $O(k \log(n/\eta))$ is taken randomly and independently from $P$. The objective here is to ensure that for each data point $p \in P$, there exists a point $s_p \in S$ such that $p$ can be assigned to $s_p$ while satisfying the fairness constraint. Since the set $C'$ constructed in steps 1-4 of Algorithm 4 using furthest-first selection rule guarantees a 2-approximation on clustering quality, a summary $\mathcal{U}$ can be constructed with $\mathcal{U} = C' \cup S$. The following lemma shows that, the summary $\mathcal{U}$ constructed in step 6 of Algorithm 4 is a $(2, 2)$-approximate solution for $P$ with size at most $O(k \log n)$. Moreover, for each data point $p \in P$, there exists at least one point $q \in \mathcal{U}$ with fair radius smaller than $2r_p$, and the distance between $p$ and $q$ is at most $2L^*$.

**Lemma 9.** *With probability at least $1 - \eta$, for each data point $p \in P$, there exists at least a data point $q \in \mathcal{U}$ with fair radius smaller than $2r_p$, and the distance between $p$ and $q$ is at most $2L^*$.*

---

**Algorithm 5** FairSampling$(P, P', k, \tau, \eta)$

---

**Input:** An individual fair $k$-center instance $(P, k, \tau)$, a target dataset $P' \subseteq P$, and a parameter $0 < \eta < 1$.
**Output:** A mapping function $\tilde{r}$.

1: Initialize a mapping $\tilde{r} \to \mathbb{R}$, where $\tilde{r}(p) = -1$ for each $p \in P'$.
2: $s = 36k\lceil \ln(2|P|/\eta) \rceil$.
3: $t = 27\lceil \ln(2|P|/\eta) \rceil$.
4: Let $S$ be a subset of size $s$ drawn uniformly at random with replacement form $P$.
5: **for** $p \in P'$ **do**
6:     Let $x$ be the $t$-nearest neighbor of $p$ in $S$.
7:     $\tilde{r}(p) = \delta(p, x)$.
8: Set $\mathcal{I} \leftarrow P'$.
9: Initialize $C = \{p'\}$, where $p' = \arg\min_{h \in \mathcal{I}} \tilde{r}(h)$.
10: Compute $r_{p'}$ exactly and set $\tilde{r}(p') = r_{p'}$.
11: **while** $\mathcal{I} \neq \emptyset$ **do**
12:     $p = \arg\min_{h \in \mathcal{I}} \tilde{r}(h)$.
13:     **if** $\exists q \in C$ s.t. $\tilde{r}(q) + \tilde{r}(p) \geq \delta(p, q)$ **then**
14:       $\tilde{r}(p) = \delta(p, q) + \tilde{r}(q)$.
15:     **else**
16:       Compute $r_p$ exactly and set $\tilde{r}(p) = r_p$.
17:       $C = C \cup \{p\}$
18:     **if** $|C| > 3k$ **then**
19:       **return** "FAIL".
20: **return** $\tilde{r}$.

---

*Proof.* Recall that $\mathcal{U} = C' \cup S$. Consider an arbitrary data point $p \in P$, denote $\mathcal{N}(p)$ as the set of the nearest $\lceil n/k \rceil$ data points in $P$ to $p$. Let $\zeta = \frac{|\mathcal{N}(p)|}{|P|}$. Then, it holds trivially that $\zeta \geq \frac{1}{k}$. Our goal is to guarantee that at least one data point can be sampled from $\mathcal{N}(p)$ to construct a set $S$ for each $p \in P$. To guarantee that the probability of sampling at least one data point from $\mathcal{N}(p)$ is lower bounded by $1 - \eta$, we only need to guarantee that $1 - (1 - \frac{|\mathcal{N}(p)|}{|P|})^{|S|} \geq 1 - \eta$, which implies that $S$ should have size at least $\frac{\log \frac{1}{\eta}}{\log \frac{1}{1-\zeta}}$. Since $\zeta \geq \frac{1}{k}$, if $|S| \geq k \log \frac{1}{\eta}$, the success probability can be lower bounded by $1 - \eta$. By taking a union bound over the success probability for all data points in $P$ and replacing $\eta$ with $\frac{\eta}{n}$, a set $S$ with size $O(k \log \frac{n}{\eta})$ taken randomly and independently from $P$ can guarantee that there exists at least one point $\Phi(p) \in S$ with $\Phi(p) \in \mathcal{N}(p)$ for each $p \in P$.

For each data point $p \in P$, let $s_p$ be its nearest data point in $\mathcal{U}$. Since $C'$ is a 2-approximate solution on clustering quality for $P$ (Gonzalez, 1985), we have $\delta(p, s_p) \leq 2L^*$. Next, we show the relationships between $r_p$ and $r_{s_p}$. We first argue that for any pair of points $p, q \in P$, it holds that $r_p \leq \delta(p, q) + r_q$. Given any pair of points $p, q \in P$, there are two cases that may happen: (1) $\mathcal{N}(q) = \mathcal{N}(p)$; (2) $\mathcal{N}(q) \neq \mathcal{N}(p)$. If case (1) happens, let $a_p = \arg\max_{h \in \mathcal{N}(p)} \delta(h, p)$. Then, according to triangle inequality, we have $r_p = \delta(p, a_p) \leq \delta(p, q) + \delta(q, a_p) \leq \delta(p, q) + r_q$. If case (2) happens, let $q'$ be an arbitrary data point with $q' \in \mathcal{N}(q)$ and $q' \notin \mathcal{N}(p)$. According to the triangle inequality, we have $\delta(q', p) \leq \delta(q', q) + \delta(q, p) \leq r_q + \delta(p, q)$. If $r_p > r_q + \delta(p, q)$, it holds that $q' \in \mathcal{N}(p)$, which contradicts with the case assumption that $q' \notin \mathcal{N}(p)$. Therefore, we can get that $r_p \leq \delta(p, q) + r_q$ holds for each pair of points $p, q \in P$. Since $s_p$ is the nearest data point in $\mathcal{U}$ to $p$, we have $\delta(p, s_p) \leq \delta(p, \Phi(p)) \leq r_p$. Then, it holds that $r_{s_p} \leq \delta(s_p, p) + r_p \leq 2r_p$, which proves the lemma. □

Based on Lemma 9, we can get that with probability at least $1 - \eta$, the summary $\mathcal{U}$ constructed in step 6 of Algorithm 4 is a $(2, 2)$-approximate solution for $P$ with size at most $O(k \log(n/\eta))$. Moreover, for each point $p \in P$, there exists at least a point $s_p \in \mathcal{U}$ satisfying $r_{s_p} \leq 2r_p$ and $\delta(p, s_p) \leq 2L^*$.

**Corollary 4.** *$\mathcal{U}$ is a $(2, 2)$-approximate solution for $P$. Moreover, for each point $p \in P$, there exists at least a point $s_p \in \mathcal{U}$ satisfying $r_{s_p} \leq 2r_p$.*

In step 7 of SIFC algorithm, a fair sampling strategy is used to obtain estimations for the fair radii based on the ideas proposed in previous work (Ebbens et al., 2025). The fair sampling process guarantees that, with probability at least $1 - \eta$, a radius that closely approximates the fair radius of each data point can be obtained.

**Lemma 10.** (Ebbens et al., 2025) *Given an individual fair $k$-center instance $(P, k, d, \tau)$ and a target set $P' \subseteq P$ of data points, with probability at least $1 - \eta$, Algorithm 5 returns in time $O(|P'|dk \log(n/\eta) + ndk)$ for all $p \in P'$ a value $\tilde{r}(p)$ such that $r_p \leq \tilde{r}(p) \leq 5r_p$.*

In Algorithm 4, to obtain the final clustering solution, it tries to perform a binary search on the radii set $\mathcal{L}$ obtained through multi-scaling. However, in each radius searching process, the greedy coverage process (steps 13-15 of Algorithm 4) is executed only on the summary $\mathcal{U}$ constructed instead of the whole dataset $P$. The following lemma shows that, the greedy coverage process guarantees $(4(1+\lambda), 22)$-approximation for the individual fair $k$-center problem if the given estimation $L_m$ for the optimal clustering radius satisfies $L^* \leq L_m \leq (1 + \lambda)L^*$.

**Lemma 11.** *Let $\mathcal{C}$ be the set of the centers returned by Algorithm 4. $C'$ is a $(4(1 + \lambda), 22)$-approximate solution to $P$.*

*Proof.* We first show that given an estimation $L$ for the optimal clustering radius $L^*$ with $L \geq L^*$, steps 13-15 of Algorithm 4 can cover all the data points in $\mathcal{U}$, where $\mathcal{U}$ is the summary constructed in step 6 of Algorithm 4. Observe that in the $i$-th iteration of the for loop in step 13 of Algorithm 4, a data point $m_i \in \mathcal{U}_f$ with minimum fair radius estimation is picked as the clustering center, where $\mathcal{U}_f$ is the set of the uncovered data points in $\mathcal{U}$ during the greedy coverage process. Let $\mathcal{U}_f^i$ be the set of the uncovered data points in $\mathcal{U}_f$ before executing step 13 of Algorithm 4 in the $i$-th iteration. It holds trivially that $m_i$ must belong to some optimal cluster $P_j^* \in \mathcal{H}(P)$. For each data point $q \in P_j^* \cap \mathcal{U}$, we have $\delta(q, m_i) \leq \delta(q, c_j^*) + \delta(c_j^*, m_i) \leq 2L^* \leq 2L$ using the triangle inequality. Moreover, we can also get that $\delta(q, m_i) \leq \delta(q, c_j^*) + \delta(c_j^*, m_i) \leq \tau r_q + \tau r_{m_i} \leq \tau\tilde{r}(q) + \tau\tilde{r}(m_i) \leq 2\tau\tilde{r}(q)$, where the third inequality follows from Lemma 10, and the last inequality follows from the fact that $m_i$ is the data point in $\mathcal{U}_f^i$ with minimum fair radius estimation. Hence, in each iteration of steps 14-15 of Algorithm 4, data points in an optimal cluster $P_j^* \cap \mathcal{U}$ where $m_i \in P_j^*$ can be covered by $m_i$ with radius $2L$. Thus, by repeating the greedy selection and coverage process for $k$ rounds, all the data points in $\mathcal{U}$ can be covered.

Let $C_L$ be the set of centers obtained after executing $k$ iterations of greedy selection and coverage process using radius $L$, where $L \geq L^*$. For each $p \in \mathcal{U}$, Denote $L(p)$ as the center in $C_L$ that $p$ is assigned to. We can get that $\delta(p, L(p)) \leq 2L$ holds for each $p \in \mathcal{U}$. For a data point $p \in P$, we use $u_p$ to denote the nearest data point in $\mathcal{U}$ to $p$. According to Corollary 4, $\mathcal{U}$ guarantees a 2-approximation on clustering quality. Then, by triangle inequality, we have $\delta(p, L(u_p)) \leq \delta(p, u_p) + \delta(u_p, L(u_p)) \leq 2L^* + 2L \leq 4L$.

On the other hand, for each $p \in P$, the data point $u_p \in \mathcal{U}$ satisfies $r_{u_p} \leq 2r_p$ according to Corollary 4. By triangle inequality, we have $\delta(p, L(u_p)) \leq \delta(p, u_p) + \delta(u_p, L(u_p)) \leq 2\tau r_p + 2\tau\tilde{r}(u_p) \leq 2\tau r_p + 10\tau r_{u_p} \leq 2\tau r_p + 20\tau r_p \leq 22\tau r_p$, where the last inequality follows from the fact that $\tilde{r}(p) \leq 5r_p$ holds for each $p \in P$ using Lemma 10. Using similar ideas from Lemma 7, it also guarantees that there always exists a radius $L \in \mathcal{L}$ such that $L^* \leq L < (1 + \lambda)L^*$ such that Lemma 10 can be proved. $\square$

Putting all these together, we can obtain the following result for the individual fair $k$-center problem.

**Theorem 2.** *For the individual fair $k$-center problem, there exists an algorithm that can output a $(4(1 + \lambda), 22)$-approximate solution in near-linear running time in the data size.*

*Proof.* Observe that the theoretical guarantees for multi-scaling and the greedy selection process are all deterministic. By performing a binary search on the radii set $\mathcal{L}$, we can obtain a $(4(1+\lambda), 22)$-approximate solution with probability at least $(1 - \eta)^2$ using Lemma 11. As for the running time, constructing a summary with size $O(k \log(n/\eta))$ takes time $O(ndk)$. Then, obtaining estimations for fair clustering radii takes time $O(dk^2 \log^2(n/\eta) + ndk)$. Constructing the radii set $\mathcal{L}$ with size

$O(k \log(kd)/\lambda^2)$ takes time $O(kd \log^2 k + k \log(kd)/\lambda^2)$ using Lemma 2 and Lemma 3. Finally, since the whole process is executed for $\log |\mathcal{L}| = O(\frac{k \log(kd)}{\lambda})$ times using binary search, the total running time for Algorithm 4 is $O(ndk + dk^2 \log^2(n/\eta)/\lambda)$. □

## A.5 Algorithms for Proportionally Fair $k$-center Problem

For the proportionally fair $k$-center problem, the notion of proportionally fairness is highly related to the notion of "individual fairness". Any approximation algorithm that guarantees $\gamma$-approximation on individual fairness can be adapted to solve the proportionally fair clustering problem with $(1+\gamma)$-approximation on proportionality. Thus, our proposed multi-scaling and Algorithm 4 can also be used to solve the proportionally fair $k$-center problem.

The following lemma shows that, any individual fair $k$-center algorithm with $O(1)$-approximation on fairness can also guarantee an $O(1)$-approximation on proportionality. Hence, Algorithm 4 can be adapted to solve the proportionally fair $k$-center problem to guarantee the fairness of the given instance, where a constant approximation for proportionality can also be maintained. Putting all these together, we can get the following result for proportionally fair $k$-center problem.

**Lemma 12.** (Chen et al., 2019) *Let $C$ be a set of centers, and let $\gamma \geq 1$. If $\forall j \in P$, there exists a data point $x \in C$ such that $\delta(j,x) \leq \gamma R_j$ where $R_j$ is the distance of the $\lceil n/k \rceil$-nearest point in $P$ to $j$, then $C$ is $(1+\gamma)$-proportional. If $C$ is $\gamma$-proportional, then for each data point $j \in P$, there exists a data point $x \in C$ such that $\delta(j,x) \leq (1+\gamma)R_j$.*

**Theorem 3.** *For proportionally fair $k$-center problem, let $\mathcal{F}$ be any radius-guessing bi-criteria algorithm with running time $\mathcal{F}(n,d,k)$ for a fixed radius and approximation ratio $(\mathcal{F}(r_1), \mathcal{F}(r_2))$. Then, there exists an algorithm that returns a $(\mathcal{F}(r_1) + \epsilon, \mathcal{F}(r_2))$-approximate solution in time $\tilde{O}(nd/\epsilon^2) + O(\mathcal{F}(n,d,k) \cdot \frac{\log(kd \log n)}{\epsilon})$.*

## A.6 Algorithms for $(\alpha, \beta)$-Fair $k$-center Problem

For $(\alpha, \beta)$-fair $k$-center problem, similar to the ideas in Corollary 2, our multi-scaling method can be combined with any existing algorithms to obtain approximation results with running time independent of $\Delta$ while preserving the guarantees.

Next, we show how to design an algorithm with much faster running time. For $(\alpha, \beta)$-fair $k$-center problem, existing algorithms usually rely on LP rounding techniques for fairness adjustment. However, the running time for rounding methods have polynomial dependence in the number of variables, which is the key obstacle for obtaining fast approximation schemes. Partly inspired by the idea in previous work (Bera et al., 2022), we propose a summary construction method for the $(\alpha, \beta)$-fair $k$-center problem using greedy ball coverage strategies. Given an estimation $L$ of the optimal radius, the summary can be constructed by a simple furthest-first strategy.

The algorithm for solving the $(\alpha, \beta)$-fair $k$-center problem is given in Algorithm 6 (GFK: Greedy Fair $k$-center). There are mainly three phases within the algorithm. In the first phase (steps 6-8 of Algorithm 6), a standard Gonzalez's algorithm (Gonzalez, 1985) is used to find a set $C$ of $k$ representative data points that are close to the optimal clustering centers for data reduction. Then, each data point $p \in P$ is assigned to its closest center in $C$. Based on the data representations obtained, in the second phase (steps 9-13 of Algorithm 6), each data point $c_i \in C$ is duplicated for $\Gamma$ data points $\{c_i^1, c_i^2, ..., c_i^\Gamma\}$ co-located at $c_i$, where each $c_i^j$ is assigned with a different color $j \in [\Gamma]$. Moreover, a weight $w$ is assigned to $c_i^j$ such that $w$ is the total number of data points in $P$ with color $j$ that are assigned to $c_i$. Then, a new set $U$ of data representations is obtained. Finally, in the third phase (steps 13-23 of Algorithm 6), a weighted algorithm (Algorithm 7) is used to solve the weighted fair assignment task on $U$ to satisfy the fairness constraints.

In the following, we first define the linear programming relaxation for weighted fair assignment problem. Let $(U, C, k)$ be a weighted fair $k$-center instance, where $U$ is a set of weighted data points and $C$ is a set of centers with size $k$. In the set of the weighted data points, each point $u \in U$ is assigned with a weight $w_u$. The formal algorithm for solving the weighted fair $k$-center assignment problem is described in Algorithm 7. To establish the linear programming formulation for weighted

fair assignments, we first construct a bipartite graph $G(U, C)$, where there is an edge $e(u, v)$ for each $u \in U$ and $c \in C$ (step 2 of Algorithm 7). Then, given a radius estimation $L$, a pre-processing step is used to restrict the assignments from data point in $U$ to the centers in $C$. More specific, we require that a data point $u \in U$ can be assigned to a center $c \in C$ if $\delta(u, c) \leq L$. If the distance between a data point $u \in U$ and a center $c \in C$ is larger than $L$, we delete the edge $e(u, c)$ between $u$ and $c$ (step 3 of Algorithm 7). This can be done by calculating all the pairwise distances between $U$ and $C$, which takes time $O(|U||C|d) = O(\Gamma k^2 d)$. Based on the bipartite graph $G(U, C)$ constructed, the relaxed linear programming formulation for fair assignment (called WFA for short) is given in equations (1)-(4), where $a_{u,c}$ is a positive real number denoting the weight units of a point $u \in U$ assigned to the center $c \in C$. In WFA, Equation (1) guarantees that, for each $u \in U$, the sum of the weight units of $u$ assigned to the centers in $C$ equals $w_u$. Equation (2) and Equation (3) guarantee that lower bound and upper bound requirements for fairness are satisfied for each clustering and each protected group $i \in [\Gamma]$, respectively. By solving the WFA linear programming formulation, we can obtain a fractional solution $\vec{a} = \{a_{u,c} : u \in U, c \in C\}$. The following lemma shows that such a fractional solution always exists if $L \geq 6L^*$.

$$\text{WFA:} \sum_{c \in C} a_{u,c} = w_u, \forall u \in U \tag{1}$$

$$\alpha_i \sum_{u \in U} a_{u,c} \leq \sum_{u' \in \mathcal{X}_i \cap U} a_{u',c}, \forall c \in C, i \in [\Gamma] \tag{2}$$

$$\beta_i \sum_{u \in U} a_{u,c} \geq \sum_{u' \in \mathcal{X}_i \cap U} a_{u',c}, \forall c \in C, i \in [\Gamma] \tag{3}$$

$$a_{u,c} \geq 0, \forall u \in U, c \in C \tag{4}$$

**Lemma 13.** *Given an estimation $L$ of the optimal clustering radius, after a preprocessing process in steps 2-3 of Algorithm 7, it can find a feasible fractional solution $\vec{a}$ for WFA if $L \geq 6L^*$.*

*Proof.* The furthest-first strategy used in steps 6-7 of Algorithm 6 guarantees that $C$ is a 2-approximate solution to the standard $k$-center problem. We use $L'^*$ to denote the optimal clustering radius of $P$ for the standard $k$-center problem. It holds trivially that $L'^* \leq L^*$ since there are additional assignment constraints for $(\alpha, \beta)$-fair $k$-center problem. For each data point $p \in P$, we use $o_p$ to denote the optimal clustering center that $p$ is assigned to in the optimal solution for the $(\alpha, \beta)$-fair $k$-center instance. For each optimal clustering center $c_i^* \in C^*$, we use $s(c_i^*)$ to denote the data point in $P$ that is closest to $c_i^*$. For each data point $p \in P$, denote $z_p$ as its closest center in $C$ to $p$. For each data point $p \in P$, let $u_p$ be the center in $U$ such that $p$ is assigned to $u_p$, where $u_p$ and $p$ share the same color and $u_p$ is a data point co-located with $z_p$. Let $C' = \{z_{s(c_1^*)}, z_{s(c_2^*)}, ..., z_{s(c_k^*)}\}$ be the set of the centers in $C$ that are close to the optimal clustering centers in $C^*$. We will show that $C'$ can be used as a bridge to construct a feasible solution $\vec{f}$ for WFA. Firstly, initialize $f_{u,c} = 0$ for each $u \in U$ and $c \in C$. To determine the value of $f_{u,c}$ for each $u \in U$ and $c \in C$, we add a weight unit to $f_{u_p, z_{s(o_p)}}$ for each $p \in P$. In the above procedure, for each color $i \in [\Gamma]$, the total weight units of points with color $i$ assigned to a center $z_{s(c_i^*)}$ is equal to the number of points of this color assigned to $c_i^*$. Thus, the fairness constraints in Equation (2) and Equation (3) can be satisfied. Next, we will show that if $L \geq 6L^*$, for each data point $p \in P$, the distance between $u_p$ and $z_{s(o_p)}$ can be bounded by $6L^*$. According to the triangle inequality, we have $\delta(u_p, z_{s(o_p)}) \leq \delta(u_p, p) + \delta(p, z_{s(o_p)}) \leq 2L'^* + \delta(p, o_p) + \delta(o_p, z_{s(o_p)}) \leq 2L'^* + L^* + \delta(o_p, s(o_p)) + \delta(s(o_p), z_{s(o_p)}) \leq 2L'^* + L^* + L^* + 2L'^* \leq 4L'^* + 2L^* \leq 6L^*$, where the first and fourth inequality follows from the fact that $U$ is a 2-approximate solution to $P$ for the standard $k$-center problem, and the last inequality follows from the fact that $L'^* \leq L^*$. Hence, if the estimation $L$ of the optimal clustering radius is larger than $6L^*$, $\vec{f} = \{f_{u,c} : u \in U, c \in C\}$ is a feasible solution for WFA. $\square$

According to Lemma 13, we know that a fractional solution $\vec{a} = \{a_{u,c} : u \in U, c \in C\}$ can be obtained by solving the linear programming formulations for WFA. Next, we present the residual rounding process, which can construct an integral solution to the weighted fair assignment task. For each $u \in U$, we construct an integral weight as follows.

---

**Algorithm 6** GFK$(P, k, \lambda, \vec{\alpha}, \vec{\beta}, \Gamma)$

---

**Input:** A $k$-center instance $(P, k)$, protected group number $\Gamma$, fairness vectors $\vec{\alpha} = \{\alpha_1, \alpha_2, ..., \alpha_\Gamma\}$, $\vec{\beta} = \{\beta_1, \beta_2, ..., \beta_\Gamma\}$, and a parameter $0 < \lambda < 1$.
**Output:** A set $C \subset \mathbb{R}^d$ of clustering centers, and an assignment function $\phi_f$.
 1: Apply JL-transformation to reduce the dimension $d$ to $O(\log n)$ while preserving $(1 + \epsilon)$ approximation loss on pairwise distances.
 2: $\mathcal{L} = \text{Muli-Scaling}(P, \lambda, 8)$.
 3: Sort the clustering radii in $\mathcal{L}$ in non-decreasing order.
 4: Let $L_i$ be the $i$-th radius in $\mathcal{L}$, and initialize $u_{id} = |\mathcal{L}|$, $l_{id} = 1$, $f(\mathcal{L}) = +\infty$.
 5: Initialize $C = \emptyset$, $U = \emptyset$, and add an arbitrary data point $p \in P$ to $C$.
 6: **for** $i = 1$ to $k - 1$ **do**
 7:     Let $q = \arg\max_{p \in P} \delta(p, C)$, and add $q$ to $C$.
 8: For each $p \in P$, denote $s_p$ as the closest center in $C$ to $p$, and initialize $\phi(p, C) = s_p$.
 9: **for** $i = 1$ to $k$ **do**
10:     Construct a point $c_i^j$ co-located at $c_i$ with color $j$ for each $j \in [\Gamma]$.
11:     Assign a weight $w_i^j$ to $c_i^j$ as $w_i^j = |\{q : \phi(q, C) = c_i, q \in \mathcal{X}_j\}|$.
12:     Add $c_i^j$ to $U$ for each $j \in [\Gamma]$.
13: **while** $l_{id} \leq u_{id}$ **do**
14:     $m = \lfloor (l_{id} + u_{id})/2 \rfloor$.
15:     **if** FairSolver$(U, C, \vec{\alpha}, \vec{\beta}, 6L_m)$ returns a feasible assignment **then**
16:         $\phi' = \text{FairSolver}(U, C, \vec{\alpha}, \vec{\beta}, 6L_m)$.
17:         **for** $c_i \in C$, $u \in U$ **do**
18:             Greedily assign $\phi'(u, c_i)$ points from $\{v : \phi(v, C) = u\}$ to $c_i$ to obtain a new assignment $\phi''$.
19:         **if** $L_m < f(\mathcal{L})$ **then**
20:             $\phi_f = \phi''$, $f(\mathcal{L}) = L_m$.
21:         $u_{id} = m - 1$.
22:     **else**
23:         $l_{id} = m + 1$.
24: **return** $C, \phi_f$.

---

$$w'_u = w_u - \sum_{c \in C} \lfloor a_{u,c} \rfloor. \tag{5}$$

$w'_u$ represents the residual fractional weight of the point $u \in U$ assigned to the centers in $C$. Moreover, for each $c \in C$, we construct $A_c$ as follows.

$$A_c = \sum_{u \in U} a_{u,c} - \lfloor a_{u,c} \rfloor. \tag{6}$$

$A_c$ is the residual fraction of data points that are assigned to $c$ in the fractional solution $\vec{a}$. Finally, for each $i \in [\Gamma]$ and $c \in C$, we construct $A_{c,i}$ as follows.

$$A_{c,i} = \sum_{u \in \mathcal{X}_i \cap U} a_{u,c} - \lfloor a_{u,c} \rfloor. \tag{7}$$

$A_{c,i}$ the residual fraction of data points with color $i$ that are assigned to $c$. Based on the fractional variables constructed, the residual LP formulation (denoted as RES-LP for short) can be defined as follows.

---

**Algorithm 7** FairSolver$(U, C, \vec{\alpha}, \vec{\beta}, L)$

---

**Input:** A weighted point set $U$, a set $C$ of centers, fairness vectors $\vec{\alpha} = \{\alpha_1, \alpha_2, ..., \alpha_\Gamma\}$, $\vec{\beta} = \{\beta_1, \beta_2, ..., \beta_\Gamma\}$, and a radius estimation $L$.
**Output:** An assignment function $\phi$.
1: Initialize an assignment function $\phi(u, c) \leftarrow 0$, $\forall u \in U$ and $c \in C$.
2: Construct a bipartite graph $G(U, C)$ with an edge $e(u, c)$ for each $u \in U$ and $c \in C$.
3: Delete an edge $e(u, c)$ for each $u \in U$ and $c \in C$ if $\delta(u, c) > L$.
4: Solve the linear programming formulation (WFA) in equations (1) - (4).
5: **if** there exists a feasible solution for WFA **then**
6:     Let $\vec{a} = \{a_{u,c} : u \in U, c \in C\}$ be a feasible solution for WFA.
7: **else**
8:     **return** "False".
9: For each $u \in U$ and $c \in C$, set $\phi(u, c) \leftarrow \lfloor a_{u,c} \rfloor$ for each $u \in U$ and $c \in C$.
10: Compute $w'_u$, $A_c$, and $A_{c,i}$ for each $u \in U$, $c \in C$, and $i \in [\Gamma]$ using equations (5) - (7).
11: Construct RES-LP using equations (8) - (11).
12: **while** $\exists u \in U$ such that $\sum_{c \in C} \phi(u, c) \neq w'_u$ **do**
13:     Solve RES-LP, and let $\vec{f} = \{\hat{a}_{u,c} : u \in U, c \in C\}$ be a feasible solution.
14:     For each $\hat{a}_{u,c} = 0$, remove the variable from RES-LP.
15:     For each $\hat{a}_{u,c} = 1$, set $\phi(u, c) = \phi(u, c) + 1$, reduce $A_c$, $A_{c,i}$ by 1, and remove variable $\hat{a}_{u,c}$ from RES-LP.
16:     For each $c \in C$, if $|\{\hat{a}_{u,c} : 0 < \hat{a}_{u,c} < 1, u \in U\}| \leq 3$, remove the respective constraints related to $\hat{a}_{u,c}$ in equation (9) from RES-LP.
17:     For each $c \in C$ and $i \in [\Gamma]$, if $|\{\hat{a}_{u,c} : 0 < \hat{a}_{u,c} < 1, u \in \mathcal{X}_i \cap U\}| \leq 3$, remove the constraints related to $\hat{a}_{u,c}$ in equation (10) from RES-LP.
18: **return** $\phi$.

---

$$\text{RES-LP:} \sum_{c \in C} \hat{a}_{u,c} = w'_u, \forall u \in U \qquad (8)$$

$$\lfloor A_c \rfloor \leq \sum_{u \in U} \hat{a}_{u,c} \leq \lceil A_c \rceil, \forall c \in C \qquad (9)$$

$$\lfloor A_{c,i} \rfloor \leq \sum_{u \in \mathcal{X}_i \cap U} \hat{a}_{u,c} \leq \lceil A_{c,i} \rceil, \forall c \in C, i \in [\Gamma] \qquad (10)$$

$$0 \leq \hat{a}_{u,c} \leq 1, \forall u \in U, c \in C \qquad (11)$$

Let $\hat{a} = \{\hat{a}(u, c) : u \in U, c \in C\}$ be a feasible integral solution to RES-LP. Then, it holds trivially that $a'' = \{\lfloor a_{u,c} \rfloor + \hat{a}(u, c) : u \in U, c \in C\}$ forms a feasible integral solution to WFA according to Claim 3.5 in Bera et al. (2022). Thus, by using Theorem 3.6 in Bera et al. (2019), we can get that Algorithm 7 can output an integral assignment with 7 additive fairness violation.

**Lemma 14.** (Bera et al., 2019) *Given a radius estimation $L$ for Algorithm 7 with $L > 6L^*$, Algorithm 7 can output an integral assignment which violates the fairness constraints by an additive factor of at most 7. Moreover, any weighted point $u \in U$ is assigned to a center $c \in C$ such that $\delta(u, c) \leq L$.*

Putting all these together, Theorem 4 can be proved.

**Theorem 4.** *For the $(\alpha, \beta)$-fair $k$-center problem, there exists an algorithm that outputs a $(8 + \epsilon)$-approximate solution with 7 additive fairness violation in near-linear time in the data size.*

*Proof.* Since Lemma 3 guarantees that there must exist an estimation $L$ of the optimal clustering radius such that $L^* \leq L < (1 + \lambda)L^*$, by performing a binary search on the radii set $\mathcal{L}$ with the summary constructed, we can obtain a solution with radius $8(1 + \lambda)L^*$ and 7 additive fairness violation. As for the running time, constructing the summary can be executed in time $O(ndk + k^2 d\Gamma)$. The multi-scaling process take time $\tilde{O}(nd/\lambda^2)$ according to Lemma 1 and Lemma 2. For each radius searching process, since there are at most $\Gamma k$ data points in the summary,

it takes time $dpoly(k, \Gamma)$ for weighted fair assignments. Finally, since the whole process is executed for $O(\log |\mathcal{L}|) = O(\frac{\log(nd)}{\lambda})$ times using binary search, the total running time for Algorithm 6 is $\tilde{O}(\Gamma ndk/\lambda^2) + O(dLP(k^2\Gamma, k^2\Gamma)) \log(n \log(d))/\lambda$. $\qquad\square$

## A.7 COMPLEMENTARY EXPERIMENTS

### A.7.1 THE RANGE OF ASPECT RATIO ON CLUSTERING DATASETS

In the table below, we report the aspect ratio (or its estimations for large-scale data) for several popular synthetic and real-world datasets from UCI machine learning repository [1] and other clustering tasks (Ren et al., 2022). For datasets with sizes smaller than 43,500, we calculate the exact value of aspect ratio by checking all the pairwise distances of the data points. For datasets with sizes over 43,500, since obtaining the exact value for $\Delta$ requires an enumeration for $O(n^2)$ pairwise distances (which is impractical for large-scale datasets), we use the projection techniques (Cohen-Addad et al., 2022) to give an estimation (with $n^4$-approximation). It can be seen that $\Delta$, compared to the data size, can be significantly large even by several orders of magnitude. For example, on datasets Glass, HF and Hemi, $\Delta$ is 41705, 199 and 157 times larger than the corresponding data sizes, respectively. On dataset KDD, $\Delta$ is 2.08E+15 times larger than the data size. On dataset SIFT, $\Delta$ is 3.01E+15 times larger than the data size.

| Dataset | Aspect Ratio |
|---|---|
| Computer Hardware(209*8) | 35744 |
| Glass(214*9) | 8924873 |
| HF(299*12) | 59532.03 |
| Rasin(900*7) | 13058 |
| Forest Fire(500*9) | 9001.82 |
| Hemi(1,195*7) | 188382 |
| pr3292(2,392*2) | 16868 |
| Iranian Churn(2,850*14) | 17113 |
| KEGG(43,500*16) | 8377932 |
| Skin(245,057*3) | 1.14E+14 |
| SUSY(5,000,000*18) | 1.75E+21 |
| KDD(4,898,431 * 37) | 1.02E+22 |
| SIFT(100,000,000*128) | 3.01E+24 |

Table 3: Aspect ratio for different synthetic and real-world datasets.

### A.7.2 PERFORMANCES WITH VARYING STOPPING CRITERIA FOR TREE CONSTRUCTION

To further accelerate the tree construction, as stated in the parameter settings of experiments, we stop the tree decomposition process when a tree node contains fewer than $\frac{\xi n}{k}$ data points for a fixed $\xi = 0.01$. The influence of $\xi$ on clustering quality and running time is evaluated in Table 4. It can be seen from the table that a smaller $\xi$ increases the running time without significantly influence the clustering quality.

### A.7.3 CLUSTERING PERFORMANCES WITH VARYING PARAMETER $\lambda$

Table 5 shows the influence of different parameter settings on clustering performances of our proposed $k$-center with outliers algorithm.

The parameter $\lambda$ is the parameter to control the accuracy of the estimation for optimal clustering radius during radii set construction process and the number of additional outliers discarded. It can be seen from the tables that smaller $\lambda$ lead to higher running time. However, the clustering quality is not significantly influenced.

### A.7.4 COMPLEMENTARY EXPERIMENTS FOR $k$-CENTER WITH OUTLIERS

Tables 6, 7, and 8 show the results on small datasets. For clustering cost, our algorithm achieves better results on most of the small datasets with an average $7.6\%$ improvements on clustering quality.

---

[1] https://archive.ics.uci.edu/

| Dataset | $\xi$ | cost | time | Dataset | $\xi$ | cost | time |
|---|---|---|---|---|---|---|---|
| | 0.01 | 0.03002 | 44.51 | | 0.01 | 0.05078 | 3.56 |
| | 0.02 | 0.03014 | 35.18 | | 0.02 | 0.05048 | 3.41 |
| | 0.03 | 0.03022 | 35.43 | | 0.03 | 0.05035 | 3.73 |
| | 0.04 | 0.03006 | 38.64 | | 0.04 | 0.05062 | 3.51 |
| | 0.05 | 0.03002 | 37.75 | | 0.05 | 0.05032 | 3.50 |
| COVERTYPE | 0.06 | 0.03021 | 37.37 | SKIN | 0.06 | 0.05086 | 3.78 |
| | 0.07 | 0.03015 | 37.07 | | 0.07 | 0.05008 | 3.26 |
| | 0.08 | 0.03005 | 37.82 | | 0.08 | 0.05016 | 3.60 |
| | 0.09 | 0.03019 | 37.66 | | 0.09 | 0.05030 | 3.11 |
| | 0.10 | 0.03001 | 32.67 | | 0.10 | 0.05005 | 3.11 |

Table 4: Clustering performances on datasets COVERTYPE and SKIN with varying parameter $\xi$ for fixed $k = 50$, $z = 1\%n$ and $\lambda = 0.1$

| Dataset | $\lambda$ | cost | time | Dataset | $\lambda$ | cost | time |
|---|---|---|---|---|---|---|---|
| | 0.10 | 0.03051 | 31.13 | | 0.01 | 0.05026 | 2.91 |
| | 0.20 | 0.03052 | 30.18 | | 0.20 | 0.05029 | 2.41 |
| | 0.30 | 0.03057 | 33.21 | | 0.30 | 0.05028 | 2.85 |
| | 0.40 | 0.03057 | 32.03 | | 0.40 | 0.04959 | 2.81 |
| | 0.50 | 0.02952 | 29.89 | | 0.50 | 0.04883 | 2.93 |
| COVERTYPE | 0.60 | 0.02850 | 28.19 | SKIN | 0.60 | 0.04894 | 2.33 |
| | 0.70 | 0.02859 | 26.82 | | 0.70 | 0.04736 | 2.31 |
| | 0.80 | 0.03092 | 27.39 | | 0.80 | 0.04623 | 2.27 |
| | 0.90 | 0.02757 | 26.61 | | 0.90 | 0.04533 | 2.32 |
| | 1.00 | 0.02715 | 25.90 | | 1.00 | 0.04532 | 2.30 |

Table 5: Clustering performances on dataset COVERTYPE and SKIN with varying $\lambda$ for fixed $k = 50$, $z = 1\%n$ and $\xi = 0.1$

For running time, our algorithm outperforms other algorithms to achieve an average reduction of 20% compared with the other algorithm on small datasets.

Tables 9, 10, and 11 show the results on large-scale datasets. The results show that our algorithm achieves comparable results on clustering cost compared with the state-of-the-art algorithm, with much faster running time than other algorithms. On average, our algorithm is at least 1.52 time faster than the state-of-the-art algorithm.

| Dataset | Method | $k$ | Cost | Time(s) | $z$ | Cost | Time(s) |
|---|---|---|---|---|---|---|---|
| NIPS | Ours | 10 | $0.0350 \pm 0.0014$ | $1.1304 \pm 0.3660$ | 1%n | $0.0301 \pm 0.0009$ | $1.2585 \pm 0.7259$ |
| | Greedy | | $0.0348 \pm 0.0010$ | $1.1999 \pm 0.2746$ | | $0.0310 \pm 0.0011$ | $0.9224 \pm 0.4750$ |
| | Two-Stage | | $0.0428 \pm 0.0023$ | $0.9081 \pm 0.1551$ | | $0.0317 \pm 0.0007$ | $2.4286 \pm 0.6103$ |
| | Sampling | | $0.0362 \pm 0.0012$ | $0.8324 \pm 0.1285$ | | $0.0324 \pm 0.0010$ | $1.2493 \pm 0.2971$ |
| | Ours + DR | | $\mathbf{0.0347 \pm 0.0010}$ | $\mathbf{0.4133 \pm 0.0929}$ | | $\mathbf{0.0300 \pm 0.0008}$ | $0.6744 \pm 0.1628$ |
| | Greedy + DR | | $0.0350 \pm 0.0015$ | $0.5146 \pm 0.5811$ | | $0.0315 \pm 0.0008$ | $\mathbf{0.5359 \pm 0.2526}$ |
| NIPS | Ours | 20 | $0.0320 \pm 0.0009$ | $1.1172 \pm 0.4080$ | 2%n | $\mathbf{0.0186 \pm 0.0004}$ | $1.5796 \pm 0.5465$ |
| | Greedy | | $0.0323 \pm 0.0007$ | $1.0390 \pm 0.5505$ | | $0.0193 \pm 0.0006$ | $1.9033 \pm 0.6459$ |
| | Two-Stage | | $0.0375 \pm 0.0019$ | $1.7551 \pm 0.4380$ | | $0.0214 \pm 0.0011$ | $3.9764 \pm 0.7778$ |
| | Sampling | | $0.0344 \pm 0.0013$ | $1.1060 \pm 0.5229$ | | $0.0203 \pm 0.0004$ | $2.1102 \pm 0.4292$ |
| | Ours + DR | | $\mathbf{0.0318 \pm 0.0009}$ | $\mathbf{0.4508 \pm 0.0324}$ | | $0.0191 \pm 0.0004$ | $0.5167 \pm 0.0384$ |
| | Greedy + DR | | $0.0329 \pm 0.0008$ | $2.6822 \pm 1.1662$ | | $0.0189 \pm 0.0003$ | $\mathbf{0.4573 \pm 0.0132}$ |
| NIPS | Ours | 30 | $0.0306 \pm 0.0008$ | $3.2932 \pm 0.3091$ | 3%n | $0.0141 \pm 0.0005$ | $2.5908 \pm 1.0479$ |
| | Greedy | | $0.0307 \pm 0.0010$ | $3.7879 \pm 0.5326$ | | $0.0139 \pm 0.0003$ | $3.8333 \pm 0.6982$ |
| | Two-Stage | | $0.0314 \pm 0.0004$ | $5.1539 \pm 0.4637$ | | $0.0162 \pm 0.0010$ | $5.1698 \pm 0.9049$ |
| | Sampling | | $0.0322 \pm 0.0009$ | $1.6452 \pm 0.3345$ | | $0.0141 \pm 0.0003$ | $2.2711 \pm 0.7477$ |
| | Ours + DR | | $\mathbf{0.0296 \pm 0.0004}$ | $\mathbf{0.5159 \pm 0.0341}$ | | $\mathbf{0.0135 \pm 0.0002}$ | $0.5077 \pm 0.0346$ |
| | Greedy + DR | | $0.0312 \pm 0.0007$ | $2.6306 \pm 2.0778$ | | $0.0135 \pm 0.0003$ | $\mathbf{0.4733 \pm 0.0053}$ |
| NIPS | Ours | 40 | $0.0293 \pm 0.0006$ | $3.9286 \pm 0.6113$ | 4%n | $\mathbf{0.0108 \pm 0.0001}$ | $2.3744 \pm 0.6358$ |
| | Greedy | | $0.0306 \pm 0.0011$ | $4.0193 \pm 0.6298$ | | $0.0109 \pm 0.0002$ | $3.2523 \pm 0.7811$ |
| | Two-Stage | | $\mathbf{0.0290 \pm 0.0005}$ | $6.4945 \pm 1.2757$ | | $0.0129 \pm 0.0006$ | $5.2224 \pm 0.7551$ |
| | Sampling | | $0.0311 \pm 0.0006$ | $2.8837 \pm 0.4391$ | | $0.0117 \pm 0.0004$ | $2.6015 \pm 0.4733$ |
| | Ours + DR | | $0.0292 \pm 0.0006$ | $0.6567 \pm 0.0092$ | | $\mathbf{0.0108 \pm 0.0001}$ | $0.5524 \pm 0.0297$ |
| | Greedy + DR | | $0.0307 \pm 0.0009$ | $\mathbf{0.5627 \pm 0.0251}$ | | $0.0113 \pm 0.0003$ | $\mathbf{0.5090 \pm 0.0017}$ |
| NIPS | Ours | 50 | $0.0285 \pm 0.0007$ | $2.2380 \pm 0.9067$ | 5%n | $0.0091 \pm 0.0002$ | $2.8030 \pm 0.5926$ |
| | Greedy | | $0.0291 \pm 0.0004$ | $2.2580 \pm 1.1923$ | | $0.0091 \pm 0.0004$ | $2.8543 \pm 1.0815$ |
| | Two-Stage | | $\mathbf{0.0272 \pm 0.0003}$ | $7.2048 \pm 1.4567$ | | $0.0105 \pm 0.0006$ | $4.9429 \pm 0.6144$ |
| | Sampling | | $0.0302 \pm 0.0008$ | $3.1261 \pm 0.4379$ | | $0.0094 \pm 0.0001$ | $2.6312 \pm 0.3222$ |
| | Ours + DR | | $0.0284 \pm 0.0012$ | $0.7757 \pm 0.0203$ | | $\mathbf{0.0088 \pm 0.0002}$ | $0.7254 \pm 0.0557$ |
| | Greedy + DR | | $0.0299 \pm 0.0004$ | $\mathbf{0.7750 \pm 0.0433}$ | | $0.0090 \pm 0.0002$ | $\mathbf{0.5101 \pm 0.0063}$ |

Table 6: Comparison results on dataset NIPS, where $z$ is fixed as 1%n for varying $k$ while $k$ is fixed as 30 for varying $z$

| Dataset | Method | $k$ | Cost | Time(s) | $z$ | Cost | Time(s) |
|---|---|---|---|---|---|---|---|
| SKIN | Ours | 10 | $0.1066 \pm 0.0113$ | $0.9128 \pm 0.3411$ | 1%n | $0.0644 \pm 0.0026$ | $2.8746 \pm 0.6789$ |
| | Greedy | | $0.1026 \pm 0.0097$ | $1.3804 \pm 0.2011$ | | $0.0754 \pm 0.0064$ | $3.8980 \pm 0.3934$ |
| | Two-Stage | | $0.1655 \pm 0.0158$ | $5.0075 \pm 0.1529$ | | $0.0772 \pm 0.0023$ | $32.7118 \pm 3.4150$ |
| | Sampling | | $0.2422 \pm 0.0294$ | $4.0020 \pm 0.7028$ | | $0.2282 \pm 0.0453$ | $10.5591 \pm 1.1310$ |
| | Ours + DR | | $0.1018 \pm 0.0073$ | $\mathbf{0.3116 \pm 0.0142}$ | | $\mathbf{0.0640 \pm 0.0017}$ | $\mathbf{2.1404 \pm 0.1292}$ |
| | Greedy + DR | | $\mathbf{0.0970 \pm 0.0055}$ | $0.7286 \pm 0.0636$ | | $0.0707 \pm 0.0071$ | $2.5127 \pm 0.1082$ |
| SKIN | Ours | 20 | $\mathbf{0.0748 \pm 0.0025}$ | $1.4543 \pm 0.2992$ | 2%n | $\mathbf{0.0593 \pm 0.0046}$ | $3.2244 \pm 0.5385$ |
| | Greedy | | $0.0839 \pm 0.0032$ | $3.0992 \pm 0.4458$ | | $0.0681 \pm 0.0024$ | $5.5423 \pm 0.4404$ |
| | Two-Stage | | $0.0994 \pm 0.0085$ | $16.2383 \pm 4.8136$ | | $0.0616 \pm 0.0038$ | $37.3150 \pm 5.9751$ |
| | Sampling | | $0.2621 \pm 0.0269$ | $6.6021 \pm 0.4143$ | | $0.2205 \pm 0.0255$ | $12.4205 \pm 2.1795$ |
| | Ours + DR | | $0.0769 \pm 0.0052$ | $\mathbf{1.0546 \pm 0.0825}$ | | $0.0628 \pm 0.0080$ | $\mathbf{1.9479 \pm 0.1017}$ |
| | Greedy + DR | | $0.0784 \pm 0.0076$ | $1.9887 \pm 0.1809$ | | $0.0681 \pm 0.0036$ | $2.6051 \pm 0.0269$ |
| SKIN | Ours | 30 | $0.0644 \pm 0.0042$ | $1.7304 \pm 0.0761$ | 3%n | $\mathbf{0.0524 \pm 0.0027}$ | $3.3270 \pm 0.4576$ |
| | Greedy | | $0.0711 \pm 0.0025$ | $4.8860 \pm 0.1578$ | | $0.0624 \pm 0.0039$ | $4.2014 \pm 0.7786$ |
| | Two-Stage | | $0.0827 \pm 0.0082$ | $30.9739 \pm 4.8796$ | | $0.0578 \pm 0.0061$ | $29.6466 \pm 5.2862$ |
| | Sampling | | $0.2632 \pm 0.0297$ | $9.5161 \pm 0.7233$ | | $0.2149 \pm 0.0324$ | $9.7589 \pm 1.4293$ |
| | Ours + DR | | $\mathbf{0.0641 \pm 0.0045}$ | $\mathbf{1.5935 \pm 0.0737}$ | | $0.0552 \pm 0.0012$ | $\mathbf{1.9795 \pm 0.0999}$ |
| | Greedy + DR | | $0.0677 \pm 0.0024$ | $2.4713 \pm 0.2438$ | | $0.0629 \pm 0.0048$ | $2.6725 \pm 0.1074$ |
| SKIN | Ours | 40 | $\mathbf{0.0555 \pm 0.0038}$ | $2.4776 \pm 0.8502$ | 4%n | $\mathbf{0.0478 \pm 0.0012}$ | $2.4209 \pm 0.5078$ |
| | Greedy | | $0.0646 \pm 0.0024$ | $5.6313 \pm 0.6909$ | | $0.0567 \pm 0.0046$ | $2.9170 \pm 0.1662$ |
| | Two-Stage | | $0.0646 \pm 0.0042$ | $49.4164 \pm 4.4296$ | | $0.0531 \pm 0.0034$ | $26.5577 \pm 1.4391$ |
| | Sampling | | $0.2409 \pm 0.0579$ | $13.7834 \pm 2.6777$ | | $0.1767 \pm 0.0344$ | $9.1663 \pm 0.7160$ |
| | Ours + DR | | $0.0587 \pm 0.0020$ | $\mathbf{2.0984 \pm 0.1928}$ | | $0.0530 \pm 0.0045$ | $\mathbf{0.7232 \pm 0.0353}$ |
| | Greedy + DR | | $0.0617 \pm 0.0059$ | $3.1414 \pm 0.2731$ | | $0.0627 \pm 0.0038$ | $1.1399 \pm 0.0079$ |
| SKIN | Ours | 50 | $\mathbf{0.0500 \pm 0.0035}$ | $2.8278 \pm 0.3748$ | 5%n | $\mathbf{0.0457 \pm 0.0025}$ | $1.9389 \pm 0.2229$ |
| | Greedy | | $0.0645 \pm 0.0110$ | $3.8978 \pm 0.1493$ | | $0.0583 \pm 0.0030$ | $3.7051 \pm 0.5779$ |
| | Two-Stage | | $0.0527 \pm 0.0007$ | $52.0802 \pm 8.6786$ | | $0.0512 \pm 0.0019$ | $27.0038 \pm 1.4849$ |
| | Sampling | | $0.2514 \pm 0.0280$ | $19.4643 \pm 4.6017$ | | $0.1687 \pm 0.0347$ | $11.1944 \pm 1.6566$ |
| | Ours + DR | | $0.0535 \pm 0.0021$ | $\mathbf{2.3737 \pm 0.2118}$ | | $0.0504 \pm 0.0033$ | $\mathbf{0.6270 \pm 0.0045}$ |
| | Greedy + DR | | $0.0609 \pm 0.0071$ | $3.6162 \pm 0.1025$ | | $0.0548 \pm 0.0065$ | $1.0130 \pm 0.0078$ |

Table 7: Comparison results on dataset SKIN, where $z$ is fixed as 1%n for varying $k$ while $k$ is fixed as 30 for varying $z$

| Dataset | Method | $k$ | Cost | Time(s) | $z$ | Cost | Time(s) |
|---|---|---|---|---|---|---|---|
| COVERTYPE | Ours | 10 | $0.0575 \pm 0.0023$ | $13.5989 \pm 2.5095$ | 1%n | $0.0373 \pm 0.0011$ | $31.5356 \pm 9.1877$ |
| | Greedy | | $0.0585 \pm 0.0041$ | $17.1253 \pm 2.4813$ | | $\mathbf{0.0372 \pm 0.0022}$ | $34.6518 \pm 3.7889$ |
| | Two-Stage | | $0.0907 \pm 0.0073$ | $37.5894 \pm 7.2899$ | | $0.0617 \pm 0.0033$ | $167.8396 \pm 32.8809$ |
| | Sampling | | $0.1401 \pm 0.0108$ | $102.0333 \pm 19.3175$ | | $0.1431 \pm 0.0186$ | $145.0001 \pm 34.7815$ |
| | Ours + DR | | $0.0595 \pm 0.0035$ | $\mathbf{1.1673 \pm 0.0445}$ | | $0.0356 \pm 0.0006$ | $\mathbf{14.2711 \pm 1.1290}$ |
| | Greedy + DR | | $\mathbf{0.0564 \pm 0.0018}$ | $3.2026 \pm 0.0845$ | | $0.0385 \pm 0.0013$ | $18.6965 \pm 0.3669$ |
| COVERTYPE | Ours | 20 | $0.0435 \pm 0.0021$ | $14.6536 \pm 0.7409$ | 2%n | $\mathbf{0.0354 \pm 0.0011}$ | $19.9475 \pm 2.0730$ |
| | Greedy | | $0.0454 \pm 0.0040$ | $16.9253 \pm 1.7457$ | | $0.0365 \pm 0.0023$ | $26.5433 \pm 1.6679$ |
| | Two-Stage | | $0.0689 \pm 0.0037$ | $89.5877 \pm 26.0475$ | | $0.0532 \pm 0.0018$ | $180.3789 \pm 14.6965$ |
| | Sampling | | $0.1403 \pm 0.0172$ | $91.4262 \pm 19.8675$ | | $0.1319 \pm 0.0317$ | $105.5303 \pm 24.0374$ |
| | Ours + DR | | $0.0448 \pm 0.0015$ | $\mathbf{3.8427 \pm 0.2612}$ | | $0.0360 \pm 0.0015$ | $\mathbf{3.9062 \pm 0.3151}$ |
| | Greedy + DR | | $\mathbf{0.0427 \pm 0.0014}$ | $6.3269 \pm 0.2543$ | | $0.0360 \pm 0.0013$ | $6.0786 \pm 0.0302$ |
| COVERTYPE | Ours | 30 | $0.0374 \pm 0.0011$ | $24.6556 \pm 2.6859$ | 3%n | $0.0356 \pm 0.0014$ | $17.0749 \pm 1.2798$ |
| | Greedy | | $0.0372 \pm 0.0013$ | $25.4423 \pm 3.0428$ | | $0.0355 \pm 0.0021$ | $22.5766 \pm 0.6302$ |
| | Two-Stage | | $0.0601 \pm 0.0039$ | $176.5862 \pm 14.3160$ | | $0.0524 \pm 0.0024$ | $158.0413 \pm 5.7666$ |
| | Sampling | | $0.1481 \pm 0.0155$ | $85.3249 \pm 33.6549$ | | $0.1140 \pm 0.0216$ | $99.9459 \pm 24.9948$ |
| | Ours + DR | | $\mathbf{0.0362 \pm 0.0015}$ | $\mathbf{14.2459 \pm 0.8364}$ | | $\mathbf{0.0344 \pm 0.0004}$ | $\mathbf{2.5261 \pm 0.1810}$ |
| | Greedy + DR | | $0.0375 \pm 0.0009$ | $17.1394 \pm 0.8048$ | | $0.0353 \pm 0.0019$ | $4.4678 \pm 0.0704$ |
| COVERTYPE | Ours | 40 | $0.0339 \pm 0.0009$ | $27.1542 \pm 1.8319$ | 4%n | $0.0349 \pm 0.0017$ | $16.0116 \pm 0.1461$ |
| | Greedy | | $0.0343 \pm 0.0017$ | $28.2664 \pm 1.5727$ | | $\mathbf{0.0345 \pm 0.0021}$ | $19.3929 \pm 0.0378$ |
| | Two-Stage | | $0.0532 \pm 0.0026$ | $203.1495 \pm 25.8802$ | | $0.0507 \pm 0.0027$ | $160.1470 \pm 10.6595$ |
| | Sampling | | $0.1410 \pm 0.0238$ | $95.3402 \pm 20.4499$ | | $0.1067 \pm 0.0198$ | $113.7008 \pm 23.0611$ |
| | Ours + DR | | $\mathbf{0.0329 \pm 0.0010}$ | $\mathbf{17.0986 \pm 0.9284}$ | | $0.0348 \pm 0.0013$ | $\mathbf{1.7397 \pm 0.1126}$ |
| | Greedy + DR | | $0.0351 \pm 0.0020$ | $23.4499 \pm 0.1153$ | | $0.0387 \pm 0.0034$ | $3.6062 \pm 0.0355$ |
| COVERTYPE | Ours | 50 | $0.0307 \pm 0.0007$ | $32.4487 \pm 2.6131$ | 5%n | $\mathbf{0.0338 \pm 0.0017}$ | $16.8582 \pm 0.4661$ |
| | Greedy | | $0.0310 \pm 0.0013$ | $35.8792 \pm 3.8063$ | | $0.0352 \pm 0.0030$ | $21.1230 \pm 1.4801$ |
| | Two-Stage | | $0.0503 \pm 0.0021$ | $237.5156 \pm 40.7197$ | | $0.0465 \pm 0.0028$ | $139.0697 \pm 11.9575$ |
| | Sampling | | $0.1334 \pm 0.0251$ | $110.1734 \pm 39.5945$ | | $0.0859 \pm 0.0108$ | $111.2797 \pm 24.1783$ |
| | Ours + DR | | $\mathbf{0.0296 \pm 0.0004}$ | $\mathbf{23.0864 \pm 2.1924}$ | | $0.0344 \pm 0.0008$ | $\mathbf{1.3543 \pm 0.0907}$ |
| | Greedy + DR | | $0.0317 \pm 0.0011$ | $28.3490 \pm 0.2539$ | | $0.0352 \pm 0.0032$ | $3.1814 \pm 0.0164$ |

Table 8: Comparison results on dataset COVERTYPE, where $z$ is fixed as 1%n for varying $k$ while $k$ is fixed as 30 for varying $z$

| Dataset | Method | $k$ | Cost | Time(s) | $z$ | Cost | Time(s) |
|---|---|---|---|---|---|---|---|
| SUSY | Ours | 10 | 0.0269 ± 0.0013 | 32.5839 ± 0.7688 | 1%n | 0.0229 ± 0.0006 | 78.7815 ± 10.9634 |
| | Greedy | | 0.0277 ± 0.0008 | 38.5677 ± 1.6567 | | **0.0228 ± 0.0003** | 129.7031 ± 11.1350 |
| | Two-Stage | | 0.0353 ± 0.0047 | 223.1164 ± 25.9293 | | 0.0359 ± 0.0016 | 691.0173 ± 133.4923 |
| | Sampling | | 0.0311 ± 0.0012 | 438.5341 ± 36.2850 | | 0.0291 ± 0.0008 | 561.7739 ± 68.3504 |
| | Ours + DR | | 0.0265 ± 0.0008 | **0.8683 ± 0.0558** | | **0.0227 ± 0.0001** | **5.9234 ± 0.5424** |
| | Greedy + DR | | **0.0261 ± 0.0005** | 37.3405 ± 4.1586 | | 0.0232 ± 0.0005 | 44.3394 ± 6.6151 |
| SUSY | Ours | 20 | **0.0240 ± 0.0012** | 57.6690 ± 3.6351 | 2%n | **0.0209 ± 0.0001** | 108.4065 ± 4.5296 |
| | Greedy | | 0.0241 ± 0.0003 | 78.6868 ± 1.6391 | | 0.0213 ± 0.0006 | 116.1560 ± 0.5248 |
| | Two-Stage | | 0.0390 ± 0.0016 | 401.0464 ± 15.3808 | | 0.0336 ± 0.0019 | 736.1188 ± 69.7549 |
| | Sampling | | 0.0311 ± 0.0006 | 615.5734 ± 55.4289 | | 0.0276 ± 0.0012 | 605.9208 ± 36.1592 |
| | Ours + DR | | 0.0243 ± 0.0003 | **2.4991 ± 0.2699** | | **0.0207 ± 0.0002** | **3.5415 ± 0.8077** |
| | Greedy + DR | | 0.0246 ± 0.0004 | 43.2292 ± 4.0222 | | 0.0208 ± 0.0002 | 42.8927 ± 2.9612 |
| SUSY | Ours | 30 | **0.0225 ± 0.0009** | 84.2789 ± 1.5387 | 3%n | **0.0195 ± 0.0002** | 78.7590 ± 1.5889 |
| | Greedy | | 0.0233 ± 0.0005 | 134.7705 ± 36.0326 | | 0.0202 ± 0.0002 | 108.9682 ± 2.6809 |
| | Two-Stage | | 0.0386 ± 0.0023 | 677.4151 ± 45.8826 | | 0.0319 ± 0.0028 | 706.0817 ± 49.4091 |
| | Sampling | | 0.0296 ± 0.0027 | 786.4258 ± 79.7412 | | 0.0263 ± 0.0007 | 622.0476 ± 118.9278 |
| | Ours + DR | | 0.0226 ± 0.0004 | **5.7034 ± 0.7939** | | **0.0195 ± 0.0002** | **2.2782 ± 1.0002** |
| | Greedy + DR | | 0.0226 ± 0.0002 | 37.8490 ± 3.9431 | | 0.0197 ± 0.0004 | 36.8952 ± 3.8674 |
| SUSY | Ours | 40 | **0.0218 ± 0.0002** | 125.0136 ± 9.6780 | 4%n | **0.0187 ± 0.0001** | 83.5672 ± 2.0747 |
| | Greedy | | **0.0218 ± 0.0002** | 135.6640 ± 2.6707 | | 0.0193 ± 0.0003 | 103.5041 ± 1.1838 |
| | Two-Stage | | 0.0357 ± 0.0021 | 827.9813 ± 22.7853 | | 0.0294 ± 0.0021 | 600.7398 ± 28.4158 |
| | Sampling | | 0.0310 ± 0.0023 | 1094.9908 ± 105.4723 | | 0.0232 ± 0.0008 | 608.1590 ± 60.4433 |
| | Ours + DR | | **0.0218 ± 0.0002** | **11.1069 ± 1.4334** | | 0.0188 ± 0.0003 | **2.6306 ± 1.4872** |
| | Greedy + DR | | 0.0221 ± 0.0004 | 36.7406 ± 6.2434 | | 0.0193 ± 0.0004 | 24.6447 ± 5.8975 |
| SUSY | Ours | 50 | 0.0209 ± 0.0003 | 166.6646 ± 8.5655 | 5%n | 0.0184 ± 0.0001 | 79.6801 ± 3.7445 |
| | Greedy | | 0.0214 ± 0.0002 | 182.6342 ± 39.8295 | | 0.0186 ± 0.0004 | 108.9974 ± 3.2804 |
| | Two-Stage | | 0.0377 ± 0.0030 | 981.9786 ± 87.6431 | | 0.0289 ± 0.0021 | 633.1870 ± 56.1982 |
| | Sampling | | 0.0293 ± 0.0013 | 1722.2746 ± 506.4558 | | 0.0221 ± 0.0004 | 570.2540 ± 70.8671 |
| | Ours + DR | | 0.0211 ± 0.0002 | **21.5935 ± 2.0675** | | **0.0183 ± 0.0002** | **2.4698 ± 1.2416** |
| | Greedy + DR | | 0.0219 ± 0.0007 | 43.7314 ± 8.3107 | | 0.0188 ± 0.0003 | 27.0369 ± 4.7323 |

Table 9: Comparison results on dataset SUSY, where $z$ is fixed as 1%n for varying $k$ while $k$ is fixed as 30 for varying $z$

| Dataset | Method | $k$ | Cost | Time(s) | $z$ | Cost | Time(s) |
|---|---|---|---|---|---|---|---|
| HIGGS | Ours | 10 | 0.0398 ± 0.0005 | 99.8499 ± 4.8335 | 1%n | 0.0352 ± 0.0003 | 280.5033 ± 15.3798 |
| | Greedy | | **0.0381 ± 0.0004** | 171.1844 ± 2.8776 | | **0.0351 ± 0.0000** | 328.4120 ± 29.3414 |
| | Two-Stage | | 0.0444 ± 0.0003 | 404.2075 ± 42.8049 | | 0.0460 ± 0.0064 | 2384.6641 ± 160.0638 |
| | Sampling | | 0.0431 ± 0.0004 | 1268.4763 ± 133.4007 | | 0.0389 ± 0.0004 | 1594.1988 ± 135.1107 |
| | Ours + DR | | 0.0394 ± 0.0006 | **4.6879 ± 1.7580** | | 0.0361 ± 0.0007 | **15.7051 ± 0.0101** |
| | Greedy + DR | | 0.0389 ± 0.0006 | 66.8080 ± 7.2656 | | 0.0351 ± 0.0004 | 81.0353 ± 6.7813 |
| HIGGS | Ours | 20 | 0.0369 ± 0.0001 | 183.4142 ± 9.2872 | 2%n | 0.0337 ± 0.0001 | 205.1429 ± 10.3337 |
| | Greedy | | 0.0366 ± 0.0005 | 208.7214 ± 4.2546 | | 0.0333 ± 0.0002 | 272.9133 ± 6.2548 |
| | Two-Stage | | 0.0411 ± 0.0011 | 1667.6194 ± 40.1908 | | 0.0486 ± 0.0074 | 2519.3778 ± 53.4584 |
| | Sampling | | 0.0413 ± 0.0020 | 1338.0345 ± 23.1868 | | 0.0379 ± 0.0004 | 1364.8194 ± 160.5683 |
| | Ours + DR | | 0.0373 ± 0.0005 | **18.7664 ± 5.6960** | | 0.0341 ± 0.0005 | **11.0616 ± 2.9633** |
| | Greedy + DR | | **0.0364 ± 0.0003** | 61.7889 ± 16.4759 | | **0.0332 ± 0.0001** | 58.0903 ± 18.8576 |
| HIGGS | Ours | 30 | 0.0354 ± 0.0003 | 258.7881 ± 5.0032 | 3%n | 0.0325 ± 0.0004 | 224.2594 ± 4.2984 |
| | Greedy | | 0.0356 ± 0.0001 | 298.9308 ± 10.4497 | | 0.0324 ± 0.0004 | 294.9876 ± 11.3865 |
| | Two-Stage | | 0.0578 ± 0.0020 | 2050.4661 ± 688.6593 | | 0.0460 ± 0.0065 | 2280.0545 ± 360.2689 |
| | Sampling | | 0.0394 ± 0.0015 | 1384.4442 ± 38.9095 | | 0.0372 ± 0.0019 | 1488.2014 ± 470.1255 |
| | Ours + DR | | 0.0357 ± 0.0007 | **24.7244 ± 11.8884** | | 0.0329 ± 0.0005 | **6.8583 ± 2.9480** |
| | Greedy + DR | | 0.0354 ± 0.0001 | 53.4478 ± 17.5246 | | **0.0321 ± 0.0001** | 60.1771 ± 13.1299 |
| HIGGS | Ours | 40 | **0.0343 ± 0.0010** | 330.9360 ± 15.7700 | 4%n | 0.0315 ± 0.0001 | 259.5636 ± 36.4299 |
| | Greedy | | 0.0344 ± 0.0002 | 427.5692 ± 41.3140 | | **0.0314 ± 0.0001** | 345.7913 ± 39.4681 |
| | Two-Stage | | 0.0541 ± 0.0048 | 3293.2401 ± 563.8061 | | 0.0427 ± 0.0066 | 2536.0764 ± 231.3456 |
| | Sampling | | 0.0407 ± 0.0019 | 1647.7905 ± 143.2148 | | 0.0361 ± 0.0003 | 1283.7709 ± 126.5786 |
| | Ours + DR | | 0.0348 ± 0.0004 | 38.2540 ± 10.1062 | | 0.0317 ± 0.0004 | **7.2531 ± 3.4337** |
| | Greedy + DR | | 0.0345 ± 0.0002 | 57.6372 ± 11.0293 | | 0.0314 ± 0.0003 | 36.3633 ± 3.8089 |
| HIGGS | Ours | 50 | 0.0340 ± 0.0001 | 387.1140 ± 54.0745 | 5%n | 0.0312 ± 0.0003 | 189.7611 ± 3.3655 |
| | Greedy | | 0.0338 ± 0.0001 | 460.2348 ± 2.9129 | | **0.0306 ± 0.0001** | 264.2276 ± 33.5890 |
| | Two-Stage | | 0.0382 ± 0.0002 | 3719.7063 ± 91.2475 | | 0.0347 ± 0.0005 | 2158.6359 ± 88.5672 |
| | Sampling | | 0.0390 ± 0.0010 | 1291.9443 ± 31.7212 | | 0.0359 ± 0.0011 | 1075.6615 ± 23.4050 |
| | Ours + DR | | 0.0338 ± 0.0002 | **41.1764 ± 5.0006** | | 0.0315 ± 0.0003 | **6.6952 ± 3.7315** |
| | Greedy + DR | | **0.0337 ± 0.0002** | 57.7491 ± 7.1993 | | 0.0308 ± 0.0002 | 49.3977 ± 10.4825 |

Table 10: Comparison results on dataset HIGGS, where $z$ is fixed as 1%n for varying $k$ while $k$ is fixed as 30 for varying $z$

| Dataset | Method | z | Cost | Time |
|---|---|---|---|---|
| SIFT | Ours | 1%n | $0.5058 \pm 0.0024$ | $3729.49 \pm 360.98$ |
| | Greedy | | $0.4716 \pm 0.0002$ | $6527.16 \pm 386.65$ |
| | Ours + DR | | $0.4715 \pm 0.0022$ | $\mathbf{73.81 \pm 5.01}$ |
| | Greedy + DR | | $\mathbf{0.4707 \pm 0.0022}$ | $1720.70 \pm 393.72$ |
| SIFT | Ours | 2%n | $0.5004 \pm 0.0003$ | $3344.86 \pm 105.39$ |
| | Greedy | | $0.4630 \pm 0.0005$ | $6043.58 \pm 47.61$ |
| | Ours + DR | | $\mathbf{0.4623 \pm 0.0019}$ | $\mathbf{32.18 \pm 0.40}$ |
| | Greedy + DR | | $0.4642 \pm 0.0001$ | $807.07 \pm 5.41$ |
| SIFT | Ours | 3%n | $0.4972 \pm 0.0012$ | $3185.32 \pm 41.27$ |
| | Greedy | | $\mathbf{0.4582 \pm 0.0010}$ | $6168.16 \pm 333.91$ |
| | Ours + DR | | $0.4592 \pm 0.0002$ | $\mathbf{14.06 \pm 0.29}$ |
| | Greedy + DR | | $0.4596 \pm 0.0007$ | $816.96 \pm 21.49$ |
| SIFT | Ours | 4%n | $0.4964 \pm 0.0015$ | $3150.67 \pm 62.80$ |
| | Greedy | | $0.4582 \pm 0.0010$ | $6168.16 \pm 333.91$ |
| | Ours + DR | | $0.4553 \pm 0.0005$ | $\mathbf{12.08 \pm 1.50}$ |
| | Greedy + DR | | $\mathbf{0.4543 \pm 0.0018}$ | $872.34 \pm 34.77$ |
| SIFT | Ours | 5%n | $0.4896 \pm 0.0016$ | $3143.28 \pm 17.24$ |
| | Greedy | | $0.4521 \pm 0.0033$ | $6340.72 \pm 719.94$ |
| | Ours + DR | | $0.4537 \pm 0.0004$ | $\mathbf{9.66 \pm 2.05}$ |
| | Greedy + DR | | $\mathbf{0.4513 \pm 0.0001}$ | $879.47 \pm 23.52$ |

Table 11: Comparison results on dataset SIFT with fixed $k = 30$ and varying $z$

A.7.5    EXPERIMENTS FOR INDIVIDUAL FAIR $k$-CENTER

In this subsection, we evaluate the experimental performances of our proposed algorithm for individual fair $k$-center problem.

**Datasets.** For the individual fair $k$-center problem, we evaluate our proposed algorithm (Algorithm 4) on three datasets (Diabetes: $101,765 \times 2$, Bank: $4,520 \times 3$, Census: $32,560 \times 5$) used in Mahabadi & Vakilian (2020); Negahbani & Chakrabarty (2021) for fair comparison. Following the prior work (Negahbani & Chakrabarty, 2021; Mahabadi & Vakilian, 2020), we consider the following numerical attributes for the datasets used in the experiments.

- Bank. This dataset corresponds to information from a Portuguese Bank. We use the attributes of "age", "balance" and "duration-of-account" as fair features.
- Diabetes. This dataset corresponds to the information and outcome regarding patients related to diabetes from 1999 to 2008 at 130 hospitals across US. We use the attributes of "age" and "time-in-hospital" as fair features.
- Census. This dataset corresponds to 1994 US census. We use the attributes of "age", "fnlwgt", "education.num", "capital.gain", "hours.per.week" as fair features.

**Algorithms.** We evaluate the empirical performance of different algorithms summarized as follows. (1) Local Search (denoted as LS for short): This is the local search algorithm proposed in Mahabadi & Vakilian (2020). Following the same settings in Mahabadi & Vakilian (2020), the swap size $t$ is set to 1 for faster implementation of the algorithm. (2) Sparse LP Rounding (denoted as Sparse-LP for short): This is the linear programming rounding algorithm proposed in Negahbani & Chakrabarty (2021), which is a faster version of the linear programming rounding method that uses a sparsification pre-processing step to improve the running time in practice while incurring only a small loss in fairness and clustering cost. (3) The sampling-based fair clustering algorithm (denoted as SamplingFair for short): This is the fair clustering algorithm proposed in Ebbens et al. (2025), which guarantees a 10-fairness approximation with $(2 + \epsilon)$-approximation on clustering quality. (4) Our individual fair $k$-center algorithm described in Algorithm 4 (denoted as Ours for short). For fair comparison, we set parameters according to the experimental sections in Mahabadi & Vakilian (2020); Negahbani & Chakrabarty (2021). In our experiments, following the same settings for $k$-center with outliers problem, for faster implementation of multi-scaling process, we stop the tree decomposition process when constructing a tree if the number of data points within a tree node is smaller than $\frac{0.01n}{k}$.

**Experimental Setup.** In our experiments, each algorithm is executed for 10 times, and we report the average results for clustering radius, fairness and running time. In previous work Mahabadi & Vakilian (2020); Negahbani & Chakrabarty (2021), for each dataset, a set of samples with size 1,000 is taken from the dataset to test the clustering performances for the algorithms. To test the scalability of different algorithms, we conduct our experiments on different sample sizes ranging from 1,000 to the sizes of the original datasets. In all experiments, follow the settings in previous work Mahabadi & Vakilian (2020), the input parameter $\tau$ is the fairness approximation returned by Jung's algorithm (Jung et al., 2020).

**Results.** Figure 2, Figure 3 and Figure 4 show that experimental results on dataset Bank, Census, and Diabetes, respectively. For clustering cost, it can be seen from the figures that, the local search method achieves the best performances on clustering quality. Our proposed method achieves comparable results across all the datasets used in the experiments. On dataset with smaller samples, the clustering costs of our proposed method are much smaller than those of SamplingFair and Sparse-LP methods. On datasets with larger samples and the original datasets, our proposed algorithm still outperform SamplingFair algorithm on clustering quality.

For fairness guarantees, our proposed algorithm achieves comparable results with that of Sparse-LP algorithm, which is the state-of-the-art algorithm with best fairness guarantees. Although our proposed algorithm has fairness violations slightly larger than that of SamplingFair algorithm, the clustering cost of our proposed algorithm is much smaller than that of SamplingFair algorithm.

As for the running time, our algorithm is the fastest among different algorithms. It can be seen that, as the datasizes grow, there is a sharp increase in the running time for LS and Sparse-LP methods, while our proposed multi-scaling method demonstrates significant advantages over these

algorithms. Our proposed method is around 50% faster than SamplingFair algorithm on the original datasets, which shows a better scalability of our proposed multi-scaling and radii set construction techniques.

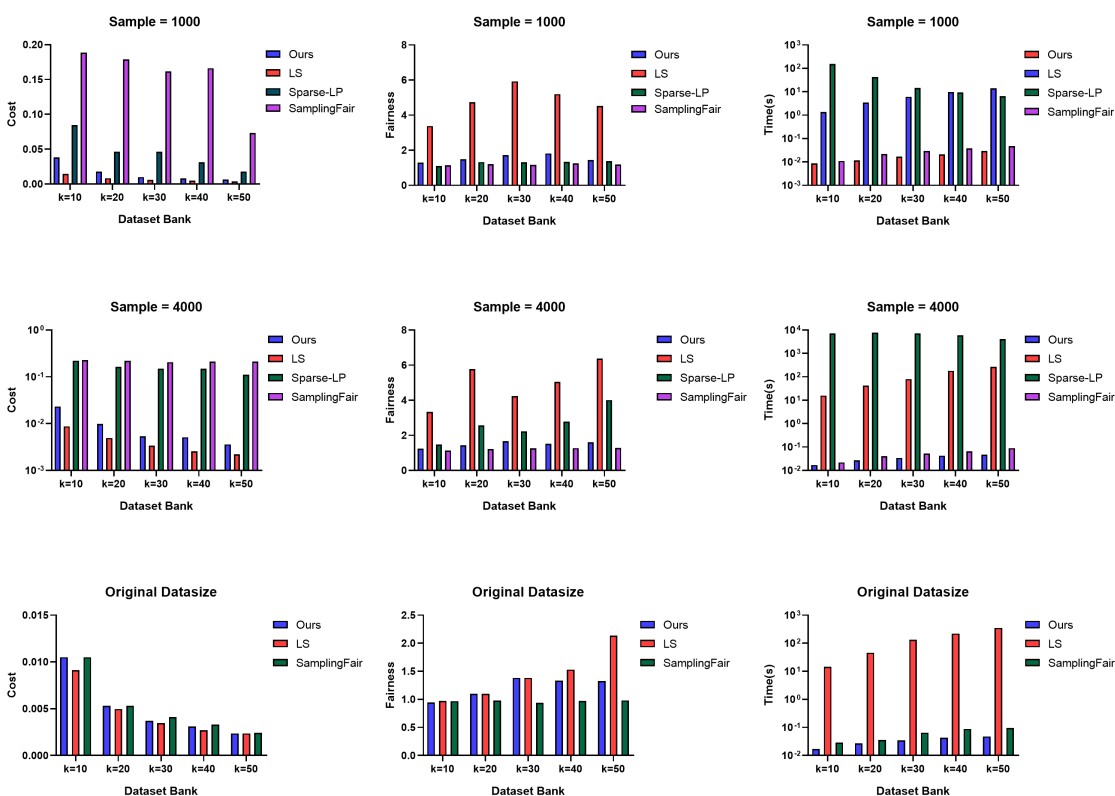

Figure 2: Comparison results for individual fair $k$-center on dataset Bank. For Sparse-LP method, since it requires high memory storage and large computational complexities when handling large-scale datasets, the results are not reported in the figure.

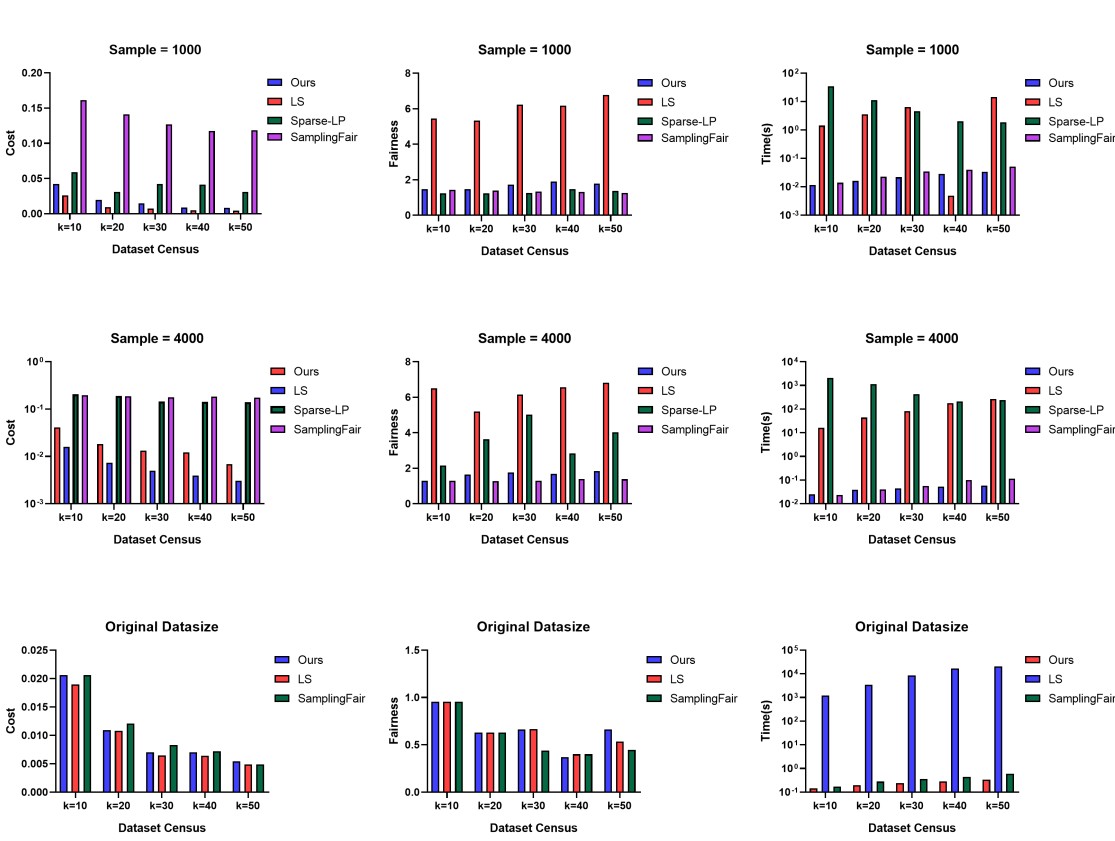

Figure 3: Comparison results for individual fair $k$-center on dataset Census. For Sparse-LP method, since it requires high memory storage and large computational complexities when handling large-scale datasets, the results are not reported in the figure.

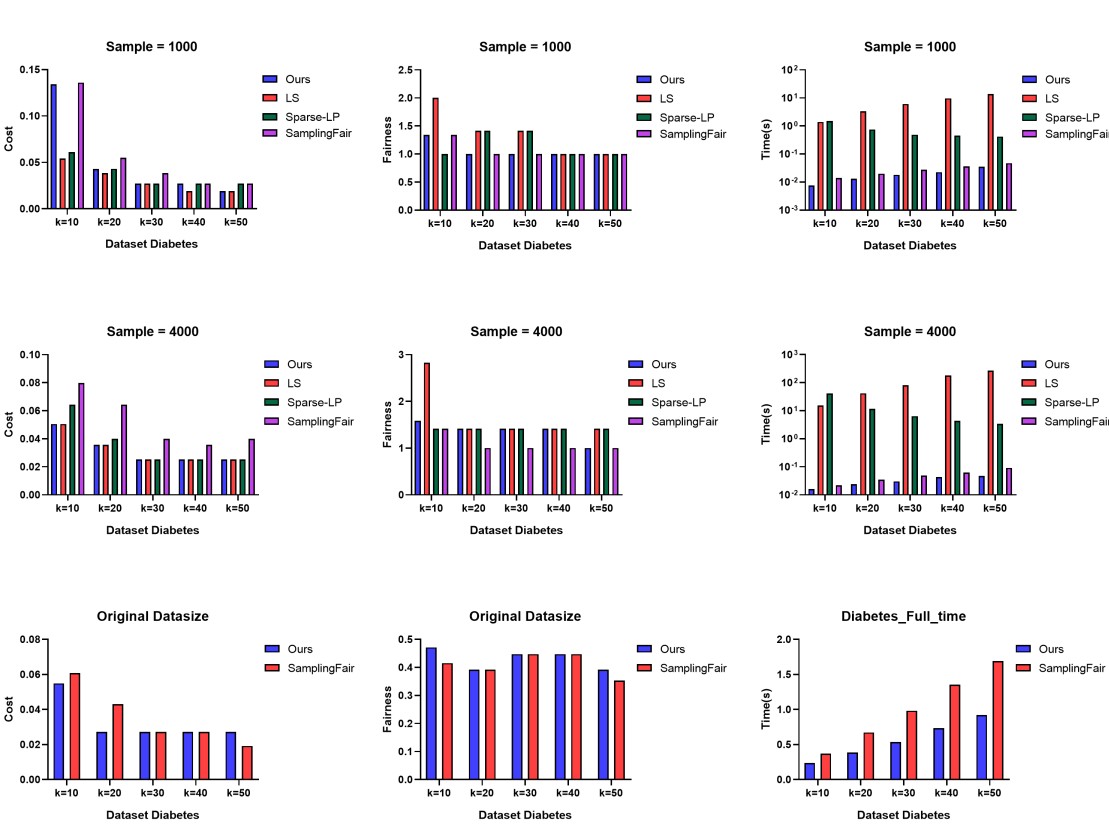

Figure 4: Comparison results for individual fair $k$-center on dataset Diabetes. For Sparse-LP method, since it requires high memory storage and large computational complexities when handling large-scale datasets, the results are not reported in the figure. For LS algorithm, since it cannot output a solution on the original Diabetes dataset within 8 hours, the results are not reported in the figure.

### A.7.6 EXPERIMENTS FOR $(\alpha, \beta)$-FAIR $k$-CENTER

In this subsection, we evaluate the experimental performances of our proposed methods for the $(\alpha, \beta)$-fair $k$-center problem.

**Datasets.** Following the settings in Harb & Shan (2020); Bera et al. (2019), we conduct experiments on 6 real-world datasets: reuters (sample size: 2,500, features: 10, number of groups: 50, protected group features: "author"), victorian (sample size: 4,500, features: 10, number of groups: 45, protected group features: "author"), 4area (sample size: 35,385, features: 8, number of groups: 4, protected group features: "author"), bank (sample size: 4,521, features: 3, number of groups: 5, protected group features: "marital", "default"), census (sample size: 30,000, features: 13, number of groups: 8, protected group features: "marriage", "education").

**Algorithms.** We evaluate the empirical performance of different algorithms summarized as follows. (1) KFC: This is the linear programming method with good scalability proposed in Harb & Shan (2020). Following the settings in Harb & Shan (2020), the parameter $\epsilon$ is f ixed as $\epsilon = 0.1$; (2) LP Rounding (denoted as bera for short): This is the linear programming rounding algorithm proposed in Bera et al. (2019) using $\Theta(n^2)$ LP variables; (3) Greedy Algorithm (denoted as Greedy for short): This is the standard greedy algorithm proposed in Gonzalez (1985), which has no approximation on fairness guarantees; (4) Our $(\alpha, \beta)$-fair $k$-center algorithm described in Algorithm 6 (denoted as Ours for short).

**Experimental Setup.** Following the settings in Bera et al. (2019); Harb & Shan (2020), we evaluate our algorithm and the baselines using clustering cost, fairness violation and running time. The clustering cost (denoted as Cost in our experiments) is defined as the $k$-center cost, or the maximum distance from the points to their assigned centers by the algorithm. The fairness violation (denoted as Fairness in our experiments) is defined as the additive violation of fairness constraint. The running time (denoted as Running time in our experiments) is the total running time of the algorithm in the experiments.

**Results.** Table 12 shows the experimental results on different datasets with fixed $k = 25$. For clustering cost, it can be seen from the table that, our proposed method achieves comparable results compared with the LP rounding methods. For fairness guarantees, our proposed algorithm achieves better results compared with other state-of-the-art algorithms. As for the running time, our algorithm is the fastest among different algorithms. It can be seen that, as the datasize grows, there is a sharp increase on the running time for LP methods, while our proposed multi-scaling method demonstrates much better performances on the running time over these algorithms.

| Dataset | alpha | Cost | | | | Fairness | | | | Running time | | | |
|---|---|---|---|---|---|---|---|---|---|---|---|---|---|
| | | KFC | Greedy | Bera | Ours | KFC | Greedy | Bera | Ours | KFC | Greedy | Bera | Ours |
| reuters | 0.05 | 1.91 | 1.57 | 1.92 | 2.69 | 1.8 | 16.8 | 2 | **1** | 13.59 | 0.11 | 20.37 | **5.11** |
| | 0.2 | **1.75** | 1.57 | 1.78 | 2.69 | 0.8 | 9.8 | 1 | **0.6** | 10.35 | 0.11 | 17.24 | **5.11** |
| | 0.4 | **1.75** | 1.57 | 1.78 | 2.68 | 1 | 3.2 | 1 | **0** | 10.28 | 0.11 | 17.24 | **5.12** |
| victorian | 0.1 | **4.38** | 3.13 | 4.62 | 4.44 | 1.4 | 36.2 | **1** | 1.4 | 21.69 | 0.18 | 60.12 | **5.93** |
| | 0.3 | 4.09 | 3.13 | 4.14 | **4.03** | 1 | 20.6 | 1 | 1 | 15.59 | 0.18 | 58.24 | **5.86** |
| | 0.5 | 4.09 | 3.13 | **3.82** | 4.03 | 1 | 20.6 | **0** | 1 | 15.93 | 0.18 | 49.91 | **5.99** |
| 4area | 0.45 | **9.65** | 9.68 | 9.77 | 9.85 | 1.2 | 17.2 | **0** | 1.2 | 4.72 | 1.58 | 512.12 | **3.72** |
| | 0.6 | **9.65** | 9.68 | 9.77 | 9.84 | 0.8 | 0 | 0 | 0.6 | 4.81 | 1.58 | 482.37 | **3.87** |
| | 0.8 | **9.65** | 9.68 | 9.77 | 9.84 | **0** | **0** | **0** | **0** | 4.72 | 1.59 | 439.28 | **3.71** |
| bank | 0.8 | **19566** | 1271 | N/A | 36922 | 1 | 1 | N/A | **0.6** | 0.42 | 0.16 | TLE | **0.39** |
| | 0.9 | **29143** | 1271 | N/A | 35940 | 1 | 1 | N/A | **0.6** | 0.41 | 0.15 | TLE | **0.32** |
| | 1 | 18256 | 1277 | N/A | **1352** | **0** | **0** | N/A | **0** | **0.33** | 0.16 | TLE | **0.33** |
| census | 0.86 | 367398 | 58067 | N/A | **118959** | **0.2** | 116.2 | N/A | 0.8 | 2.51 | 1.26 | TLE | **2.41** |
| | 0.9 | 367398 | 58067 | N/A | **127713** | **0.4** | 12.8 | N/A | **0.4** | 2.62 | 1.27 | TLE | **2.36** |
| | 0.94 | 367398 | 58067 | N/A | **127713** | 0.6 | 1 | N/A | **0.4** | 2.64 | 1.24 | TLE | **2.37** |
| creditcard | 0.6 | **1251856** | 565766 | N/A | 1260210 | **1** | 8 | N/A | 3.2 | 4.61 | 2.18 | TLE | **3.82** |
| | 0.7 | **1251856** | 565766 | N/A | 1260211 | **1.6** | 1.4 | N/A | 2.4 | 4.72 | 2.18 | TLE | **3.84** |
| | 0.8 | **1251856** | 565766 | N/A | 1260211 | 1.2 | 1 | N/A | **1** | 4.66 | 2.16 | TLE | **3.81** |

Table 12: Comparison results for $(\alpha, \beta)$-fair $k$-center problem with fixed $k = 25$. If an algorithm fails to output a solution within 1 hour, the clustering cost is set as "N/A", and the running time is set as "TLE".

## B DISCUSSION ON THE EXTENSION TO THE METRIC SPACES

Our proposed multi-scaling with data reduction can easily been extended to the metric space with slightly worse running time than Euclidean space. For general metric spaces, even if the input already contains all $O(n^2)$ pairwise distances, sorting these $O(n^2)$ values and performing a binary search over candidate radii requires $O(n^2 \log n)$ time. Unlike the Euclidean setting, where random Gaussian projections can be used to estimate the minimum pairwise distance in near-linear time, we are not aware of any comparably efficient procedure for general metrics. Thus, either one assumes prior knowledge of the minimum pairwise distance, or one incurs $O(n^2 \log n)$ preprocessing time.

As for our proposed method, the primary differences between metric space and Euclidean space lies in the construction of the tree. Although, a tree can be constructed in time $O(nd \log^2 n)$ with distortion polynomial $\mathcal{P}_{HST}(n, d) = nd$ in Euclidean space, it is much more challenging for the case in metric space. In such setting, an HST can be built via an MST (Minimum Spanning Tree) construction method in $O(n^2)$ preprocessing time. After establishing the HST, we can then constructs only $\tilde{O}(1)$ candidate radii through multi-scaling, where the overall running time for multi-scaling is dominated by the time for MST construction. The main theorem for HST construction in general metrics is as follows.

**Lemma 15.** (Har-Peled, 2011) *Given a dataset $P$ in a metric space, let $G$ be a complete graph obtained by connecting all the pairwise distances in $P$. A tree $\mathcal{T}$ can be obtained in time $O(n^2)$ with a distortion polynomial $\mathcal{P}_{HST}(n) = O(n)$ by constructing a Minimum Spanning Tree on $G$, and the number of nodes in $\mathcal{T}$ is bounded by $O(n)$.*

Based on HST construction, we can obtain the following result in general metrics.

**Theorem 5.** *Let $\mathcal{A}$ be an $\mathcal{A}(r_1)$-approximation algorithm for a constrained $k$-center problem that relies on radius guessing with running time $T(n, k)$ for a fixed radius. By combining $\mathcal{A}$ with multi-scaling, an $(\mathcal{A}(r_1) + \epsilon)$-approximation can be achieved in time $O(n^2/\epsilon^2 + T(n, k) \cdot \frac{\log(n)}{\epsilon})$.*

According to Lemma 15, directly performing a multi-scaling on the given dataset $P$ requires a running time of $O(n^2)$, where the quadratic running time may limit the scalability of the multi-scaling method. Instead, we can construct the HST after data reduction on a compressed dataset. Since data reduction usually compresses the data size from $n$ to $\tilde{O}(k)$, compared to the results in Euclidean space, there is only an additional $\tilde{O}(k)$ factor loss caused by HST construction on the compressed data via MST. The corresponding theorem is as follows.

**Theorem 6.** *Let $T(n, d, k)$ be the running time for data reduction based multi-scaling for Euclidean space. The running time complexity for metric space can be bounded by $\tilde{O}(k \cdot T(n, k))$.*

## C EXPLICIT COMPARISONS WITH EXISTING ALGORITHMS

Table 13 presents the results showing how our framework integrates with concrete state-of-the-art algorithms. For each constrained $k$-center variant, we instantiate algorithm $\mathcal{A}$ with the corresponding SOTA method. For $k$-center with outliers, we select to combine with the $(4+\epsilon, 1+\epsilon)$-approximation (or 3-approximation for single-criteria scenarios (Charikar et al., 2001)) algorithm proposed in Biabani et al. (2024), since it gives the current best approximation on clustering quality with only $(1 + \epsilon)z$ outliers discarded. For individual fair $k$-center problem, we select to combine with the $(2 + \epsilon, 10)$-approximation algorithm proposed in Ebbens et al. (2025) as it provides the fastest near-linear running time in the data size. As shown in the table, our proposed multi-scaling method preserves the approximation guarantees of each algorithm while reducing their radius-guessing overhead (typically $\log \log \Delta$) to a $\Delta$-free $O(\log(n \log d))$ factor. When combined with the data-reduction strategy, the overall running time can be further reduced.

| Problem | Approximation | Time | Constraints | Ref |
|---|---|---|---|---|
| $(k,z)$-center | $3+\epsilon$ | $O(n^2 d \cdot \frac{\log\log(n\Delta)}{\epsilon})$ | - | Charikar et al. (2001) |
| | $2$ | $d\,\mathrm{poly}(n)$ | - | Chakrabarty et al. (2020) |
| | $13+\epsilon$ | $O(nd(k+z) + d(k+z)^2 \log\log(n\Delta)/\epsilon)$ | - | Malkomes et al. (2015) |
| | $5$ | $d\,\mathrm{poly}(k)$ | $\begin{array}{c}\|P_h^*\| = \Omega(n/k)\\ z = \Omega(n/k)\end{array}$ | Huang et al. (2021) |
| | $3+\epsilon$ | $O(n^2 d \cdot \frac{\log(n\log d)}{\epsilon})$ | - | Ours (Multi-Scaling) |
| | $3+\epsilon$ | $O(n^2 d \cdot \frac{\log((k+z)\log d)}{\epsilon})$ | - | Ours (Multi-Scaling with DR) |
| | $(2+\epsilon, O(\log k))$ | $O(\frac{ndk\log\log(n\Delta)}{\epsilon})$ | - | Bhaskara et al. (2019) |
| | $(2, 1+\epsilon)$ | $O(ndk/\epsilon)$ | $O(\frac{k}{\epsilon})$ centers opened | Ding et al. (2019) |
| | $(14+\epsilon, 1+\epsilon)$ | $O\left(\left(\frac{ndk\log k}{\epsilon} + d(\frac{k\log k}{\epsilon})^2\right) \cdot \frac{\log\log(n\Delta)}{\epsilon}\right)$ | - | Chan et al. (2023) |
| | $(4+\epsilon, 1+\epsilon)$ | $O(\frac{ndk^3 \log\log(n\Delta)}{\epsilon^2})$ | - | Biabani et al. (2024) |
| | $(4+\epsilon, 1+\epsilon)$ | $O(\frac{ndk^3 \log(n\log d)}{\epsilon^2})$ | - | Ours (Multi-Scaling) |
| | $(8+\epsilon, 1+\epsilon)$ | $O(\frac{ndk^3 \log(kd\log n)}{\epsilon})$ | - | Ours (Multi-Scaling with DR) |
| Idv-Fair $k$-center | $(O(\log n), 7)$ | $O(dn^5 k^5 \log(n\Delta))$ | - | Mahabadi & Vakilian (2020) |
| | $(2+\epsilon, 3)$ | $O(n^4 kd)$ | - | Negahbani & Chakrabarty (2021) |
| | $(3+\epsilon, 3)$ | high-order polynomial | - | Vakilian & Yalciner (2022) |
| | $(2+\epsilon, 2)$ | $O(n^2 + ndk \cdot \frac{\log\log(n\Delta)}{\epsilon})$ | - | Ebbens et al. (2025) |
| | $(2+\epsilon, 10)$ | $O(ndk\log(n/\delta) + k^2 d/\epsilon)$ | - | Ebbens et al. (2025) |
| | $(2+\epsilon, 10)$ | $O(ndk\log(n/\delta) + kd\log k/\epsilon)$ | - | Ours (Multi-Scaling) |
| | $(4(1+\epsilon), 22)$ | $O(ndk + dk^2 \log^2(n/\eta)/\epsilon)$ | - | Ours (Multi-Scaling with DR) |
| Prop-Fair $k$-center | $(O(1), O(1))$ | $O(ndk\log(n/\delta) + kd\log k)$ | - | Ours (Multi-Scaling) |
| $(\alpha,\beta)$-Fair $k$-center | $4$ | high-order polynomial | 7 additive violation | Bera et al. (2019) |
| | $3+\epsilon$ | $O(ndk + \frac{\log\log(n\Delta)}{\epsilon} \cdot (ndk\Gamma + LP(nk, 3nk)))$ | 0 additive violation in expectation | Harb & Shan (2020) |
| | $3+\epsilon$ | $O(ndk + \frac{\log(n\log d)}{\epsilon} \cdot (ndk\Gamma + LP(nk, 3nk)))$ | 0 additive violation in expectation | Ours (Multi-Scaling) |
| | $8+\epsilon$ | $\tilde{O}(\Gamma ndk/\epsilon^2) + O(dLP(k^2\Gamma, k^2\Gamma)\log(n\log(d))/\epsilon)$ | 7 additive violation | Ours (Multi-Scaling with DR) |

Table 13: Explicit comparison of the results for constrained $k$-center problems. Here, $n$ is the data size, $d$ is dimension, $\Delta$ is aspect ratio, $\eta$ and $\delta$ are the success probability parameters, and $\Gamma$ is the number of protected groups for $(\alpha,\beta)$-fair clustering. $LP(m_1, m_2)$ denotes the time to solve a linear program with $m_1$ variables and $m_2$ constraints. Results on doubling metrics are excluded since the running time is exponentially dependent on $d$. Here, $(k,z)$-center denotes the $k$-center with outliers problem, Idv-Fair $k$-center denotes the individual fair $k$-center problem, and Prop-Fair $k$-center denotes the proportionally fair $k$-center problem.

## D    THE EFFECTIVENESS OF DATA REDUCTION ON EXISTING ALGORITHMS

Table 14 presents the results showing how our data reduction method integrates with concrete $k$-center with outliers algorithms. In the single-criteria setting, running a radius-guessing algorithm $\mathcal{A}$ on the summary incurs at most a constant factor loss in approximation guarantee, while reducing the data size from $n$ to $k + z$. In the bi-criteria setting, data reduction yields essentially the same type of guarantee (again up to a constant-factor loss), and in most known results improves the dominant running-time term by roughly a $\Theta(1/z)$ factor.

For fair clustering variants, the summary is specifically designed to be used together with our multi-scaling and fairness-aware clustering rules. Applying a generic radius-guessing algorithm directly on the summary may violate fairness constraints or lose guarantees, so in these settings the summary is not intended as a standalone replacement.

| Problem | Approximation | Time | Constraints | Ref |
|---|---|---|---|---|
| | $13 + \epsilon$ | $O(nd(k+z) + d(k+z)^2 \cdot \frac{\log\log(n\Delta)}{\epsilon})$ | - | Charikar et al. (2001) |
| | $4$ | $O(nd(k+z) + d\mathrm{poly}(k))$ | - | Chakrabarty et al. (2020) |
| | $13 + \epsilon$ | $O(nd(k+z) + d(k+z)^2 \log\log(n\Delta)/\epsilon)$ | - | Malkomes et al. (2015) |
| | $5$ | $d\mathrm{poly}(k)$ | $\begin{array}{c} |P_h^*| = \Omega(n/k) \\ z = \Omega(n/k) \end{array}$ | Huang et al. (2021) |
| $(k,z)$-center | $3 + \epsilon$ | $O(n^2 d \cdot \frac{\log(n\log d)}{\epsilon})$ | - | Ours (Multi-Scaling) |
| | $3 + \epsilon$ | $O(n^2 d \cdot \frac{\log((k+z)\log d)}{\epsilon})$ | - | Ours (Multi-Scaling with DR) |
| | $(4 + \epsilon, O(\log k))$ | $\tilde{O}(nd + \frac{ndk\log\log(n\Delta)}{z\epsilon})$ | - | Bhaskara et al. (2019) |
| | $(4, 1 + \epsilon)$ | $\tilde{O}(nd + ndk/(z\epsilon))$ | $O(\frac{k}{\epsilon})$ centers opened | Ding et al. (2019) |
| | $(28 + \epsilon, 1 + \epsilon)$ | $\tilde{O}\left(nd + (\frac{ndk}{z\epsilon} + d(\frac{k}{\epsilon})^2) \cdot \frac{\log\log(n\Delta)}{\epsilon}\right)$ | - | Chan et al. (2023) |
| | $(8 + \epsilon, 1 + \epsilon)$ | $\tilde{O}(nd + \frac{ndk^3 \log\log(n\Delta)}{z\epsilon^2})$ | - | Biabani et al. (2024) |
| | $(4 + \epsilon, 1 + \epsilon)$ | $O(\frac{ndk^3 \log(n\log d)}{\epsilon^2})$ | - | Ours (Multi-Scaling) |
| | $(8 + \epsilon, 1 + \epsilon)$ | $O(\frac{ndk^3 \log(kd\log n)}{\epsilon})$ | - | Ours (Multi-Scaling with DR) |

Table 14: Comparison of the results for $k$-center with outliers combined with data reduction technique. Here, $n$ is the data size, $d$ is dimension, $\Delta$ is aspect ratio, $\eta$ and $\delta$ are the success probability parameters. Results on doubling metrics are excluded since the running time is exponentially dependent on $d$. Here, $(k, z)$-center denotes the $k$-center with outliers problem.

## E    THE USE OF LARGE LANGUAGE MODELS (LLMS)

Large Language Models were used solely as a writing assistant to improve grammar, clarity, and fluency of the manuscript. They were not involved in the algorithm design, theoretical analysis, experimental setup, or analysis. All technical contributions, proofs, and experiments were conceived, implemented, and validated entirely by the authors.

