# OpenReview forum: "Removing Aspect Ratio on the Running Time for Constrained k-center Clustering"
_ICLR.cc/2026/Conference — Submitted to ICLR 2026_

### Official Review · Reviewer_4hMR · 2025-10-27

**Soundness:** 3
**Presentation:** 3
**Contribution:** 2
**Rating:** 6
**Confidence:** 4

**Summary:**

Many algorithms for constrained $k$-center problems (such as those with outliers, fairness constraints, etc.) rely on the assumption that a near-optimal radius can be efficiently approximated. Typically, this step incurs a dependence on $\Delta$, the distance aspect ratio of the input. The paper proposes a preprocessing subroutine that eliminates this dependence by producing a list guaranteed to contain a near-optimal radius.
In Euclidean spaces, the proposed subroutine constructs a list of size $O(n \log (nd))$, which, when combined with a standard binary search, results in only $O(\log (n \log d))$ iterations of the downstream algorithms. The preprocessing itself runs in $O(nd \log^2 n)$ time (ignoring $\epsilon$-dependent factors). Consequently, the approach removes the $\Delta$-dependence from the running time of many existing algorithms.

The core idea is based on constructing a Hierarchically Separated Tree (HST) over the input points. For Euclidean settings, such trees can be built in $O(nd \log^2 n)$ time [Har-Peled, 2011], independently of $\Delta$. Once the HST is obtained, the authors introduce a bottom-up bucketing strategy that groups distances associated with each tree node. The key observation is that the distance between points represented by the children of a node is bounded in the HST, allowing both the bucket sizes and the number of buckets to remain independent of $\Delta$.

Additionally, the paper presents specific preprocessing schemes for certain problem settings, achieving faster running times for corresponding algorithms. Finally, the authors conduct an extensive set of experiments demonstrating the empirical speedups achieved by incorporating the proposed subroutine compared to existing approaches.

**Strengths:**

1. The paper proposes a single, unified framework that applies to a variety of constrained $k$-center problems in a consistent manner.

2. A comprehensive set of experiments is presented, clearly demonstrating the benefits and efficiency gains of using the proposed preprocessing framework.

**Weaknesses:**

1. I find it difficult to understand the practical motivation for designing such a preprocessing subroutine. From a theoretical perspective (real RAM model), the motivation is clear: $\Delta$ can be arbitrarily large, making such a subroutine useful. However, the paper’s motivation seems to be based on practical datasets. In such cases, the aspect ratio is typically bounded in terms of $n$, since the input precision is inherently limited. For example, assuming $n$ points with each coordinate represented using $O(n)$ bits is a reasonable practical assumption. In this case, $\Delta = 2^{O(n)}$, which leads to a multiplicative overhead (as mentioned in the paper) of $O(\log \log (n\Delta)) = O(\log n + \log \log n)$—comparable to the overhead obtained using the proposed methodology.

2. The techniques used are quite standard: constructing HSTs and applying bucketing. Similarly, the data reduction approach for $k$-center with outliers follows the classic Gonzalez algorithm.

3. The results appear to primarily target Euclidean spaces. Although the authors provide a short extension to general metrics in the appendix, the final theorem statement is not presented, leaving the precise bounds unclear. Moreover, for general metrics, the input already has $n^2$ distance entries, so sorting and applying binary search is already near-linear time, reducing the apparent benefit of the proposed approach.

**Questions:**

1. The data reduction technique used for $k$-center with outliers seems fairly standard. Could you highlight the challenges and techniques used for data reduction in the other problem variants?

2. Can you state the exact theorem for general metrics? (See Weakness #3.)

3. Could you provide a reference for the existing multiplicative overhead of $O(\log \log (n \Delta))$ mentioned on line 75? This seems surprisingly efficient. Does this result hold in general metrics as well?

4. Figures and Table are difficult to read; the font size is too small to interpret properly.

5. In Table 1, for the $k$-center with outliers row, why is the bicriteria algorithm denoted as $(A'(r_1), A'(r_2))$? Shouldn’t it be $(A'(r), A'(z))$, since the bicriteria refers to the approximation for $r$ and $z$?

---

> ### Author Response · Authors · 2025-11-21
> **Response to Reviewer 4hMR**
>
> We thank the reviewer for the positive ratings and constructive feedbacks. Below, we address the concerns.
>
> **Weakness 1: I find it difficult to understand the practical motivation for designing such a preprocessing subroutine. From a theoretical perspective (real RAM model), the motivation is clear: $\Delta$ can be arbitrarily large, making such a subroutine useful. However, the paper’s motivation seems to be based on practical datasets. In such cases, the aspect ratio is typically bounded in terms of $n$, since the input precision is inherently limited. For example, assuming $n$ points with each coordinate represented using $O(n)$ bits is a reasonable practical assumption. In this case, $\Delta = 2^{O(n)}$, which leads to a multiplicative overhead (as mentioned in the paper) of $O(\log\log(n\Delta)) = O(\log n + \log \log n) $—comparable to the overhead obtained using the proposed methodology.**
>
> Response: Thank you for this thoughtful comment. We agree that under a realistic finite-precision model, where each coordinate is stored with bounded bit-length, the overhead of classical radius guessing with a $\log\log\Delta$ factor and that of our pure multi-scaling scheme are indeed of comparable order. In this sense, pure multi-scaling alone does not yield much better improvements in such settings.
>
> The more practical benefit arises when multi-scaling is combined with our data-reduction technique. For $k$-center with outliers, the radius-search overhead becomes $O(\log(kd\log n))$. In typical applications, $k$ is much smaller than the data size $n$, and $d$ can be reduced to $O(\log n)$ via standard dimensionality-reduction methods. This shifts the dependence in the radius-search complexity from the full data size $n$ to the much smaller parameters $k$. For individual fair and $(\alpha,\beta)$-fair clustering, our framework saves at least an $O(\log n)$ factor in the radius-guessing process, which becomes meaningful for large-scale instances. In the revised version, we have provided additional experiments on data-reduction-based multi-scaling. The results indicate an average of over 8x speedup on large-scale datasets, demonstrating the practical effectiveness of the proposed framework.
>
> **Question 1 and Weakness 2: The data reduction technique used for k-center with outliers seems fairly standard. Could you highlight the challenges and techniques used for data reduction in the other problem variants?**
>
> Response: Thank you for raising this important question. While there is prior work on data reduction for $k$-center with outliers, extending a summary structure to work jointly with multi-scaling and constrained variants introduces two main challenges: (1) after reduction, the distance distribution may change, so it is not obvious that running multi-scaling on the reduced instance still yields a radius close to the true optimum; and (2) constraints such as individual fairness and group fairness can easily be violated when many points are removed.
>
> Classical coreset-style reductions mainly shrink the dataset size but do not reduce the aspect ratio, and the reduced instance may miss critical distances. Thus, the radius-guessing bottleneck remains. In our framework, we first prove that for each constrained variant (such as $k$-center with outliers or individual fair $k$-center), there exists at least one “representative’’ distance, either a pairwise distance or a furthest-point distance, that still approximates the optimal radius after data reduction. These distances serve as anchors for constructing a reliable multi-scaling radius list, where multi-scaling can yield a $\Delta$-free running time while keeping the radius close to optimal.
>
> To handle constraint preservation, we design variant-specific reductions. For individual fair $k$-center, we combine Gonzalez’s furthest-first traversal with randomized sampling so that every point retains a representative within its fair radius, preserving individual fairness with bounded loss. For $(\alpha,\beta)$-fair $k$-center, we build threshold graphs on the reduced data and apply distance-thresholding so points cannot be matched to centers that are too far away. Together with the LP constraints, this maintains both approximation quality and group fairness.

---

> ### Author Response · Authors · 2025-11-21
> **Additional Clarifications**
>
> **Question 2 and Weakness 3: Can you state the exact theorem for general metrics? (See Weakness #3.)**
>
> Response: Thank you for pointing this out. We apologize that the final theorem for general metric spaces was not stated clearly in the main text. Below, we first clarify the complexity issue in general metrics and then state the precise bounds.
>
> For general metric spaces, even if the input already contains all $O(n^2)$ pairwise distances, sorting these $O(n^2)$ values and performing a binary search over candidate radii requires $O(n^2\log n)$ time. Unlike the Euclidean setting, where random Gaussian projections can be used to estimate the minimum pairwise distance in near-linear time, we are not aware of any comparably efficient procedure for general metrics. Thus, either (i) one assumes prior knowledge of the minimum pairwise distance, or (ii) one incurs $O(n^2 \log n)$ preprocessing time.
>
> Our multi-scaling approach does not improve the inherent $n^2$ barrier, but it avoids sorting all $n^2$ distances directly. It builds an HST via an MST construction method in $O(n^2)$ preprocessing time and then constructs only $\tilde{O}(1)$ candidate radii through multi-scaling. The main theorem for multi-scaling-based constrained clustering in general metrics is as follows.
>
> Theorem 1: Let $\mathcal{A}$ be an $\mathcal{A}(r_1)$-approximation algorithm  for a constrained $k$-center problem that relies on radius guessing with running time $T(n, k)$ for a fixed radius. By combining $\mathcal{A}$ with multi-scaling, an $(\mathcal{A}(r_1) + \epsilon)$-approximation can be achieved in time $O(n^2/\epsilon^2+ T(n, k) \cdot \frac{\log(n)}{\epsilon}).$
>
> Although constructing an HST directly on $n$ points takes $O(n^2)$ time, we can instead construct the HST after data reduction on a compressed dataset of size $\tilde{O}(k)$, thereby reducing the preprocessing time. Compared to the results in Euclidean space, there is an additional $\tilde{O}(k)$ factor loss caused by HST construction on the compressed data via MST. The corresponding theorem is as follows.
>
> Theorem 2: Let $T(n, d, k)$ be the running time for data reduction based multi-scaling for Euclidean space. The running time complexity for metric space can be bounded by $\tilde{O}(k \cdot T(n, k))$.
>
> **Question 3: Could you provide a reference for the existing multiplicative overhead of $O(\log\log(n\Delta))$ mentioned on line 75? This seems surprisingly efficient. Does this result hold in general metrics as well?**
>
> Response: Thank you for raising this question. The multiplicative overhead comes from how the candidate radius range is discretized in radius-guessing–based algorithms. Classical radius-guessing schemes typically incur an $O(\log\Delta)$ overhead by discretizing the feasible radius interval into a geometric sequence of candidate values.
>
> In Euclidean spaces, this overhead can be further reduced using several known techniques. First, the minimum pairwise distance can be approximated (up to a polynomial factor) in $O(nd\log n)$ time using Gaussian projections, as shown by Cohen-Addad et al. in [1] (see Appendix C.6 for details). Second, a greedy furthest-point strategy provides a 2-approximation to the metric diameter in $O(nd)$ time, following Cohen-Addad et al. in [2] (see the tree-embedding section). These two ingredients avoid enumerating the pairwise distances. Third, using the discretization ideas of Guo and Li [3] (see Appendix B), the radius interval can be reduced to $O(\log(n\Delta)/\epsilon)$ candidate radii, and a binary-search argument then lowers the multiplicative overhead to $O(\log\log\Delta/\epsilon)$.
>
> Finally, this refined overhead relies on geometric tools that are specific to Euclidean space (such as Gaussian projections). To the best of our knowledge, there is no comparable near-linear time procedure for estimating the minimum pairwise distance in general metrics. Therefore, this improved $O(\log\log\Delta/\epsilon)$ overhead does not extend to general metric spaces.
>
> [1] Cohen-Addad V, Mirrokni V, Zhong P. Massively Parallel k-Means Clustering for Perturbation Resilient Instancee. Proc. International Conference on Machine Learning. PMLR, 2022: 4180-4201.
>
> [2] Cohen-Addad V, Lattanzi S, Norouzi-Fard A, et al. Fast and accurate k-means++ via rejection sampling. Proc. 34th International Conference on Neural Information Processing Systems. 2020: 16235-16245.
>
> [3] Guo X, Li S. Distributed k-clustering for data with heavy noise. Proc. 32nd International Conference on Neural Information Processing Systems. 2018: 7849-7857.

---

> ### Author Response · Authors · 2025-11-21
> **Final Clarifications**
>
> **Question 4: Figures and Table are difficult to read; the font size is too small to interpret properly.**
>
> Response: Thank you for the constructive feedbacks and we apologize for the inconvenience. In the revised version, we have adjusted all figures and tables  to ensure that every element is readable.
>
> **Question 5: In Table 1, for the k-center with outliers row, why is the bicriteria algorithm denoted as $(\mathcal{A}(r_1),\mathcal{A}(r_2))$? Shouldn’t it be $(\mathcal{A}(r), \mathcal{A}(z))$, since the bicriteria refers to the approximation for  $r$ and $z$?**
>
> Response: Thanks for the great suggestion. We have corrected the notation to $(\mathcal{A}(r), \mathcal{A}(z))$ to accurately reflect the bi-criteria guarantees for this problem.

---

### Official Review · Reviewer_ULwt · 2025-10-28

**Soundness:** 2
**Presentation:** 2
**Contribution:** 2
**Rating:** 4
**Confidence:** 3

**Summary:**

Optimal radius guessing is a widely used technique for solving the $k$-center clustering problem and its variants. This paper proposes a new optimal radius guessing algorithm—a multi-scaling method—along with problem-dependent acceleration strategies.
This method recursively constructs a hierarchical separation tree to generate a set of candidate radii. The authors prove that this set is guaranteed to contain at least one radius that approximates the optimal value. This candidate set can then be integrated with existing radius-guessing-based algorithms to solve specific $k$-center problems.

The list-generating algorithm returns a set of size $O(n\log n d)$ in $O(nd\log^2(n)/\lambda^2)$ time. To further accelerate this process. The authors first compute a data summary and then generate the radii based on this summary. This strategy reduces the overall time complexity.

This work has two main contributions:

- the size of candidate radii set is independent of the aspect ration $\Delta$. Consequently, the running time of the k-clustering problem also becomes independent of $\Delta$.
- The concept of summaries is introduced to accelerate the radius guessing process.

**Strengths:**

- it is the first to remove the dependence on the aspect ratio $\Delta$ for the optimal radius guessing.

**Weaknesses:**

- The current comparison lacks clarity. Notations (e.g. $\mathcal{A}$) depends on the specific radius-guessing based algorithms selected, making it difficult to determine whether the presented framework truly induces a better algorithm for specific constrained k-center clustering variants.
- For the Individual Fair k-center and (α,β)-Fair k-center, the results are not competitive with those of existing methods.

**Questions:**

- The time complexity of existing radius guessing methods typically includes a factor of  $\log\log(\Delta)$. Could this be really large in practice or just theoretically important?
- The proposed data summary concept appears to have independent research value, particularly as it seems to differ significantly from the well-studied coreset concept. Can this summary be combined with other radius guessing algorithms, and what would be the resulting approximation guarantees and time complexities?

---

> ### Author Response · Authors · 2025-11-21
> **Response to Reviewer ULwt**
>
> We thank the reviewer for the thoughtful comments and constructive feedbacks. Below, we address the concerns.
>
> **Weakness 1: The current comparison lacks clarity. Notations (e.g., $\mathcal{A}$ ) depends on the specific radius-guessing based algorithms selected, making it difficult to determine whether the presented framework truly induces a better algorithm for specific constrained k-center clustering variants.**
>
> Response: Thank you for pointing this out. We use the notation $\mathcal{A}$ to represent a generic radius-guessing-based subroutine because different constrained $k$-center variants rely on different baseline algorithms. The purpose is to present a unified framework without committing to a single algorithmic choice.
>
> To make the comparison clearer, we have added explicit examples in the revised version showing how the framework integrates with concrete state-of-the-art algorithms (see Appendix C for details). For each constrained $k$-center variant, we instantiate algorithm $\mathcal{A}$ with the corresponding SOTA method. As shown in the updated comparison table, our proposed multi-scaling method preserves the approximation guarantees of each algorithm while reducing their radius-guessing overhead (typically $\log\log\Delta$) to a $\Delta$-free $O(\log(n\log d))$ factor. When combined with the data-reduction strategy, the overall running time can be further reduced.
>
> In summary, the notation $\mathcal{A}$ serves only as a placeholder for the chosen baseline. Once $\mathcal{A}$ is fixed, the resulting algorithm keeps nearly the same approximation ratio and achieves a better theoretical runtime bound on high-aspect-ratio scenarios.
>
> **Question 1: The time complexity of existing radius guessing methods typically includes a factor of $\log\log\Delta$. Could this be really large in practice or just theoretically important?**
>
> Response: Thank you for raising this important question. The dependence on $\log\log\Delta$ matters both theoretically and practically.
>
> From a theoretical standpoint, prior work shows that the aspect ratio $\Delta$ can be arbitrarily large in the worst case as pointed out in the literature [1]-[2]. The aspect ratio issue has long been a key bottleneck in clustering algorithms design and other settings. In such regimes, even a $\log\log\Delta$ term becomes a non-negligible multiplicative factor in the number of radius guesses. Removing this dependence eliminates a long-standing bottleneck in radius-guessing-based methods and is exactly what our pure multi-scaling algorithm is designed to achieve.
>
> From a practical standpoint, extremely large aspect ratios can appear in large-scale datasets where very small pairwise distances arise from noise, rounding, or (near) duplicate points. In such cases, existing radius-guessing algorithms must evaluate many more candidate radii (above 8x in our measurements), while our multi-scaling procedure keeps the number of radius guesses consistently small (fewer than 4 across the datasets). This already yields 1.8x speedups for the proposed pure multi-scaling method.
> In the revised version, we additionally evaluate a fast version of our proposed algorithm, which combines multi-scaling with our data-reduction scheme. On large-scale datasets, this full pipeline achieves much better speedups (often over 8x, and up to 30x in heavy noise regimes) compared with Greedy, while maintaining comparable clustering quality. These new experiments show that removing the $\log\log\Delta$ dependence is not only theoretically meaningful but also leads to practical benefits when integrated with data reduction.
>
> [1] Nguyen H L, Nguyen T, Jones M. Fair range k-center[J]. arXiv preprint arXiv:2207.11337, 2022.
>
> [2] Bhattacharjee R, Moshkovitz M. No-substitution k-means clustering with adversarial order. Proc. Algorithmic Learning Theory. PMLR, 2021: 345-366.

---

> ### Author Response · Authors · 2025-11-21
> **Additional Clarifications**
>
> **Question 2: The proposed data summary concept appears to have independent research value, particularly as it seems to differ significantly from the well-studied coreset concept. Can this summary be combined with other radius guessing algorithms, and what would be the resulting approximation guarantees and time complexities?**
>
> Response: Thank you for this insightful question. The proposed data summary can be directly combined with existing radius-guessing-based algorithms only for the $k$-center with outliers problem. In the single-criteria setting, running a radius-guessing algorithm $\mathcal{A}$ on the summary incurs at most a constant factor loss in approximation guarantee, while reducing the data size from $n$ to $k+z$. In the bi-criteria setting, data reduction yields essentially the same type of guarantee (again up to a constant-factor loss), and in most known results improves the dominant running-time term by roughly a $\Theta(1/z)$ factor.
>
> For fair clustering variants, the summary is specifically designed to be used together with our multi-scaling and fairness-aware clustering rules. Applying a generic radius-guessing algorithm directly on the summary may violate fairness constraints or lose guarantees, so in these settings the summary is not intended as a standalone replacement.
>
> Even for $k$-center with outliers, using the summary alone does not remove aspect-ratio dependence: the reduced instance still requires essentially the same number of candidate radii as the original one, because the summary compresses the number of points but not the distance range. Thus, a $\log\log(n\Delta)$–type factor remains in the running time. In the revised version, we include explicit examples and a comparison table in Appendix D showing that summaries alone can accelerate existing algorithms for $k$-center with outliers but retain their $\Delta$-dependent overhead, whereas combining the summary with our multi-scaling approach both preserves the approximation guarantees and yields a runtime completely independent of the aspect ratio.

---

### Official Review · Reviewer_uqRX · 2025-10-30

**Soundness:** 3
**Presentation:** 3
**Contribution:** 3
**Rating:** 6
**Confidence:** 3

**Summary:**

This submission introduces a technique to make the radius guessing step for constrained k-center problems (e.g. k-center with outliers, individually fair k-center) independent of the aspect ratio $\Delta$, which is defined by the ratio between the largest and the smallest pairwise distance between two points.

The proposed multi-scaling method consists of two steps: First all data points get sorted into a hierarchical tree structure, where each point is represented by a leaf and any inner vertex of the tree represents a subset of points which is associated with an upper bound for the diameter. Then the tree is traversed in a bottom-up fashion and the points are partitioned into groups of well-separated clusters. The inter cluster radii of this partition are then used to build a candidate set of possible cluster radii of size $O(n log(nd)\lambda^2)$ in sub-quadratic running time that is independent of $\Delta$, where $d$ is the dimension and $\lambda$ is a parameter that decides the separation of the partition. It is proven that this radius guessing technique only loses an $\epsilon$ in the approximation guarantee for the underlying clustering problems. Later also problem-specific data reduction algorithms are proposed, which further improve upon the running time for the k-center problem with outliers.

The authors conclude the paper with experiments on multiple data sets, which show a runtime improvement in comparison with existing methods for the specific problems and instances, while getting slightly worse costs/fairness.

**Strengths:**

The technical contribution is quite involved and novel. The algorithm performs reasonably well in experiments.

**Weaknesses:**

The writing is quite condensed and particularly the partitioning step of the algorithm is quite confusing and hard to understand due to the large amount of parameters. It would be helpful to discuss the meaning of the parameters in more detail.

The proposed method is only interesting for data sets with very high aspect ratio. In the experiments the speed up in comparison to existing methods is only a small constant factor (often smaller than 2).

**Questions:**

Could you give an intuition why the high distortion $O(nd)$ of the HST is not a problem for the approximation factor?

The text within Figure 1 and Table 1 is too small to be readable.

In Table 6 the highlighted running times are often not the best.

---

> ### Author Response · Authors · 2025-11-21
> **Response to Reviewer uqRX**
>
> We thank the reviewer for the positive rating and constructive feedbacks. Below, we address the concerns.
>
> **Weakness 1: The writing is quite condensed and particularly the partitioning step of the algorithm is quite confusing and hard to understand due to the large amount of parameters. It would be helpful to discuss the meaning of the parameters in more detail.**
>
> Response: Thank you for pointing this out, and we apologize for the confusion caused by the lack of parameter explanations. The partitioning step uses two key parameters $\lambda$ and $\rho$, and we have added detailed descriptions of their roles before the algorithm descriptions in the revised version.
>
> More specifically, $\lambda$ controls the granularity of the partition. A smaller $\lambda$ produces finer partitions, which produces more blocks with smaller gaps between them. This leads to larger candidate radii list and smaller approximation loss. On the contrary, a larger $\lambda$ yields a coarser partition with fewer candidates and faster runtime. As for parameter $\rho$, it serves as a scaling factor used to compensate for the distortion introduced by the HST embedding. It compresses the distance range (through $\gamma = \rho \times \text{distortion}$) so that the additional approximation loss caused by the embedding is absorbed. Our analysis shows that once $\rho$ exceeds a certain threshold, the distortion-induced loss becomes arbitrarily small with only a logarithmic factor loss on the running time.
>
> These explanations have been added before the algorithm description (highlighted in blue color) to make the partitioning step and parameter choices clearer.
>
> **Weakness 2: The proposed method is only interesting for data sets with very high aspect ratio. In the experiments the speed up in comparison to existing methods is only a small constant factor (often smaller than 2).**
>
> Response: Thank you for this valuable observation. Our paper actually proposes two algorithms: (1) a pure multi-scaling method that replaces the radius-guessing step, and (2) an accelerated variant that combines multi-scaling with our data-reduction scheme.
> In the initial submission, the experiments evaluated only the first algorithm (pure multi-scaling), without using data reduction. This was done to isolate the effect of removing radius guessing. Under this controlled setup, the observed speedups over existing methods (including Greedy) are indeed small constant factors, which explains the modest improvements you observed.
>
> To better demonstrate the practical impact of the full framework, we have now added experiments for the second algorithm (multi-scaling + data reduction). On large-scale datasets, this full pipeline achieves over 8x speedups compared with Greedy, and up to 30x speedups in regimes with large outlier budgets, while maintaining comparable clustering quality. These new results show that once data reduction is incorporated, the framework provides both theoretical guarantees (aspect-ratio-free) and much better empirical gains.
>
> At the same time, the main motivation of our work is to establish a unified framework that can theoretically remove the aspect ratio dependency for constrained $k$-center problems. The aspect-ratio dependence is a well-known and non-trivial bottleneck in radius-guessing-based clustering algorithms. This is challenging precisely because most existing algorithms fundamentally rely on radius guessing to obtain theoretical bounds. Our framework is the first approach that provably gives $\Delta$-free algorithms for a broad family of constrained $k$-center problems, while preserving approximation guarantees. This serves as one of the main contributions of this paper.
>
> We have incorporated these clarifications and the new experimental results into the revised version.

---

> ### Author Response · Authors · 2025-11-21
> **Additional Clarifications**
>
> **Question 1: Could you give an intuition why the high distortion of the HST is not a problem for the approximation factor?**
>
> Response: Thank you for the question. The high distortion of the HST does not affect the final approximation ratio because our algorithm explicitly compensates for it using a scaling step and a fine-grained partitioning scheme.
>
> First, we introduce a scaling factor $\rho$ to offset the distortion introduced by the HST embedding. During the tree traversal (step 6 of Algorithm 1), each partition interval is scaled down by $\gamma = \rho \times \text{distortion}$. We have proved that the scaled intervals still cover the true optimal radius, even though distances in the HST may be stretched.
>
> Second, after establishing this scaled range, we perform a fine-grained partitioning controlled by parameter $\lambda$. The radius range is divided into small intervals, with each serving as a candidate radius. A smaller $\lambda$ yields finer intervals, ensuring that despite the distortion in the tree metric, at least one interval remains close to the optimal radius.
>
> Although the HST may have distortion up to $nd$, this only incurs an $O(\log (nd))$ runtime overhead when constructing the intervals. Additionally, it does not affect the approximation ratio. The combination of the scaling factor and the fine partitioning ensures that the candidate radius list remains accurate enough to preserve the final approximation guarantee.
>
> **Question 2: The text within Figure 1 and Table 1 is too small to be readable.**
>
> Response: Thank you for pointing this out, and we apologize for the inconvenience. In the revised version, we have adjusted both Figure 1 and Table 1 to ensure that texts can  be readable.
>
> **Question 3: In Table 6 the highlighted running times are often not the best.**
>
> Response: Thank you for the constructive feedback, and we apologize for the confusion. We have carefully re-checked the data in Table 6 and corrected the highlighted running times to accurately reflect the best results.

---

> > ### Comment · Reviewer_uqRX · 2025-11-23
> >
> > Dear Authors,
> >
> > Thank you very much for your detailed answer and the additional information provided. While some details are now clearer to me, my overall impression has not changed and I will keep my current score.
> >
> > Best wishes,
> > Reviewer

---

### Official Review · Reviewer_dwsA · 2025-11-01

**Soundness:** 4
**Presentation:** 3
**Contribution:** 3
**Rating:** 8
**Confidence:** 4

**Summary:**

The paper studies constrained $k$-center in Euclidean space and eliminates runtime dependence on the aspect ratio $\Delta$. It introduces a multi-scaling preprocessing step that builds a hierarchically well-separated tree (HST) and performs a bottom-up tree mapping to produce a small set of candidate radii. By combining the obtained clustering radii into any radius-guessing routine, the proposed methods can yield nearly the same approximation (or bi-criteria) guarantees without $\Delta$ dependency on the runtime complexities. To further accelerate the multi-scaling process, a problem-specific data-reduction scheme is proposed, where near-linear time results can be achieved with similar guarantees by executing multi-scaling on a compact unweighted “coreset”. Experiments on $k$-center with outliers and fair $k$-center variants show consistent speedups over existing algorithms with essentially comparable cost. In general, this is a good paper with neat and sound techniques.

**Strengths:**

1.  The paper eliminates runtime dependence on the aspect ratio by replacing radius guessing with a multi-scaling process while preserving existing approximation guarantees. The proposed methods can be used to solve a series of constrained $k$-center problems. In my opinion, it  is a timely and practically important task to remove ∆ from the runtime for constrained k-center.

2.  The complexity, HST construction in O(d n log^2 n) and a candidate set of O(n log (nd)/\lambda^2), looks reasonable in high-dimensional regimes where ∆ can be huge.

3. Running on SIFT (100M points) and other large sets is compelling, and the 1.5–1.8× speedups over a strong greedy baseline is impressive.

**Weaknesses:**

1. Overhead remains $O(\log(n logd))$ even after multi-scaling, and it is unclear when data reduction should be used to beat strong $\Delta$-dependent approaches.

2. Although removing the aspect ratio dependency makes good contributions to theoretical analysis, the runtime improvements appear modest in several experimental settings.

3. For most constrained k-center problems the optimal radius is attained at a pairwise distance among the n input points. Hence one can sort the O(n^2) candidate pairwise distances and binary-search them with a feasibility oracle to obtain an approximate solution. Would you give more details about comparing your method with this standard binary-search approach?

4. Although this paper is well-written, there are still a few typos. The authors should fixed them in the future version (minor comments):

a) “Removing Aspect-Ratio Dependence *on* the Running Time” should be *in*

b) Line 34, page 1, "Among various mathematical *formulation*, " should be * formulations*

c) “even in a plane” → “even in the plane”

d) “To our best knowledge” → “To the best of our knowledge”

e) “where points in each Xi share a same color” → “where points in each Xi  share the same color”

f) Standardize “running time” vs “runtime” (pick one).

g) Replace “Due to space limit(s)” with “Due to space limitations” and move longer proofs to an appendix while keeping a proof sketch in the main text.

**Questions:**

1. In general, the paper primarily targets $k$-center. A direct question is: can the multi-scaling framework extend to $k$-median and $k$-means, and what modifications (if any) are needed so that comparable approximation and runtime guarantees can be achieved?

2. Besides the centralized Euclidean setting, can the multi-scaling and summary-based pipelines be extended to distributed or streaming settings and to other related problems?

3. See the weakness section.

---

> ### Author Response · Authors · 2025-11-21
> **Response to Reviewer dwsA**
>
> We thank the reviewer for the positive rating and constructive feedbacks. Below, we address the concerns.
>
> **Question 1: In general, the paper primarily targets $k$-center. A direct question is: can the multi-scaling framework extend to $k$-median and $k$-means, and what modifications (if any) are needed so that comparable approximation and runtime guarantees can be achieved?**
>
> Response: Thank you for raising this important question. The multi-scaling framework has the potential to be extended to $k$-median and $k$-means objectives, but the extension can be non-trivial. For $k$-center, the aspect-ratio dependence arises from the radius-guessing step, where our multi-scaling method directly resolves it. However, for $k$-median/means settings, the dependence usually comes from bad initializations rather than radius guessing, so the multi-scaling procedure does not apply directly.
>
> Extending multi-scaling would require constructing multi-scale candidate partitions rather than candidate radii, together with a mechanism for selecting an initialization that guarantees bounded loss. The separations produced by multi-scaling naturally yield meaningful partitions, suggesting that such an extension is feasible. However, identifying and analyzing partitions that admit provable guarantees remains the main technical challenge. We view this as a promising direction for future work and have added a brief discussion in the revised version.
>
> **Question 2: Besides the centralized Euclidean setting, can the multi-scaling and summary-based pipelines be extended to distributed or streaming settings and to other related problems?**
>
> Response: Thank you for the question. The multi-scaling and summary-based pipeline can indeed be extended beyond the centralized Euclidean setting. A direct application is in distributed clustering. In such settings, each machine can independently apply data reduction and multi-scaling, and then send only its candidate radii to a central coordinator. The collected candidate radii can then be merged to form the final radii list. This preserves approximation guarantees and removes the aspect-ratio dependence in distributed scenarios.
>
> For streaming settings, the extension is more challenging. An extension would require maintaining the HST structure and the multi-scale partitions dynamically as new points arrive or old points expire. While our current framework does not directly provide this capability, the underlying ideas are compatible with incremental or dynamic HST constructions. We believe that such extension represents an interesting and promising direction for future work, and we have included a brief discussion of it in the revised version.
>
> **Weakness 1: Overhead remains $O(\log(n\log d))$  even after multi-scaling, and it is unclear when data reduction should be used to beat strong $\Delta$-dependent.**
>
> Response: Thank you for the comment. While our method still incurs an $O(\log(n\log d))$ overhead, this is already a theoretical improvement over classical radius-guessing approaches, whose complexity grows with $\Delta$. The aspect ratio can become arbitrarily large in the worst case, and in practice often grows significantly in high-precision sensing, GIS data, or datasets with many near-duplicate points. In such settings, removing the $\Delta$-dependence yields a clear benefit.
>
> Regarding data reduction, it is most helpful when the number of clusters $k$ is much smaller than the data size $n$, which is very common in real applications. In this case, the compressed instance allows multi-scaling to run with much lower preprocessing cost while still eliminating all $\Delta$-dependences. We have added more discussions on these points in the revised manuscript.

---

> ### Author Response · Authors · 2025-11-21
> **Additional Clarifications**
>
> **Weakness 2: Although removing the aspect ratio dependency makes good contributions to theoretical analysis, the runtime improvements appear modest in several experimental settings.**
>
> Response: Thank you for the observation. The modest speedups on some benchmark datasets are expected, since these datasets have only moderate aspect ratios, where the $\log\log\Delta$ term is not the dominant runtime factor. Moreover, in the original experiments, we intentionally evaluated pure multi-scaling without data reduction, so that the effect of replacing radius guessing could be isolated without additional accelerations.
>
> In the revised version, we have added experiments that combine multi-scaling with our data-reduction scheme. On large-scale datasets, this full pipeline achieves much better speedups (often over 8x) while maintaining comparable clustering quality. On smaller datasets, it typically yields around 5x speedups. These new results demonstrate that the proposed framework is effective both theoretically and in practical large-scale settings.
>
> **Weakness 3: For most constrained $k$-center problems the optimal radius is attained at a pairwise distance among the $n$ input points. Hence one can sort the $O(n^2)$ candidate pairwise distances and binary-search them with a feasibility oracle to obtain an approximate solution. Would you give more details about comparing your method with this standard binary-search approach?**
>
> Response: Thank you for the question. The standard binary-search approach sorts all $O(n^2)$ pairwise distances and then queries a feasibility oracle (such as existing algorithms) for performing binary searching process. Although simple, this requires $\Theta(n^2\log n)$ time for sorting alone, which can become the dominant cost for most constrained $k$-center problems and limit the scalability for large $n$.
>
> Our method improves on this in two important ways. First, we avoid sorting all $n^2$ distances. In Euclidean space, we construct the HST in $O(nd \log^2 n)$ time, and in general metrics this takes $O(n^2)$. Second, the proposed multi-scaling technique produces only $O(\log(n \log d)/\epsilon)$ candidate radii. This reduces the overhead by an $O(n)$ factor in Euclidean space and by an $O(\log n)$ factor in general metric spaces compared to the standard sorting-based pipeline. These savings lead to better scalability on large datasets.
>
> **Weakenss 4: Regarding the typos**
>
> Response: Thanks for pointing these out. In the revised version, we have carefully corrected the typos to make the presentation better and clearer.

---

> > ### Comment · Reviewer_dwsA · 2025-11-26
> >
> > Thanks for the reply. I am satisfied with your response, particularly the clarification regarding weakness #3, which clears up my confusion. I will maintain my positive score.

---

### Official Review · Reviewer_d9kd · 2025-11-01

**Soundness:** 3
**Presentation:** 3
**Contribution:** 2
**Rating:** 4
**Confidence:** 4

**Summary:**

This paper addresses constrained k-center problems and identifies a key limitation in existing algorithms: their running time depends on the dataset's aspect ratio (Δ), which hinders scalability. To overcome this, the authors propose a multi-scaling method that partitions data based on relative distances and generates a compact set of candidate radii independent of Δ. This eliminates the need for exhaustive radius guessing.

**Strengths:**

The proposed multi-scaling method completely removes the runtime dependency on the aspect ratio (Δ), a significant bottleneck in prior work. This enhances the algorithm's scalability and makes it more suitable for large-scale datasets.

By combining multi-scaling with a novel data reduction technique, the method achieves near-linear runtime in data size while preserving approximation guarantees. This offers an excellent balance between computational efficiency and solution quality, as validated by empirical results.

**Weaknesses:**

While the experimental results indicate that the proposed algorithm generally achieves shorter running times and demonstrates competitive clustering loss, these advantages are not pronounced. This is particularly evident when compared to the Greedy algorithm, which was proposed six years ago. Consequently, I think the significance and novelty of this paper are marginal.

**Questions:**

NaN

---

> ### Author Response · Authors · 2025-11-21
> **Response to Reviewer d9kd**
>
> We thank the reviewer for the thoughtful comments and constructive feedbacks. Below, we address the concerns.
>
> **Weakness 1: While the experimental results indicate that the proposed algorithm generally achieves shorter running times and demonstrates competitive clustering loss, these advantages are not pronounced. This is particularly evident when compared to the Greedy algorithm, which was proposed six years ago. Consequently, I think the significance and novelty of this paper are marginal.**
>
> Response: Thank you for this valuable feedback. Our paper actually proposes two algorithms: (1) a pure multi-scaling method that replaces radius guessing, and (2) an accelerated variant that combines multi-scaling with a data-reduction scheme. In the original submission, the experiments evaluated only the first algorithm (pure multi-scaling). This was done to isolate the effect of replacing radius guessing, but it also means the observed speedups over Greedy on benchmark datasets might be modest.
>
> To directly address your concern, we have now added experiments for the second algorithm (multi-scaling + data reduction). The new results show much better gains. On large-scale datasets, our data-reduction-based multi-scaling achieves over 8x speedups on average compared with the pure Greedy algorithm, while maintaining comparable clustering quality. When the outlier budget $z$ is large, the speedup can reach about 30x, showing that the improvement is much better than a small constant factor.
>
> Regarding significance and novelty, we would like to emphasize that our contribution is not limited to empirical speedups. The main contribution is the first framework that theoretically eliminates aspect-ratio dependence for a broad family of constrained $k$-center problems. Aspect-ratio dependence has been a long-standing and nontrivial bottleneck: existing approaches (including Greedy) fundamentally rely on radius guessing to search for a suitable radius and therefore incur an $O(\log\log \Delta)$ overhead. In scenarios when $\Delta$ is large, this may dominate the runtime. Similar $\Delta$-dependent phenomena also arise in other important settings, such as $k$-clustering in distributed or fully dynamic settings [1]-[4], which highlights that compressing aspect ratio is a genuinely challenging and widely relevant issue rather than a technical detail of a single problem.
>
> Our multi-scaling framework replaces radius guessing with a $\Delta$-free radius construction process while preserving approximation guarantees. Moreover, it is designed to extend across multiple constrained variants, where Greedy is hard to generalize.
>
> We have updated the manuscript to include the new experimental results, making both the theoretical and practical significance of our framework more evident.
>
> [1] Cohen-Addad V, Mirrokni V, Zhong P. Massively parallel k-means clustering for perturbation resilient instancee. Proc. International Conference on Machine Learning. PMLR, 2022: 4180-4201.
>
> [2] Chan T H H, Lattanzi S, Sozio M, et al. Fully dynamic k-center clustering with outliers[J]. Algorithmica, 2024, 86(1): 171-193.
>
> [3] Biabani L, Hennes A, Monemizadeh M, et al. Faster query times for fully dynamic k-center clustering with outliers. Proc. 37th International Conference on Neural Information Processing Systems. 2023: 9226-9247.
>
> [4] Pellizzoni P, Pietracaprina A, Pucci G. k-center clustering with outliers in sliding windows[J]. Algorithms, 2022, 15(2): 52.

---

> > ### Comment · Reviewer_d9kd · 2025-11-26
> >
> > The rebuttal stage serves as an opportunity to discuss the submitted manuscript. If there are misunderstandings from reviewers or inaccuracies in the original paper, this is an appropriate chance to clarify and correct them.
> >
> > While the revised version may include new contributions and potentially be an improved manuscript, it does not alter my assessment of the original submission.

---

### Author Response · Authors · 2025-11-21
**Response to All Reviewers**

We sincerely thank all reviewers for their insightful comments and constructive feedback. Below, we address the concerns separately. To better reflect the suggestions, we have uploaded a revised version of the manuscript, with most changes highlighted in blue. The main updates are:
- Added a discussion of the theoretical contributions in the introduction, emphasizing the theoretical contributions for removing aspect-ratio dependence.
- Expanded the introduction to discuss the role and effect of the proposed data-reduction–based multi-scaling method.
- Conducted new experiments to evaluate the fast version of the proposed algorithm (multi-scaling + data reduction) and demonstrate its practical speedups over existing methods.
- Added explicit discussions on how our multi-scaling framework can be combined with specific existing algorithms for different constrained $k$-center variants.
- Included a more detailed discussion of possible extensions and future work, such as applications to other clustering objectives and computational settings.

---

### Meta-Review · Area_Chair_db8t · 2026-01-05

**Summary:**

1. The empirical runtime improvement compared to the baseline is not significant
2. Clarity on notations, proofs and technical novelties

**Reviewer Concerns:**

The rebuttal clarifies the concern about the clarity of notations, proofs and technical novelties.

The rebuttal provides empirical results for the second proposed algorithm, and claims it addresses the concern about the runtime improvement. However, the new contexts does not address reviewer's concern about the original version.

**Reviewer Scores:**

No change

---

### Decision · Program_Chairs · 2026-01-26

Reject